# Oxytocin and vasopressin within the ventral and dorsal lateral septum modulate aggression in female rats

Vinícius Elias de Moura Oliveira [1] Michael Lukas [2], Hannah Nora Wolf [1], Elisa Durante [1], Alexandra Lorenz [1], Anna-Lena Mayer [1], Anna Bludau [1], Oliver J. Bosch [1], Valery Grinevich [3], Veronica Egger [2], Trynke R. de Jong[1,4] & Inga D. Neumann [1✉]

In contrast to male rats, aggression in virgin female rats has been rarely studied. Here, we established a rat model of enhanced aggression in females using a combination of social isolation and aggression-training to specifically investigate the involvement of the oxytocin (OXT) and arginine vasopressin (AVP) systems within the lateral septum (LS). Using neuropharmacological, optogenetic, chemogenetic as well as microdialysis approaches, we revealed that enhanced OXT release within the ventral LS (vLS), combined with reduced AVP release within the dorsal LS (dLS), is required for aggression in female rats. Accordingly, increased activity of putative OXT receptor-positive neurons in the vLS, and decreased activity of putative AVP receptor-positive neurons in the dLS, are likely to underlie aggression in female rats. Finally, in vitro activation of OXT receptors in the vLS increased tonic GABAergic inhibition of dLS neurons. Overall, our data suggest a model showing that septal release of OXT and AVP differentially affects aggression in females by modulating the inhibitory tone within LS sub-networks.

[1] Department of Neurobiology and Animal Physiology, Behavioural and Molecular Neurobiology, University of Regensburg, Universitaetstraße, Regensburg, Bavaria, Germany. [2] Department of Neurobiology and Animal Physiology, Neurophysiology, University of Regensburg, Regensburg, Germany. [3] Department of Neuropeptide Research in Psychiatry, Central Institute of Mental Health, Medical Faculty Mannheim, University of Heidelberg, Mannheim, Germany. [4] Medische Biobank Noord-Nederland B.V., Groningen, Netherlands. ✉email: inga.neumann@ur.de

Aggressive behavior is expressed by most, if not all, mammalian species, including humans. Typically, aggression is displayed whenever conspecifics have to compete for resources, such as food, territory, or mating partners[1]. When expressed out-of-context or in an exacerbated manner, aggressive behavior becomes disruptive and harms both aggressor and victim. In humans, excessive or pathological aggression as seen, for example, in individuals suffering from conduct or antisocial personality disorder constitutes a severe burden to society[1,2]. To better understand the neurobiology of aggression and to develop potential treatment options, laboratory animal models of aggression have been successfully used for decades[1,3,4]. However, these models were mostly developed in male rodents, whereas females have been rather understudied, except during the physiologically unique period of lactation[5]. Given the fact that (i) females demonstrate disruptive aggression similarly to males[6–8], and (ii) the neurobiology of aggression appears to be sex-dimorphic[5,8–11], the use of non-lactating female animal models is required to increase our understanding of the neurobiology aggression in females and to identify potential targets for the treatment of pathological aggression in both sexes.

In order to study the neurobiological mechanisms underlying the aggression of virgin female Wistar rats (defined here as female aggression), we first established an animal model to robustly enhance their mild levels of aggression[12]. We predicted that a combination of social isolation and aggression-training by repeated exposure to the female intruder test (FIT)[12], i.e., to an unknown same-sex intruder, enhances female aggressiveness, as both social isolation[11,13] and repeated engagement in conflict with conspecifics ("winner effect")[14–16] exacerbate aggression in solitary and aggressive rodent species, independent of sex.

Next, we investigated the role of oxytocin (OXT) and arginine vasopressin (AVP) in the aggressive behavior displayed by female Wistar rats. Both neuropeptides have been associated with various social behaviors including aggression in males and lactating females[17–24], and are known to be affected by social isolation in a sex-dependent manner[11,13]. In this context, we hypothesized that the effects of OXT and AVP on the aggression expressed by female rats are predominantly mediated in the lateral septum (LS) —a target region for neuropeptide actions[24] and known to suppress aggression in both sexes[10,25]. Electrical stimulation of the LS[26,27], and more specifically optogenetic stimulation of septal GABAergic projections to the ventromedial nucleus of the hypothalamus (VMH)[28], reduces aggression in male rodents. In contrast, pharmacological inhibition or lesioning of the LS triggers exaggerated aggression ("septal rage") in male and female hamsters[10,29,30].

Although several pieces of evidence point towards the gating role of the LS in aggression, the underlying mechanisms, especially the involvement of septal OXT and AVP in aggression displayed by female rats, are still unknown. The release of both neuropeptides in the LS has been tied to various social behaviors, including intermale aggression, although the results are somewhat conflicting[24]. For example, in male Wistar rats, attenuated[21], as well as increased[22] AVP release, have been described during the display of abnormal and high[31] aggression, respectively. Regarding the role of the septal OXT system, increased OXT receptor (OXTR) binding has been reported in both dominant male mice[32] and lactating female rats[33]. Importantly, rats have a markedly specific expression of OXTRs and AVP V1a receptors (V1aRs) in two distinct subregions of the LS, i.e., OXTRs are exclusively found in the ventral LS (vLS), whereas V1aRs are predominantly found in the dorsal LS (dLS)[34].

Here, using a reliable rat model of female aggression, we first compared central release patterns of endogenous OXT and AVP between low-aggressive (group-housed; GH) and high-aggressive (isolated and trained; IST) rats by measuring neuropeptide content in microdialysates sampled within the LS or in the cerebrospinal fluid (CSF), during or after exposure to the FIT, respectively. In addition, we assessed septal OXTR and V1aR binding in GH and IST females. Next, we employed pharmacological, chemogenetic, and optogenetic approaches to selectively manipulate central OXT or AVP signaling, specifically in the vLS or dLS, and studied their behavioral consequences in the FIT. Finally, to specifically dissect the neuronal mechanisms within the LS controlling aggressive behavior, we activated septal OXTRs and V1aRs, while recording spontaneous GABAergic inputs to the vLS and dLS GABAergic neurons in vitro, and monitored as well as locally manipulated neuronal activity in rats with opposite levels of aggression in vivo.

## Results

### Social isolation and training reliably enhance aggression in female rats.

Training consisted of daily 10-min exposure to a FIT[12] on 3 consecutive days. GH and isolated non-trained rats (IS) were used as control groups to assess the effects of social isolation and aggression-training, i.e., repeated exposure to FIT over 3 days, respectively (Fig. 1a, Supplementary Fig. 7, and Supplementary Data 4). Both IST and IS females displayed heightened aggression, i.e., increased time spent on keep down, threat, and offensive grooming (Fig. 1b), as well as spend more time and engaged more frequently in attacking (Fig. 1c, d) compared with GH controls (Supplementary Movie 1). We found a major effect of the estrous cycle on aggression mainly reflected by lower levels of aggression in IS females in the proestrus or estrus phase than in the metestrus or diestrus phase (Fig. 1e). Concerning other behaviors, GH females spent more time displaying non-social behaviors, whereas only IST females showed increased self-grooming compared with GH females (Fig. 1f).

To further validate our model of enhanced aggression in females, we treated aggressive IST females with the selective serotonin reuptake inhibitor (SSRI) escitalopram (10 mg/kg, s.c.), as SSRIs typically decrease aggression in rodents[35]. Indeed, escitalopram significantly reduced the time spent on aggression, i.e., on keeping down, threatening and attacking, and increased the latency to attack. Consequently, escitalopram-treated females displayed more defensive and neutral behaviors (Supplementary Fig. 1).

### The display of high aggression in IST rats is accompanied by elevated OXT and decreased AVP release in the brain.

IST, IS, and GH rats were either exposed to the FIT or not (control) to assess the effects of housing conditions as well as an aggressive encounter on the endogenous OXT and AVP systems (Fig. 1a). Elevated post-FIT OXT levels were found in the CSF of IST females compared with GH females (Fig. 2a). In addition, IST and IS rats exhibited higher levels of OXT in the CSF compared with their respective non-FIT control groups. Notably, plasma OXT concentrations did not follow the same pattern (Supplementary Fig. 2). In contrast to OXT, AVP concentrations in the CSF after FIT exposure were lower only in IST females without any effect of housing or training conditions (Fig. 2d). Regarding AVP secretion into the blood, FIT exposure tended to heighten plasma AVP regardless of housing condition, although the magnitude of this effect seemed to be reduced in aggressive IST females (Supplementary Fig. 2).

### Highly aggressive IST rats had low OXTR and V1aR binding in the vLS and dLS, respectively.

We analyzed OXTR and V1aR binding in selected brain regions involved in the social/aggressive behavior network (Supplementary Fig. 2). Among the nine regions

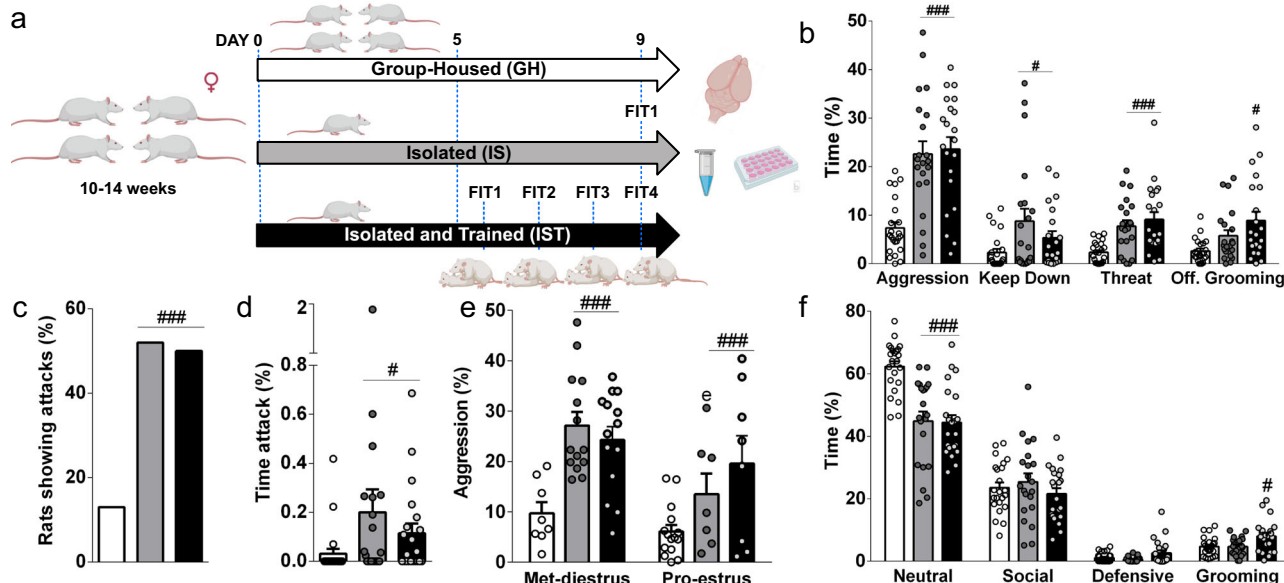

**Fig. 1 Social isolation and training reliably enhance aggression in female Wistar rats, independently of the estrous cycle. a** Scheme illustrating the animal groups and experimental timeline. Female Wistar rats were kept either group-housed (GH) or socially isolated (IS). After 5 days of social isolation, isolated and trained (IST) females were exposed to three consecutive female intruder tests (FIT1 to FIT3), i.e., to aggressive encounters with an unknown same-sex intruder. Black bars correspond to data from FIT4 (test). Partially drawn using https://biorender.com. **b** Both IS and IST rats showed increased total aggression (one-way ANOVA followed by Bonferroni $F_{(2,63)} = 16.26$, $p < 0.0001$), keep down (Kruskal–Wallis test followed by Dunn's $H_3 = 6.69$, $p = 0.035$), threat ($H_3 = 18.98$, $p < 0.0001$) and offensive grooming ($H_3 = 10.8$, $p = 0.005$). **c** IS and IST females engaged more in attacks (Chi-squared test, $X^2 = 40.81$, $p < 0.0001$), and **d** spent a higher percentage of time attacking ($H_3 = 8.8$, $p = 0.012$) than GH females. **e** IS females in the proestrus and estrus (Pro-estrus) phase of the estrous cycle displayed less aggressive behavior than metestrus-diestrus (Met-diestrus) females (two-way ANOVA followed by Bonferroni; factor housing: $F_{(2,60)} = 12.7$, $p < 0.0001$; estrous cycle: $F_{(1,60)} = 8.5$, $p = 0.005$; housing × estrous cycle: $F_{(2,60)} = 1.54$, $p = 0.22$). **f** IST and IS females compensated their increased aggression with decreased neutral behaviors ($F_{(2,63)} = 18.76$, $p < 0.0001$), only IST females spent more time self-grooming ($F_{(2,63)} = 5.79$, $p = 0.005$). All data are shown as mean+SEM. $^{\#}p < 0.05$, $^{\#\#}p < 0.001$, $^{\#\#\#}p < 0.001$ vs GH; $^{e}p < 0.05$ vs met-diestrus. GH: $n = 23$; IS: $n = 21$; IST: $n = 21$.

analyzed, only within the LS OXTRs and V1aRs were affected by housing and aggression-training conditions. In comparison with GH rats, IST females exhibited lower OXTR and V1aR binding in the vLS (Figs. 2b and 2c) and dLS (Figs. 2e and 2f), respectively.

As low glucocorticoid levels have been implicated in the development of intermale aggression in humans and animals[1,8], we assessed plasma corticosterone concentrations after exposure to the FIT. Although there was no effect of housing or training conditions, plasma corticosterone indeed negatively correlated with aggression in females (Supplementary Fig. 2).

**Endogenous OXT promotes, whereas synthetic AVP and OXT reduces aggression in females.** Because of the identified elevated levels of OXT in highly aggressive IST females, we employed pharmacological and chemogenetic methods to either increase OXT availability in the brain of low-aggressive, i.e., GH females, or to block central OXT neurotransmission in the brain of high-aggressive, i.e., IST females. In addition, to make a parallel with male studies reporting serenic effects of OXT[23], we infused IST rats with OXT i.c.v. (Fig. 3a).

In detail, intracerebroventricular (i.c.v.) infusion of synthetic OXT enhanced aggression in GH females (Fig. 3b). Surprisingly, the same treatment decreased aggression in IST females (Fig. 3c), and this anti-aggressive effect of OXT was reflected in all behaviors analyzed (Supplementary Fig. 3). Although these results might seem puzzling at first glance, the chemical similarity between OXT and AVP, which co-evolved from a single nonapeptide[36], is known to result in cross-reactivity between both peptides and each other's receptors in vitro[37] and in vivo, including in the context of aggression[38]. To exclude cross-

reactivity of OXT on V1aRs, we blocked either OXTR or V1aR with selective antagonists (OXTR-A and V1aR-A, i.c.v.)[37] 10 min prior to i.c.v. OXT infusion. Whereas the OXTR-A did not abolish the anti-aggressive effects of OXT, pre-infusion with V1aR-A did (Fig. 3d and Supplementary Fig. 3), clearly indicating that the anti-aggressive effect of synthetic OXT in IST females is mediated via V1aRs.

In order to reveal the involvement of endogenous OXT in aggression displayed by female rats, we first blocked OXTRs in IST females and found that i.c.v. infusion of the OXTR-A before FIT exposure reduced total aggression and threat behavior (Fig. 3e, Supplementary Fig. 3). Aiming to chemogenetically stimulate intracerebral OXT release we infected the hypothalamic paraventricular (PVN) and supraoptic (SON) nuclei of GH female rats with an rAAV to selectively express GqDREADD under the control of an OXT promoter fragment (Fig. 3h). The virus showed a high degree of cell-type efficiency, as 64.37% of OXT cells in the PVN and 75.1% of OXT cells in the SON were positive for mCherry, and specificity as 73.9% of mCherry cells in the PVN and 77.9% in the SON were positive for OXT in accordance with previous data[39]. However, a few mCherry-positive cells outside the PVN (10.8% of total mCherry cells) and SON (9.3% of total mCherry cells) devoid OXT immunosignals. Intraperitoneal application of the DREADD ligand clozapine-N-oxide dihydrochloride increased aggression in diestrus and metestrus GH females, thereby confirming our pharmacological results. Intriguingly, estrus and proestrus GH females showed no increase in aggression. (Fig. 3g and Supplementary Fig. 4).

Finally, as i.c.v. synthetic OXT decreased aggression via activation of V1aRs, and because low levels of AVP were found in the CSF of IST rats, we hypothesized that synthetic AVP i.c.v.

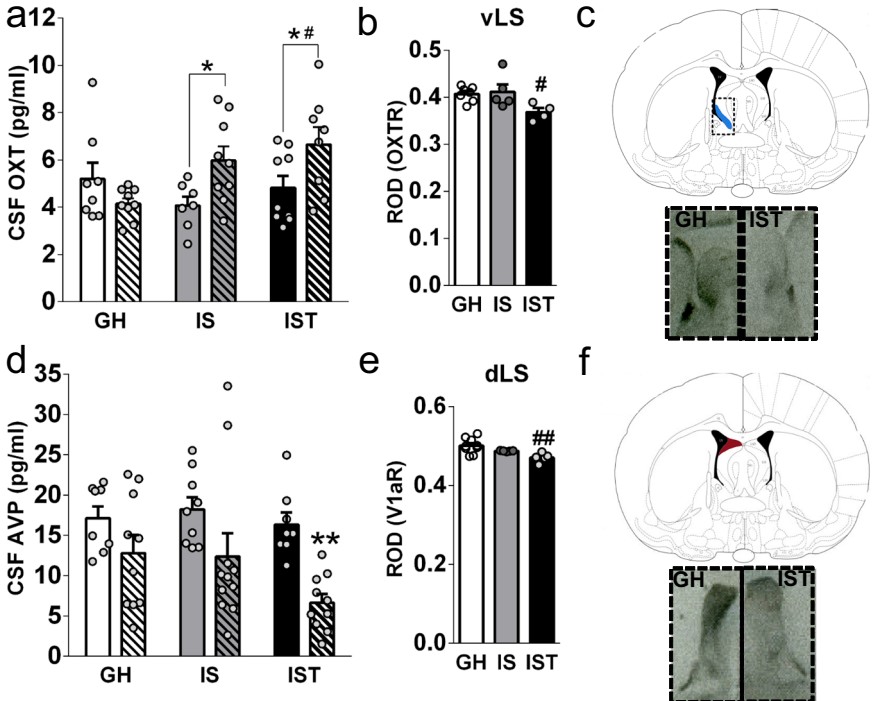

**Fig. 2 The high levels of aggression displayed by isolated and trained (IST) rats are accompanied by high OXT and low AVP concentrations in cerebrospinal fluid (CSF), and low OXT and V1a receptor binding in the lateral septum (LS). a** IS and IST females showed higher concentrations of OXT in CSF immediately after exposure to the female intruder test (FIT) compared with respective control rats (CTRL) (two-way ANOVA followed by Bonferroni: factor FIT: $F_{(1,44)} = 3.91$, $p = 0.054$; housing: $F_{(2,44)} = 1.96$, $p = 0.152$; FIT × housing: $F_{(2,44)} = 4.68$, $p = 0.014$). **b** IST females presented decreased OXT receptor (OXTR) binding, displayed as relative optical density (ROD), in the ventral portion of the LS (vLS) (Kruskal–Wallis test followed by Dunn's: $H_{(3)} = 7.12$, $p = 0.02$. **c** Scheme illustrating localization of OXTR in the vLS, and magnification of representative example autoradiograph (left: GH; right: IST). **d** FIT exposure decreased CSF AVP levels only in IST rats (two-way ANOVA, factor FIT: $F_{(1,50)} = 15.98$, $p = 0.0002$; housing: $F_{(2,50)} = 2.13$, $p = 0.129$; FIT × housing: $F_{(2,50)} = 0.90$, $p = 0.41$). **e** IST females presented decreased V1aR binding, shown as ROD, in the dorsal part of the LS (dLS) ($H_{(3)} = 8.72$, $p = 0.006$). **f** Scheme illustrating localization of V1aR in the dLS and magnification of representative autoradiograph (left: GH, right: IST). All data are presented as mean+SEM. #$p < 0.05$, ##$p < 0.01$ vs GH; *$p < 0.05$, **$p < 0.01$ vs control. Binding: GH: $n = 8$; IS: $n = 4$; IST: $n = 5$; OXT: GH: $n = 17$; IS: $n = 16$; IST: $n = 17$; AVP: GH: $n = 18$; IS: $n = 20$; IST: $n = 18$.

would also reduce aggression in IST females. Indeed, the elevation of AVP availability prior to the FIT resulted in decreased total aggression, keeping down, threatening behaviors as well as time spent on attacks (Fig. 3f and Supplementary Fig. 3).

Taken together, these results demonstrate that activation of OXTRs has a pro-aggressive effect, which is most likely mediated by endogenous OXT. In contrast, both synthetic OXT and particularly synthetic AVP showed anti-aggressive effects mediated via activation of V1aRs.

**The pro-aggressive effect of OXT is mediated in the vLS.** Based on the fact that OXTR binding was exclusively reduced in the vLS of IST females (Fig. 2b), and that the LS is known to regulate aggression[26–28], we hypothesized that the pro-aggressive effect of endogenous OXT is mediated in the vLS. Thus, we used local in vivo microdialysis, neuropharmacological and optogenetic approaches to specifically study the role of the OXT system in the vLS in aggression displayed by females (Fig. 4a).

In vivo monitoring of OXT release within the vLS revealed an increased release in IST, but not GH, rats during FIT exposure, resulting in higher levels of local OXT release observed during the FIT in IST females compared with GH controls (Fig. 4b, Supplementary Data 2–3). In order to test for the involvement of locally released OXT and subsequent OXTR-mediated signaling in the vLS during aggression in female Wistar rats, we bilaterally infused an OXTR-A into the vLS of IST rats 10 min

prior to the FIT. Blockade of vLS OXTRs resulted in decreased total aggression (Fig. 4c), threat and offensive grooming (Supplementary Fig. 5).

Further evidence for the involvement of septal OXT neurotransmission in promoting aggression in female rats was provided by optogenetic stimulation of oxytocinergic axons in the vLS of GH rats. To this end, GH rats were infected with a channelrhodopsin (ChR2) rAAV in the PVN and SON, which is expressed under the control of an OXT promoter fragment (Fig. 4d). Similarly to the GqDREADD rAAV, the ChR2 rAAV showed a high degree of cell-type efficiency, as 64.43% of OXT cells in the PVN and 75.5% of OXT cells in the SON were positive for mCherry; and cell-type specificity, as 74.4% of the mCherry cells in the PVN and 76.99% of mCherry cells in the SON were positive for OXT. The specificity of the virus for targeting OXT neurons has been proven before[40], although in the present experiment a limited number of mCherry-positive cells outside the PVN (10.4% of total mCherry cells) and SON (11.7% of total mCherry cells) devoid OXT immunosignals.

In agreement with the pro-aggressive effects of endogenous OXT (Figs. 2 and 3), blue-light stimulation of vLS OXT axons during the FIT increased the level of aggression in a time- and estrous cycle-dependent manner (see Fig. 4e–g). Cumulative analyses showed a main effect of the viral infection, i.e., only metestrus/diestrus, but not proestrus/estrus, females that expressed ChR2 and received blue-light stimulation exhibited higher aggression compared to controls (Fig. 4g, Supplementary

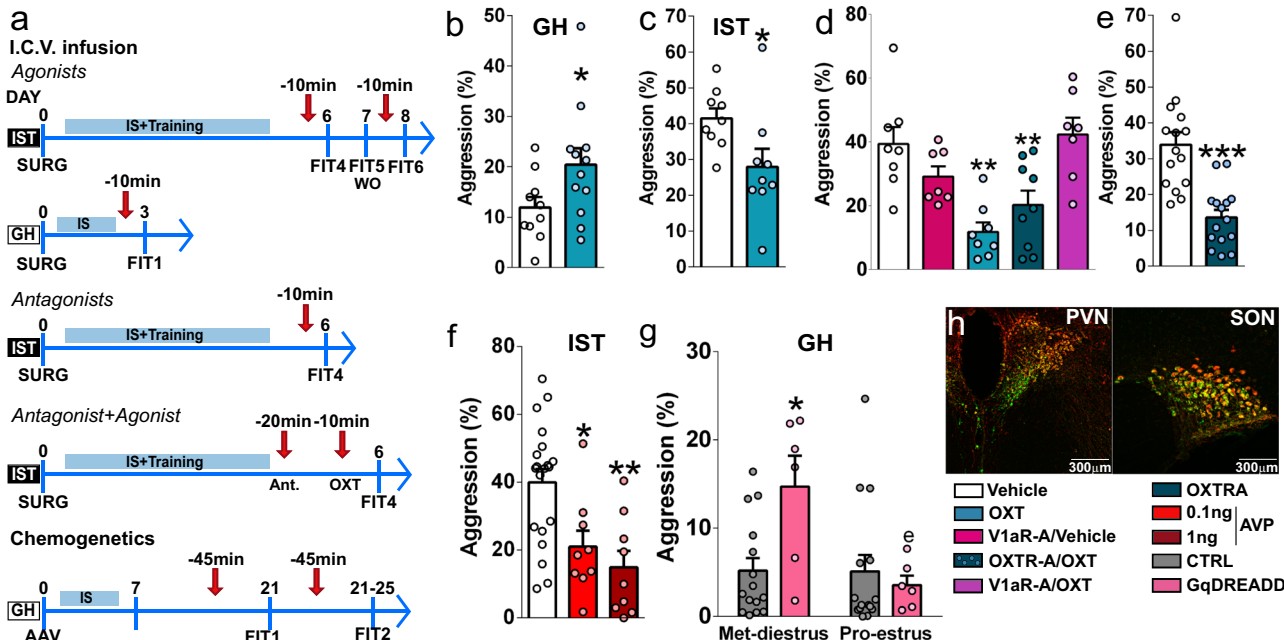

**Fig. 3 Endogenous OXT promotes, whereas synthetic AVP and OXT reduce aggression in females. a** Experimental design for pharmacological and chemogenetic experiments targeting the OXT and AVP systems in isolated and trained (IST) and group-housed (GH) rats. (*AAV* adeno-associated DREADD virus infusion into the hypothalamic paraventricular (PVN) and supraoptic (SON) nuclei; *AVP* arginine vasopressin; arrow=drug infusions; *FIT* female intruder test, *IS* social isolation, *OXT* oxytocin, *OXTR-A* OXT receptor antagonist, *SURG* surgery, *V1aR-A* V1a receptor antagonist, *WO* wash-out). **b** i.c.v. infusion of OXT (50 ng/5 µl) increased aggression in GH (two-tailed Student's t test $t_{(19)} = 2.46$, $p = 0.024$), **c** but decreased aggression in IST females ($t_{(8)} = 2.33$, p = 0.048, data corresponds to FIT4 and 6). **d** i.c.v. infusion of V1aR-A (750 ng/5 µl), but not OXTR-A (750 ng/5 µl), blocked the anti-aggressive effects of OXT in IST females (one-way ANOVA followed by Bonferroni $F_{(3,28)} = 10.1$, $p = 0.001$). Also, the V1aR-A alone did not affect aggression. Both, **e** i.c.v. infusion of OXTR-A ($t_{(28)} = 4.96$, p < 0.0001) and **f** i.c.v. AVP (0.1 or 1 ng/5 µl) reduced total aggressive behavior ($F_{(3,54)} = 7.48$, p = 0.0003) in IST rats. **g** Chemogenetic activation of OXT neurons in the PVN and SON increased aggression only in metestrus-diestrus GH rats (two-way ANOVA, factor treatment: $F_{(1,19)} = 3.342$, $p = 0.083$; estrous cycle: $F_{(1,19)} = 6.68$, $p = 0.018$; treatment × estrous cycle: $F_{(1,19)} = 6.45$, $p = 0.02$). **h** Confirmation of virus infection in the PVN (right) and SON (left). OXT-neurophysin I staining: green; mCherry (virus): red. Scale bars 300 µm. Data are shown as mean+SEM. $^*p < 0.05$; $^{**}p < 0.01$; $^{***}p < 0.0001$ vs either vehicle or control; $^e p < 0.05$ vs met-diestrus. OXT: GH: $n = 9$ and 12; IST: $n = 9$; AVP: $n = 18$, 9, and 9, respectively; OXTR-A: $n = 14$ and 15; combination OXT/OXTR-A/V1aR-A: $n = 8$, 6, 8, 9, and 7, respectively; chemogenetics: $n = 6$ and 15. Control group consisted of (i) rAAV1/2 OXTpr-mCherry+CNO, no virus infusion + (ii) saline, or (iii) CNO.

movie 2 and 3, and Supplementary Fig. 4). Surprisingly, the second light stimulation failed to enhance aggression displayed by GH rats, and this could be owing to a depletion of presynaptic OXT or non-selective binding of OXT to V1aRs. Future studies should focus on unraveling these hypotheses.

**AVP exerts an anti-aggressive effect within the dLS.** To localize the anti-aggressive effects of AVP identified above, we used intracerebral microdialysis and neuropharmacology within the dLS (Fig. 5a), as local V1aR binding was decreased in highly aggressive IST females (Fig. 2e).

AVP release was monitored within the dLS of GH and IST rats under basal conditions and during exposure to the FIT. Whereas in GH rats, AVP release was found to significantly increase to 130%, such a rise was not found in IST females resulting in lower AVP levels during the FIT compared with GH controls (Fig. 5b, Supplementary Data 2–3). Next, to prove that AVP acts on V1aRs specifically in the dLS to inhibit aggression, we infused AVP, TGOT (a selective OXTR agonist), or OXT either into the dLS, where we identified predominantly V1aRs, or into the vLS, where predominantly OXTRs were found (Fig. 2c and f). Only bilateral infusions of AVP, into the dLS of IST rats resulted in decreased total aggression, keep down, and offensive grooming (Fig. 5c and Supplementary Fig. 5). In contrast, none of the treatments in the vLS affected aggression (Fig. 5d and Supplementary Data 2). Surprisingly, V1aR-A administration enhanced aggression in

both GH and IST rats (Fig. 5e–f and Supplementary Fig. 5), confirming the anti-aggressive effect of endogenous AVP, independent of the housing condition.

**Spontaneous activity in neurons in the dLS and vLS is differentially modulated by activation of OXTRs and V1aRs.** The LS consists mostly of GABAergic neurons grouped into different subnuclei according to the expression of different neuropeptides and receptors, most importantly V1aRs and OXTRs[34,41]. It has been reported that spontaneous activity in these neurons is modulated by AVP in multiple ways[42]. In our behavioral paradigm, activation of OXTRs in the vLS enhanced, whereas activation of V1aRs in the dLS rather reduced aggression in female rats. To test whether such subregion- and neuropeptide-specific effects can also be observed at the level of network activity, we recorded spontaneous activity of GABAergic dLS and vLS neurons in brain slices from female juvenile (20–25 days old) Venus-VGAT rats (whole-cell voltage-clamp at −60 mV, biocytin-labeling), and investigated OXTR- and V1aR-mediated effects by applying either TGOT (1 µM) or a combination of OXTR-A (10 µM) and AVP (1 µM) to the bath. We also characterized these two neuronal populations with respect to morphological and molecular parameters (Fig. 6a).

Neurons in the dLS exhibited a higher morphological complexity compared with vLS neurons (Fig. 6a–f), as indicated by higher numbers of neurite branches and branching points

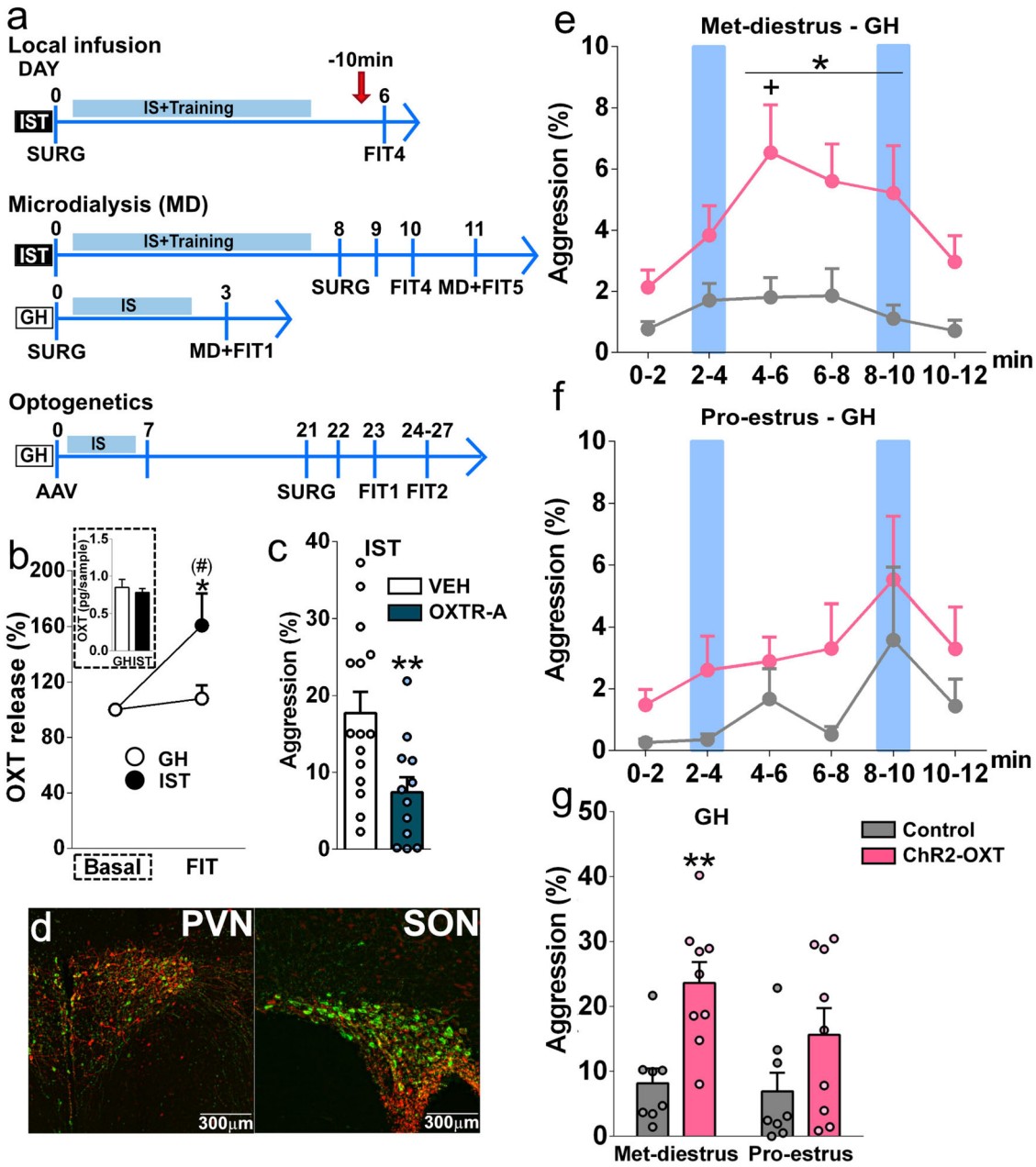

**Fig. 4 The pro-aggressive effect of OXT is mediated in the vLS. a** Scheme illustrating the experimental design for pharmacological, microdialysis, and optogenetic experiments targeting the OXT system (*AAV* adeno-associated virus microinfusion into the hypothalamic paraventricular (PVN) and supraoptic (SON) nuclei; arrow=drug infusions; *FIT* female intruder test; *GH* group-housed; *IS* social isolation; *IST* isolated and trained, *MD* microdialysis, *SURG* surgery, *OXT* oxytocin, *OXTR-A* OXT receptor antagonist, *vLS* ventral part of the lateral septum). **b** IST, but not GH females showed an increased rise in the percentage of OXT release (OXT content in microdialysates sampled during the FIT/OXT content in microdialysates sampled in the baseline × 100) in the vLS during the FIT (one sample Student's *t* test IST: $t_{(7)} = 2.65$, $p = 0.033$; GH: $t_{(7)} = 0.83$, $p = 0.43$), thus OXT release during FIT tended to be higher in IST compared with GH rats ($t_{(14)} = 2.12$, $p = 0.053$). Insert shows that absolute OXT content in microdialysates sampled under basal conditions did not differ between the groups ($t_{(14)} = 0.54$, $p = 0.60$). **c** OXTR-A (100 ng/0.5 μl) infusion into the vLS reduced total aggression (two-tailed Student's *t* test $t_{(26)} = 2.58$, $p = 0.016$) in IST females. **d**–**g** Optogenetic stimulation (indicated by blue columns) of OXT axons in the vLS of GH females during the FIT. **d** Confirmation of virus infection in OXT neurons of the PVN (left) and SON (right). OXT-neurophysin I staining: green; mCherry (virus): red. Scale bars 300 μm. Blue-light stimulation of channelrhodopsin (ChR2)-OXT fibers enhanced aggressive behavior in **e** metestrus-diestrus (Met-diestrus) (two-way ANOVA followed by Bonferroni factor: time: $F_{(5,75)} = 2.72$, $p = 0.026$; virus: $F_{(5,75)} = 20.03$, $p = 0.0004$; time × virus: $F_{(5,75)} = 1.056$, $p = 0.392$), but not in **f** proestrus-estrus (Pro-estrus) females (time: $F_{(5,70)} = 2.84$, $p = 0.02$; factor virus: $F_{(1,14)} = 2.73$, $p = 0.12$; virus × time: $F_{(5,70)} = 0.02$, $p = 0.97$). **g** Cumulative analyses show that light stimulation enhanced aggression only in ChR2-OXT females in the metestrus-diestrus phase of the cycle (factor virus: $F_{(1,15)} = 13.06$, $p = 0.0026$; estrous cycle: $F_{(1,15)} = 2.07$, $p = 0.17$; virus × estrous cycle: $F_{(1,15)} = 1.114$, $p = 0.31$). Data are shown as mean+SEM. $^{(\#)}p = 0.05$ vs GH; $^*p < 0.05$, $^{**}p < 0.01$ vs either vehicle, baseline or ChR2 control; $^+p < 0.05$ vs 0–2 time point. Microdialysis: $n = 8$; OXTR-A: $n = 13$ and 15; optogenetics: $n = 8$ and 9. Control group consisted of rAAV1/2 OXTpr-mCherry + light stimulation.

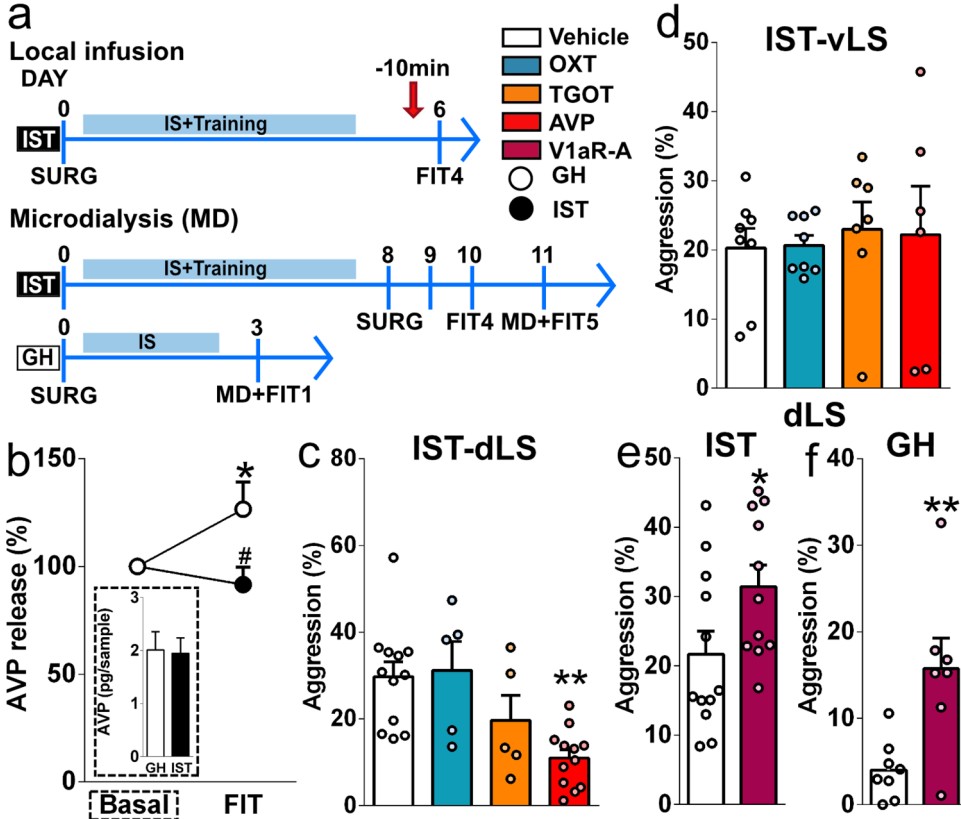

**Fig. 5 AVP exerts anti-aggressive effects within the dLS. a** Scheme illustrating the experimental design for pharmacology and microdialysis experiments targeting the AVP system (*AVP* arginine vasopressin, arrow=drug infusions, *FIT* female intruder test, *GH* group-housed, *IS* social isolation, *IST* isolated and trained; *MD* microdialysis, *SURG* surgery, *dLS* dorsal part of the lateral septum, *OXT* oxytocin, *TGOT* [Thr4,Gly]OXT, OXT receptor agonist, *vLS* ventral part of the lateral septum; *V1aR-A* V1aR receptor antagonist). **b** GH, but not IST females showed an increased rise in the percentage of AVP release (AVP content in microdialysates sampled during the FIT/AVP content in microdialysates sampled in the baseline × 100) in the dLS during the FIT (Wilcoxon signed-rank test GH: $W_{(9)} = 45$, $p = 0.0039$; IST: $W_{(7)} = -4.00$, $p = 0.81$), thus AVP release during the FIT was higher in GH than in IST rats (Mann–Whitney $U$ test = 6.00, $p = 0.0052$). Insert shows that absolute AVP content in microdialysates sampled under basal conditions did not differ between the groups. **c** Infusion of AVP, but not OXT or TGOT (all at 0.1 ng/0.5 µl), into the dLS decreased total aggression in IST females ($F_{(3,30)} = 7.292$, $p = 0.0008$). **d** AVP infusion into the vLS did not change aggression in IST females ($F_{(3,25)} = 0.11$, $p = 0.95$). Local blockade of V1aR (100 ng/0.5 µl) prior to the FIT increased aggression in **e** GH (two-tailed Student's $t$ test $t_{(13)} = 3.31$, $p = 0.006$) and **f** IST rats ($t_{(20)} = 2.14$, $p = 0.045$). Data are shown as mean +SEM. #$p < 0.05$ vs GH; *$p < 0.05$, **$p < 0.01$ vs either vehicle or baseline. Microdialysis: $n = 8$ and 9; AVP, OXT, TGOT dLS: 12, 5, 5 and 12, respectively; vLS: 8, 8,7 and 6, respectively; V1aR-A: GH: $n = 8$ and 7, IST: $n = 12$ and 11.

(Fig. 6i). Apart from the specific expression of V1aRs and OXTRs, dorsal and vLS neurons also differed regarding the expression of other markers: somatostatin-positive cell bodies were only found in the dLS (Fig. 6g), whereas estrogen receptor α (ERα) expressing cells were exclusively located in the vLS (Fig. 6h), further reinforcing that vLS and the dLS neurons are distinct populations.

Although excitatory spontaneous activity was not observed under our recording conditions, in line with previous observations[42], selective activation of OXTRs in the vLS differentially affected the spontaneous inhibitory activity in dLS versus vLS (Fig. 6j and l). In the dLS cells, the selective OXTR agonist TGOT caused an increased frequency of spontaneous IPSCs (sIPSCs). sIPSCs were entirely blocked by further addition of bicuculline (50 µM, selective GABA-A antagonist) (Fig. 6j–k), which confirmed the exclusive GABAergic origin of these currents. In agreement with those results, blockade of OXTRs by selective OXTR-A in the dLS decreased sIPSC frequency (Supplementary Fig. 6). Conversely, in the vLS neurons responded to TGOT with a decreased sIPSC frequency (Fig. 6l–m) indicating a reduced inhibitory tone following OXTR activation. Again, sIPSCs were entirely blocked by bicuculline.

V1aR activation increased sIPSC frequency specifically in the dLS neurons, whereas no consistent effect was seen in vLS cells (Supplementary Fig. 6).

Altogether, our data demonstrate that activation of OXTRs in the vLS concomitantly weakens the tonic inhibition of vLS GABAergic neurons and strengthens tonic inhibition of dLS GABAergic neurons. We also could confirm subregion-specific effects of AVP, as only dLS neurons were affected by V1aR activation, in line with behavioral experiments where AVP effects were also observed only in the dLS. In addition, dLS neurons showed extended dendritic arborizations and other cell markers than vLS neurons.

**An intrinsic GABAergic circuit within the LS regulates aggression in female Wistar rats.** After we have shown that (i) activation of vLS OXTR increases, whereas dLS V1aR decreases aggression, and (ii) spontaneous inhibition is increased in dorsal and decreased in ventral cells by OXTR activation, we hypothesized that those subregions would also be differentially activated after an aggressive encounter. Therefore, we compared the neuronal activity within the vLS and dLS of Venus-VGAT GH and

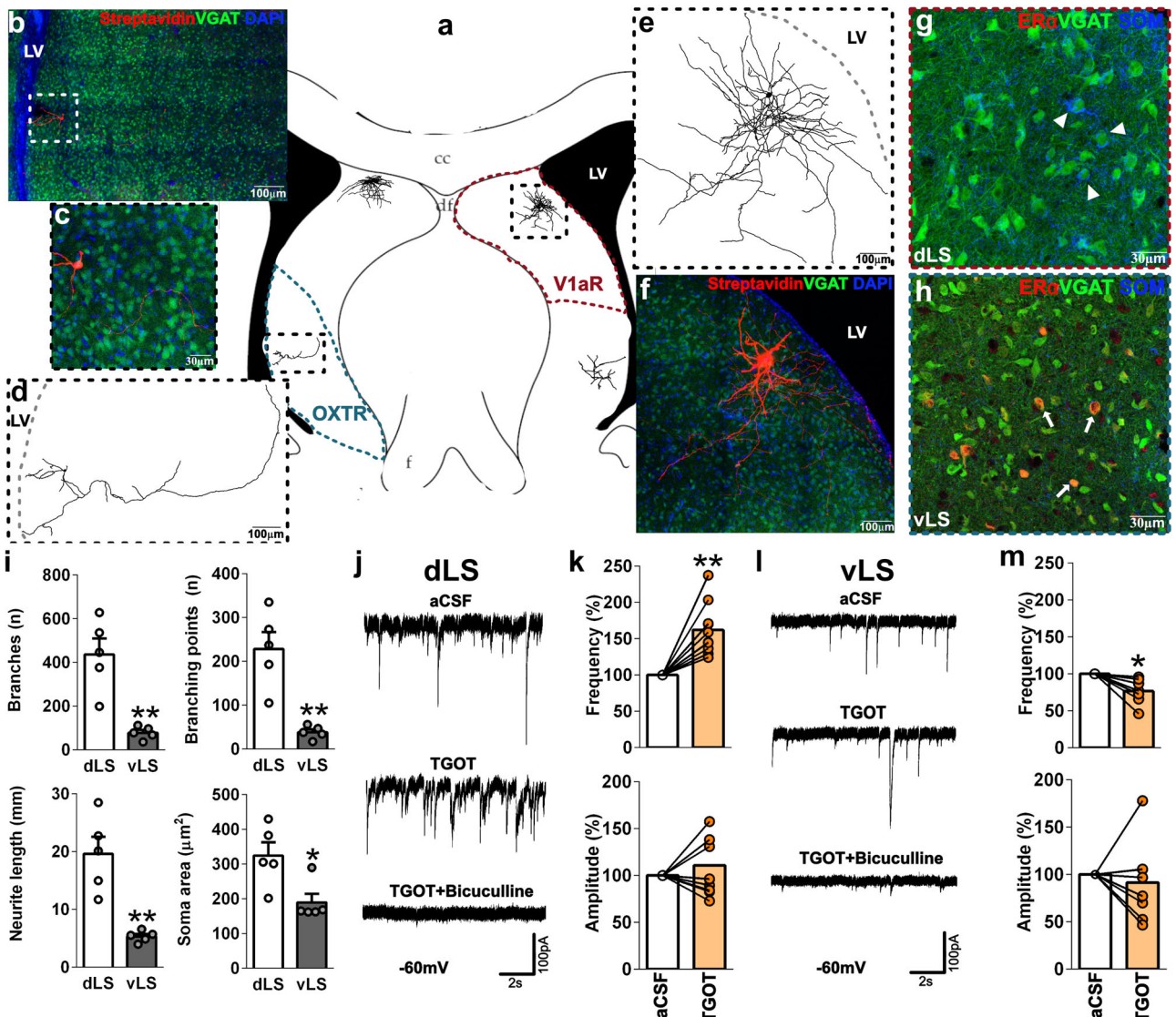

**Fig. 6 Spontaneous activity in neurons of the dLS and vLS is differentially modulated by activation of OXTRs. a** Scheme indicating the receptor binding-specific delimitation of the vLS (OXTR) and dLS (V1aR) used during voltage-clamp experiments, including two representative dLS and vLS cells. **b** Maximal z-projection of biocytin-filled vLS cell (streptavidin-CF633) overlaid with a single z-plane of VGAT and DAPI. **c** Magnification of cell body from the cell shown in **b**. **d** Neurite reconstruction of cell shown in **b**, and **e** of the cell shown in **f**; the gray-dotted line indicates the border of the lateral ventricle (LV). **f** Maximal z-projection of a biocytin-filled dLS (streptavidin-CF633) cell overlaid with a single z-plane of VGAT and DAPI. **g** Magnification of a single z-plane indicating the presence of somatostatin (SOM) cell bodies (arrowheads) only in the dLS. **h** Magnification of a single z-plane indicating the presence of ERα-positive cell bodies (arrows) only in the vLS. **i** Morphological characterization of LS neurons. dLS cells exhibited more neurite branches (two-tailed Student's $t$ test $t_{(4)} = 4.84$, $p = 0.0072$) and branching points ($t_{(4)} = 4.85$, $p = 0.0072$) longer neurite length ($t_{(4)} = 4.80$, $p = 0.0078$), and wider soma areas (Mann–Whitney $U$ test $U = 1.0$, $p = 0.016$) than vLS cells. Representative spontaneous current traces during TGOT (1 μM) and bicuculline (50 μM) bath application in dLS **j** and vLS **l** cells. **k** TGOT increased sIPSC frequencies in dLS cells (Wilcoxon signed-rank test $W_{(9)} = 55$, $p = 0.002$) and **m** decreased sIPSC frequencies in vLS cells ($W_{(7)} = -36$, $p = 0.0078$). TGOT had no effect whatsoever on the sIPSC amplitude independent of the subregion. Data are shown as mean+SEM. $^{*}p < 0.05$, $^{**}p < 0.01$, vs either dLS or aCSF. Morphology: $n = 5$; electrophysiology: dLS $n = 10$, vLS $n = 8$.

IST rats after exposure to the FIT using pERK as a neuronal activity marker[12,14,15].

We found striking regional differences in neuronal activity in the LS of highly aggressive IST females compared with low-aggressive GH rats (Fig. 7a–d). In detail, in the dLS FIT exposure and the display of high aggression by IST rats resulted in a decreased total number of both pERK-positive and VGAT/pERK-positive cells (Fig. 7a–b and e), with a negative correlation found between the amount of double-labeled cells in the dLS and aggression (Fig. 7f). In contrast, in the vLS of IST females, we found a trend towards more pERK-positive cells and an increased number of VGAT/pERK-positive cells (Fig. 7c–d and g), which

did not correlate with aggression (Fig. 7h and Supplementary Data 2–3).

To further confirm the dLS as an anti-aggressive center versus the vLS as a pro-aggressive center, and consequently to create a causal link between neuronal activity and behavior, we infused rats with a selective GABA-A agonist muscimol 10 min before the FIT. As predicted, inactivation of the dLS in GH rats enhanced aggression, threat behavior and the percentage of rats showing attacks (Fig. 7i–j and Supplementary Data 2–3). In contrast, opposing effects were seen in the vLS of IST rats, where muscimol decreased aggression and the percentage of rats showing attacks. (Fig. 7k–l and Supplementary Data 2–3).

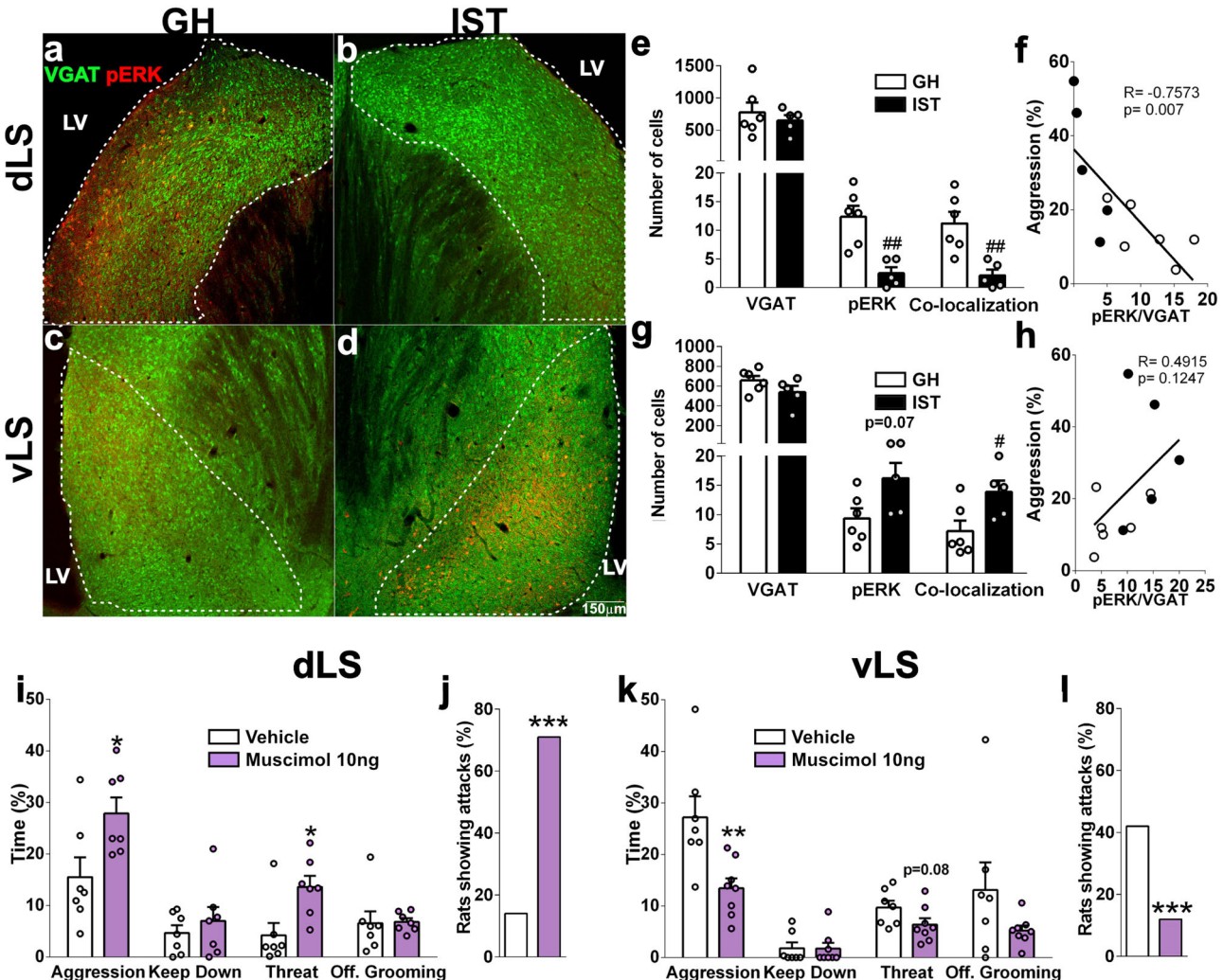

**Fig. 7 An intrinsic GABAergic circuit within the subregions of the LS regulates aggression in female Wistar rats. a–d** Example average z-projects showing pERK (Alexa594) immunostaining in Venus-VGAT females after exposure to the female intruder test (FIT). **a** dorsal LS (dLS) of a group-housed (GH), i.e., low-aggressive female, **b** dLS of an isolated and trained (IST), i.e., high-aggressive female, **c** ventral LS (vLS) of a GH female, **d** vLS of an IST female (*LV* lateral ventricle, *VGAT* vesicular GABA transporter). **e–h** Neuronal activity in the LS reflected by pERK staining after FIT exposure: **e** In the dLS of IST females less pERK-positive cells (two-tailed Student's *t* test $t_{(9)} = 4.20$, $p = 0.0023$) and fewer pERK/VGAT co-localized cells were found ($t_{(9)} = 3.75$, $p = 0.004$). **f** Aggression negatively correlated with the number of pERK/VGAT-positive cells in the dLS (Pearson's correlation $r = -0.746$, $p = 0.008$). **g** In the vLS of IST females a tendency of more pERK-positive cells ($t_{(9)} = 2.07$, $p = 0.068$) and a higher number of pERK/VGAT-positive cells was found ($T_{(9)} = 2.28$, $p = 0.049$). **h** Aggression did not correlate with the number of pERK/VGAT-positive cells in the vLS ($r = 0.4915$, $p = 0.1247$). **i** Infusion of muscimol (10 ng/0.5 μl) into the dLS increased total aggression ($t_{(12)} = 2.52$, $p = 0.027$) and threat (Mann–Whitney *U* test $U = 5.00$, $p = 0.011$) in GH rats. **j** Inhibition of the dLS also enhanced the percentage of rats showing attacks (Fisher exact test, $p < 0.0001$). **k** Muscimol in the vLS decreased total aggression ($t_{(13)} = 3.191$, $p = 0.0071$) and tended to decrease threat behavior ($t_{(13)} = 1.832$, $p = 0.090$). **l** Inhibition of the vLS also reduced the percentage of rats showing attacks ($p < 0.0001$). Data are shown as mean+SEM. #$p < 0.05$, ##$p < 0.01$ vs GH; *$p < 0.05$, **$p < 0.01$, ***$p < 0.0001$ vs vehicle. Neural activity: GH $n = 6$ and IST $n = 5$; muscimol: dLS: $n = 7$ vLS: $n = 8$ and 7.

## Discussion

Aiming to study neural mechanisms of aggression in virgin female rats, we have established a behavioral paradigm combining social isolation and aggression-training by repeated exposure to the FIT, which reliably exacerbated the mild levels of aggression typically displayed by GH female Wistar rats. In addition, we have shown that female aggression is controlled by a fine-tuned balance between OXT, AVP, and GABA neurotransmission within two distinct neuronal populations of the LS, i.e., the dLS and vLS, with contrasting subregional effects of OXT and AVP (Fig. 8). In detail, the display of high aggression by IST females during the FIT was accompanied by elevated OXT release within the vLS, which was also reflected by higher OXT levels in their CSF. This indicates the involvement of septal OXT in the high

aggression displayed by IST rats, which we further confirmed by complementary pharmacological, chemogenetic and optogenetic approaches: (i) Pharmacological blockade of OXTR either i.c.v. or locally in the vLS attenuated aggression in IST rats, whereas (ii) chemogenetic and optogenetic stimulation of OXT neurons in the PVN and SON, and of OXT axons projecting to the vLS, respectively, heightened the aggression of low-aggressive and non-receptive GH females.

Concerning the involvement of the brain AVP system in female aggression, we revealed a completely different picture. Highly aggressive IST females showed unchanged AVP release within the dLS during an aggressive encounter, which was also reflected by lower AVP content in the CSF. As further proof of an inhibitory effect of AVP on female aggression, infusion of AVP

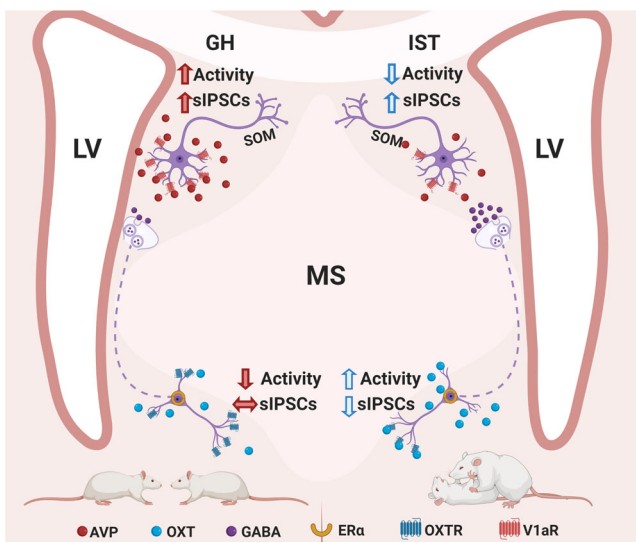

**Fig. 8 The balance between OXT and AVP regulates the inhibitory tonus within the LS in order to control aggression in female rats.** The scheme depicts the main findings of this manuscript. Furthermore, it hypothesizes that subtle alterations in OXT and AVP release within the LS may impact GABAergic neurotransmission to generate aggressive behavior in female rats. On the left side is depicted, how the brain of a GH female responds to an intruder: a combination of low OXT release ventrally and high AVP release dorsally evokes an increased activity of the dLS, thereby reducing aggression. On the right side is depicted, how the brain of an IST female responds to an intruder: a combination of high OXT release ventrally and low AVP release dorsally culminates in increased tonic inhibition of dLS (dotted line indicates hypothetical pathway) and increased activation of the vLS, promoting aggression. Dorsal and ventral neurons are shown in proportion to their real size. *AVP* arginine vasopressin, *ERα* estrogen receptor α, *IPSC* inhibitory post-synaptic current, *LV* lateral ventricle, *MS* medial septum, *OXT* oxytocin, *OXTR* oxytocin receptor, *SOM* somatostatin, *V1aR* V1a receptor. Drawn using https://biorender.com.

either i.c.v. or directly into the dLS reduced the aggression seen in IST rats, whereas blockade of V1aR in the dLS exacerbated aggression in both GH and IST rats.

OXT and AVP have been repeatedly described as important neuromodulators of social behaviors in rodents, acting either synergistically or antagonistically depending on the social context and sex[8,9,22,24,36,43]. In males, reduced aggression has been reported in rats after i.c.v. infusion with synthetic OXT[23] and in mice after optogenetic stimulation of hypothalamic OXT neurons[43]. In contrast, the few studies on OXT-effects on female aggression rather revealed pro-aggressive or antisocial effects, for example, in female rhesus monkeys[44], non-lactating women[8,45], and in lactating female rats[18,33]. Although the factors motivating the display of aggression and its severity likely differ in lactating versus virgin females[5,33,46,47], we should highlight that, from an evolutionary point of view, co-opting the same neuropeptidergic systems for promoting aggression in lactating females to protect the offspring, and in virgin females to protect their territory or to get access to resources, makes sense, as high activity of the brain OXT system reflected by elevated neuropeptide synthesis, release and receptor binding[18,24,33,36] is known to underlie maternal aggression[24,33,36,48]. Further evidence for a sex-specific effect of OXT on social behavior comes from studies on social motivation, where OXT is essential for naturally occurring social preference behavior in male rats and mice[49], but not in virgin female rats[50]. Thus, our results on a pro-aggressive effect of OXT in virgin females add an important piece of evidence to the sex-specific effects of OXT on social behaviors.

In our experiments, aggression in females was found to be dependent upon the estrous cycle. Indeed, a previous study has shown low aggression in estrus rats[51], as we have seen in the IS proestrus/estrus females. In addition, chemo- and optogenetic stimulation of the OXT system either centrally or in the vLS elevated aggression levels exclusively in metestrus/diestrus GH rats. In this context, it is of note that OXTR expression and binding in regions involved in the regulation of aggression, such as the VMH, bed nucleus of stria terminalis (BNST), and LS, undergo dynamic changes during the estrous cycle in a sex steroid-dependent manner[36,52]. Furthermore, sex steroids affect OXT expression and release as shown in vitro[53] and in vivo[54]. Although one could hypothesize that activation of local ERα in the vLS is counteracting the effects of OXT in females, the extent of how the estrous cycle is involved in the modulation of the vLS neurons and, consequently, on local OXT and AVP actions on female aggression remains to be studied.

Furthermore, we found a mismatch between OXT release and OXTR binding in the LS, i.e., high local OXT release was found in rats with reduced OXTR binding. Although this might seem confusing at first sight, similar mismatches between release and receptor binding patterns have been described before in association with increased OXT availability in male mice after fear-conditioning[36].

Although several pieces of evidence support pro-aggressive effects of AVP in males[9,22,55,56], conflicting data supporting an anti-aggressive effect of AVP have been described in animal models of high and abnormal aggression. For example, male rats bred for low anxiety and characterized by excessive and abnormal aggression[31] show low levels of AVP release in the LS during the resident-intruder test[21]. Also, aggressive male mice with a short attack latency exhibit decreased AVP innervation of the LS[57], thereby demonstrating not only an association between abnormal and high aggression, but furthermore a blunted activity of the septal AVP system. In agreement, high aggression displayed by dominant males has also been linked to reduced septal AVP signaling, i.e., alpha male mice showed decreased V1aR binding in the LS, when compared with subordinate males[32], and synthetic AVP was able to flatten dominant behavior in rhesus macaques[58]. In lactating females, the link between AVP and aggression seems to be complex as well. Whereas synthetic or endogenous AVP has been shown to decrease maternal aggression in Sprague-Dawley rats[19,20], contrasting and probably anxiety-dependent effects have been described in the central amygdala of high anxiety Wistar rats, where AVP release was associated with maternal aggression[17]. Accordingly to our data, serenic effects of AVP have also been shown in non-lactating hamsters[9,59] and rhesus macaques[44]. Interestingly, recent studies in humans have highlighted the pro-social role of AVP as a potential treatment for social dysfunctions, since intranasal AVP was able to increase risky cooperative behavior in men[60] and to enhance social skills in autistic children[61].

After identification of the specific pro-aggressive effect of OXT in the vLS, and anti-aggressive effect of AVP in the dLS, of virgin female Wistar rats (Fig. 8), in the final set of experiments we were able to link these neuropeptidergic actions with specific regional effects in neuronal activity after an aggressive encounter. In detail, elevated pERK expression reflecting high neuronal activity was found in the vLS of highly aggressive IST females, whereas pERK expression was elevated in the dLS of low-aggressive GH rats. These findings were functionally validated by manipulation of local GABAergic neurotransmission using local muscimol infusions, demonstrating that dLS neurons inhibit, whereas vLS neurons seem to promote aggression in females. Subnetwork-dependent responses were also seen after OXTR and V1aR activation in vitro. TGOT decreased the inhibitory spontaneous

activity ventrally, whereas it increased it dorsally. Thus activation of OXTRs located exclusively in the vLS leads to enhanced GABAergic inhibition of GABAergic neurons in the dLS, possibly facilitating aggression. In addition, AVP increased sIPSCs frequency dorsally, but did not affect sIPSCs ventrally. This observation refines the results from an earlier study[42] showing that V1aR activation indirectly enhances the inhibitory neurotransmission of a majority of LS neurons, notwithstanding the dorso-ventral localization of the recorded cells. Thus, according to our results, both anti-aggressive and synaptic effects of AVP appear to be mediated exclusively within the dLS.

The role of the LS in suppressing aggression has been described for decades[10,25–30]. However, only one recent paper has shown the existence of local microcircuits within the LS regulating aggression in male mice[56]. Supporting the pro-aggressive role of AVP in males[9,22,24,55], V1b receptor activation on presynaptic terminals of hippocampal fibers to the LS increased aggression via stimulation of inhibitory interneurons in the dLS projecting to the vLS. Although this data contrasts with ours, we have to keep in mind that the regulation of aggression within the LS, via OXTRs and V1aRs, might be underlined by sex- and species-dependent mechanisms, as both males and females[34], as well as mice and rats[32,34], differ in their receptor binding. In addition, we investigated the role of V1aRs, but not V1bRs and differential effects on aggression are likely, also the involvement of endogenous AVP acting on V1bRs has not been shown in the previous study[56]. Based on our finding that V1aR activation increases sIPSCs frequency exclusively in the dLS, we could hypothesize that V1aR activation might overrule V1bR effects by inhibiting V1bR-responsive interneurons, thereby decreasing aggression. Accordingly, in our model of female aggression, activation of V1aRs in the dLS seems to overshadow any possible pro-aggressive V1bR-mediated action, since AVP administration in the dLS was able to mimic the anti-aggressive effects of i.c.v. AVP in IST females. Indeed, AVP has been shown to present a higher affinity to V1aRs over V1bRs in vitro[37]. Nevertheless, future studies should address how the interplay between V1aR and V1bR activation within the dLS affects aggression in females.

Supporting our findings, a neurocircuit involving contrasting effects of OXTR and V1aR activation has been described in the context of fear in the central amygdala, wherein activation of putative OXTR-positive cells led to inhibition of V1aR-positive cells via GABAergic transmission, which abolished fear-related behavioral effects of V1aR activation[40,62]. To the best of our knowledge, such a mechanism has never been described before in the context of aggression. In further support of our finding that high levels of OXT in the vLS leads to increased tonic inhibition of the dLS via GABAergic projections, neural projections from the vLS towards the dLS have also been described in male Wistar rats[63]. In addition, blockade of GABA-A receptors in the dLS of lactating mice reduced maternal aggression[30]. Thus, aggression in animals with a higher OXT system activity[33,36] is at least partially modulated by increased GABA neurotransmission in the dLS. Although our data depict compelling evidence that OXT release in the vLS facilitates aggression via increasing tonic inhibition onto dLS neurons, whether those phenomena are functionally associated remains to be determined in further studies.

Taken together, our results shed light on the neurobiological mechanisms underlying aggression in females. We have shown that the level of aggression expressed by virgin female Wistar rats is determined by a fine-tuned balance between OXT- and AVP-mediated neuromodulation in septal sub-networks. Disturbances in this balance, such as a shift towards a more dominant OXT or AVP neurotransmission, lead to distinct aggression responses. Dorsal V1aR-expressing neurons of the LS seem to be pivotal for the AVP-induced inhibition of female aggression, whereas ventral

neurons in the OXTR-expressing LS appear to be the main generators of OXT-induced aggression. In addition, we propose a hypothetical mechanism by which activation of OXTRs reduces tonic inhibition of vLS neurons, which in turn may enhance their GABA transmission onto the dorsal cells and thus ultimately may reduce septal gating on aggression (Fig. 8).

## Methods

**Ratlines and animal care.** Adult virgin female Wistar rats (10–14 weeks old) bred in the animal facilities of the University of Regensburg (Germany) were used for behavioral experiments. Intruders, female Wistar rats were obtained from Charles Rivers Laboratories (Sulzfeld, Germany) and kept in groups of 3–5 animals in a separate animal room. All rats were kept under controlled laboratory conditions (12:12 h light/dark cycle; lights off at 11:00, 21 ± 1 °C, 60 ± 5% humidity) with access to standard rat nutrition (RM/H, Sniff Spezialdiäten GmbH, Soest, Germany) and water ad libitum. For patch-clamp (p20–25) and pERK immunohistochemistry analyses (10–14 weeks), female Venus-VGAT rats (*VGAT*, vesicular GABA transporter; lineVenus-B, W-Tg(Slc32a1-YFP*)1Yyan, Wistar background)[64] that were bred in the animal facilities of the University of Regensburg were used. All procedures were conducted following the Guidelines for the Care and Use of Laboratory Animals of the Local Government of Oberpfalz and Unterfranken.

**Female intruder test.** The FIT was performed in the early dark phase (between 12:00 and 16:00) under dim red light conditions. An unfamiliar female intruder weighing between 10 and 20% less than the resident[12] was released into the home cage of the resident for 10 min. The test was videotaped for later analysis by an experienced observer blind to treatment using JWatcher event recorder Program[65]. The percentage of time the resident spent with four main behavioral aspects was scored: (i) aggressive behaviors, i.e., keep down (standing over the intruder and keeping it against the substrate, piloerection is often present), threat behavior (moving towards the opponent, chasing or pushing it away, piloerection is often present), offensive grooming (nibbling, pulling, and biting the fur on the neck and back of the intruder), and attacks (attack bites on the neck, back or flanks of the intruder, predicted by clinch behavior); (ii) neutral behaviors, i.e., exploring and investigating the home cage, drinking, eating and self-grooming; (iii) social behaviors, i.e., non-aggressive social interactions, sniffing, following; and (iv) defensive behavior, i.e., submissive posture, kicking a pursuing intruder with hind limb. We also monitored sexual behavior (lordosis, hopping, darting and mounting) and immobility. In addition, we scored the frequency of attacks as well as the latency to the first attack. Residents had vaginal smears taken after the FIT to verify the estrous cycle; all phases of the estrous cycle were included in the study (for detailed behavioral analyses of all experiments please see Supplementary Data 2–3 and 6). The intruder's cycle was not tracked during the study.

### Overview of the in vivo experiments

*Animal groups for OXT/AVP measurements in CSF, plasma, and receptor autoradiographic analyses (Fig. 1a).* Female Wistar rats were split into three different housing conditions: GH females were kept in groups of three to five animals per cage, IS females were singly housed for 8 days, and IST females were also kept singly for 8 days, but from day 5 on, they underwent three consecutive FITs for aggression-training. On day 9, GH females were singly housed 4 h prior to the behavioral experiments. One hour after lights went off, GH, IS, and IST rats were either exposed to the FIT, whereas control rats were left undisturbed in their home cage (behavioral data displayed in Fig. 1). Immediately after the FIT, rats were deeply anesthetized using intraperitoneal (i.p.) urethane (25%, 1.2 ml/kg) to allow CSF collection via puncture of the Cisterna cerebromedullaris. After decapitation, brain and trunk blood were collected for receptor binding and hormonal measurements, respectively (data displayed in Fig. 2). For radioimmunoassays, animals were again split into two groups according to the peptide measured, i.e., half of the subjects were used for OXT radioimmunoassay (RIA), whereas the other half was used for AVP RIA. Here we again counterbalanced levels of aggression to avoid uneven groups. In addition, samples coming from rats that either we could not extract or had shown blood in their CSF were not included in the assays.

*Neuropharmacology design.* For pharmacological experiments, female rats were split into GH and IST conditions. All females underwent surgery for i.c.v. or local cannula implantation. IST females were left undisturbed for recovery for 3 (i.c.v.) or 5 (local) days. The aggression training was performed as described above, except for the fact that residents received a sham-infusion before the FIT to get used either to the i.c.v. or local infusion procedure. GH animals were kept single-housed overnight for recovery and brought back to their original groups the next day until the start of the experiments when they were transferred into an observation cage and single-housed for 4 h before the FIT. All animals were handled daily to get used to the infusion procedure. Typically, a cross-over, within-subjects design was used for all i.c.v. agonist experiments, whereas a between-subjects design was used for the local infusions and i.c.v. antagonists experiments. Additionally, to guarantee that all groups had similar average levels of aggression before pharmacological

manipulations, we allocated IST rats into the treatment groups according to their average levels of aggression expressed in the third session of training.

*Microdialysis.* OXT and AVP release within the LS was monitored in GH and IST females before and during FIT exposure. After 4 days of training, IST rats had their microdialysis probes implanted into the LS. After 1 day of recovery, IST subjects received the 5th training FIT to confirm their previous levels of aggression. On the following day, both GH and IST animals underwent the microdialysis procedure. In brief, rats were connected to a syringe mounted onto a microinfusion pump via polyethylene tubing and were perfused with sterile Ringer's solution (3.3 μl/min, pH 7.4) for 2 h before sampling of microdialysates to establish an equilibrium between inside and outside of the microdialysis membrane. One hour after lights went off, three consecutive 30-min dialysates were collected (baseline samples 1 and 2 represented in Figs. 4 and 5 as an average of both time-points) and during an ongoing FIT (sample 3). Dialysates were collected into Eppendorf tubes containing 10 μl of 0.1 M HCl, were immediately frozen on dry ice, and stored at −20 °C until subsequent quantification of AVP and OXT by radioimmunoassay (for behavioral analyses see Supplementary Data 2–3). Microdialysis data are displayed here as a percentage of release from baseline (Peptide content of sample 3/average Peptide content in sample1 + sample 2 × 100).

*Chemo- and optogenetics design.* Chemo- and optogenetic experiments targeting the OXT system were only performed on GH rats. After stereotaxic virus infusion into the PVN and SON, animals were kept single-housed for 1 week to recover. Thereafter, they were again group-housed for 2 weeks until either the experiment took place (chemogenetic) or they had their optical fiber implanted (optogenetic). Similarly to the pharmacologal experiments, chemogenetic rats were kept in groups and isolated shortly before the dark phase (see above). Subjects received an i.p. infusion of clozapine-*N*-oxide dihydrochloride (CNO, 2 mg/kg) 45 min before the FIT. They were brought back to their original groups directly after the FIT. Importantly, for those experiments, we had three control groups: (i) subjects infected with a control rAAV1/2 OXTpr-mCherry into the PVN and SON, which received CNO (virus control), non-infected rats who received either (ii) CNO or (iii) saline infusions (drug control). Since there was no difference among the levels of aggression displayed by those three groups (not shown) they were pooled together in a single control group depicted in gray in Fig. 3g. In the optogenetic experiments, after optical fiber implantation rats were single-housed for three days for recovery and to avoid damaging the fiber. Similarly to the microdialysis experiments, both controls and ChR2 animals were connected to the optogenetic cables two hours before the experiment to get used to the cables. In this experiment, the FIT lasted 12 min. Blue-light stimulation (30 ms pulses of 30 Hz delivered for 2 min; in analogy to Knobloch et al., 2012) was delivered at the 2nd and again 8th minute after the beginning of the FIT. Here, controls consisted of animals infected with a control rAAV1/2 OXTpr-mCherry into the PVN and SON. For both, chemo- and optogenetic experiments, animals were tested twice at different phases of their estrous cycle: once in proestrus/estrus, and once in metestrus/diestrus. All animals were transcardially perfused with paraformaldehyde (PFA 4%) for histological verification of the specificity of viral infections after the last test.

Although the i.c.v. and local pharmacology experiments showed robust serenic effects of AVP (Figs. 3 and 5), we decided to not genetically manipulate the AVP system centrally or locally due to the following reasons: (i) AVP is one of the main players in brain physiology and stimulation of several cell bodies in different brain regions especially in the hypothalamus could disturb homeostasis. In fact, high doses of i.c.v. AVP are known to elicit barrel rotations in rats[66]. (ii) AVP neurons are widespread in several nuclei in the rat brain such as the medial amygdala, the BNST, PVN, and SON[67], which makes the infusion of all targeted neurons challenging in terms of animal welfare. (iii) We (data not shown) and others[68] have found low specificity of the AVP directed virus to infect intra- and extrahypothalamic parvocellular AVP neurons, which are known to project to the LS[67].

*Neuronal activation after aggression.* To compare neuronal activity patterns in the dLS and vLS of IST and GH rats after exposure to the FIT, we used female Venus-VGAT rats (10–14 weeks old). Immediately after FIT exposure rats were deeply anesthetized with isoflurane (ForeneH, Abbott GmbH & Co. KG, Wiesbaden, Germany), followed by $CO_2$ transcardially perfused, and brains were harvested for subsequent immunohistochemistry stainings (behavioral data in Supplementary Data 2–3).

**Delimitation of the dLS and vLS.** The subregions of the LS were delimitated using the Paxinos rat atlas together with the receptor binding pattern of OXTRs and V1aRs as a reference.

**Stereotaxic surgery.** Rats were anesthetized with isoflurane, injected i.p. with the analgesic Buprenovet (0.05 mg/kg Buprenorphine, Bayer, Germany) and the antibiotic Baytril (10 mg/kg Enrofloxacin, Baytril, Bayer, Germany), and fixed in a stereotaxic frame. I.c.v. guide cannulas (21 G, 12 mm; Injecta GmbH, Germany) and microdialysis probes (self-made, molecular cutoff 18 kDa[22]) were implanted unilaterally, whereas local guide cannulas (25 G, 12 mm; Injecta GmbH, Germany)

were implanted bilaterally. All cannulas were implanted 2 mm above the target region to avoid lesion of the target region, whereas microdialysis probes had to be implanted directly into the target region, fixed to the skull with two jeweler's screws and dental cement (Kallocryl, Speiko-Dr. Speier GmbH, Muenster, Germany). The cannulas were closed by a stainless steel stylet (i.c.v. 25 G, local 27 G).

For viral infection in chemo- and optogenetic experiments, a solution of ketamine (100 mg/kg, Medistar, Germany) and xylazine (10 mg/kg, Medistar, Germany) was applied i.p. as anesthesia. After virus delivery, the skin was sutured. For optogenetic experiments, rats were implanted with the optic fiber (PlexBright optogenetic stimulation system fiber stub implant; 6 mm length) into the vLS, fixed to the skull with two jeweler's screws and dental cement as described above 21 days after viral infection and 2 days prior to the experiment. For specific coordinates please see Supplementary Table 1.

**Receptor autoradiography.** Brains were cryo-cut into 16-μm coronal sections, slide-mounted, and stored at −20 °C. The receptor autoradiography procedure was performed using a linear V1aR antagonist [125I]-d(CH2)5(Tyr[Me])-AVP (Perkin Elmer, USA) or a linear OXTR antagonist [125I]-d(CH2)5[Tyr(Me)2-Tyr-Nh2]9-OVT (Perkin Elmer, USA) as tracers. In brief, the slides were thawed and dried at room temperature followed by a short fixation in paraformaldehyde (0.1%). Then slides were washed two times in 50 mM Tris (pH 7.4), exposed to tracer buffer (50 pM tracer, 50 mM Tris, 10 mM MgCl2, 0.01% bovine serum albumin) for 60 min, and washed four times in Tris buffer 10 mM MgCl2. The slides were then shortly dipped in pure water and dried at room temperature overnight. On the following day, slides were exposed to Biomax MR films for 7–25 days depending on the receptor density and brain region (Kodak, Cedex, France). The films were scanned using an EPSON Perfection V800 Scanner (Epson, Germany). The optical density of V1aR and OXTR binding was measured using ImageJ (V1.37i, National Institute of Health, http://rsb.info.nih.gov/ij/). Receptor density was calculated by sampling the whole region of interest, average gray density was calculated after subtracting the tissue background data was posteriorly converted to relative optical density as previously described[69]. Bilateral measurements of 6–12 brain sections per region of interest were analyzed for each animal, thus data points represent the mean of those measurements.

**ELISA for plasma corticosterone.** Quantification of plasma corticosterone was performed using enzyme-linked immunosorbent assay (ELISA). As described before, trunk blood of FIT animals was collected after decapitation. Approximately 1 ml blood was collected in EDTA-coated tubes on ice (Sarstedt, Numbrecht, Germany), centrifuged at 4 °C (2000 × *g*, 10 min), aliquoted and stored at −20 °C until the assay was performed using a commercially available ELISA kit for corticosterone (IBL International, Hamburg, Germany) following the manufacturer's protocol.

**Radioimmunoassay for OXT and AVP.** OXT and AVP content in extracted blood and CSF, and lyophilized microdialysates was measured using a sensitive and specific radioimmunoassay (RIAgnosis, Germany; sensitivity: 0.3 pg/sample cross-reacitivity: <0.7%). All samples were measured within the same assay to avoid inter-assay variability.

**Drugs and viruses.** Animals were treated either with endogenous ligands, agonists, or antagonists to modulate the OXT and AVP systems. Usually, drugs were infused 10 min before the FIT in i.c.v. experiments (OXT: 50 ng/5 μl AVP: 0.1 ng or 1 ng/5 μl, Tocris, Germany). Antagonists were infused 10 min prior to the infusion of the respective agonist with a final volume (agonist + antagonist) of 5 μl (OXTR-A, des-Gly-NH2,d(CH2)5[Tyr(Me)2,Thr4]OVT; V1aR-A, d(CH2)5Tyr(Me)2AVP: 750 ng)[37]. For cannulas placed in the LS, agonists (OXT, AVP, TGOT, [Thr4,Gly7]OT: 0.1 ng/0.5 μl per side) were infused 5 min before the FIT, whereas antagonists (OXTR-A and V1aR-A: 100 ng/0.5 μl, per side) and muscimol (10 ng/0.5 μl, per side, Tocris) were infused 10 min before the FIT. In order to modulate the activity of the OXT neurons, the PVN and SON of GH females were infused with 280 nl of rAAV1/2 OTprhM3Dq:mCherry ($4 \times 10^{11}$ genomic copies per ml) and rAAV1/2 OXTpr-ChR2:mCherry ($4 \times 10^{11}$ genomic copies per ml) for chemo- and optogenetic experiments, respectively. Viruses were slowly infused by manual pressure infusion at 70 nl/min. After infusion, injected optogenetic animals had their water replaced by salt loaded water for 1 week (2% NaCl) to enhance virus infection and expression. Chemogenetic animals received i.p. infusions either with CNO (HB6149, HelloBio, United Kingdom) or 0.9% saline (1 ml/kg).

**Patch-clamp**

*Slice preparation.* Juvenile VGAT rats (postnatal days 15–21) were deeply anesthetized with isoflurane and decapitated. Coronal brain slices containing the LS (300 μm) were cut in ice-cold carbogenized ($O_2$ [95 %], $CO_2$ [5 %]) artificial extracellular fluid (aCSF; [mM]: 125 NaCl, 26 NaHCO3, 1.25 NaH2PO4, 20 Glucose, 2.5 KCl, 1 MgCl2, and 2 CaCl2) using a vibratome (Vibracut, Leica Biosystems, Germany) followed by incubation in carbogenized aCSF for 30 min at 36 °C and then kept at room temperature (~21 °C) until experimentation.

*Electrophysiology*. Neurons in the dLS and vLS were visualized by infrared gradient-contrast illumination via an IR filter (Hoya, Tokyo, Japan) and patched with pipettes sized 8–10 MΩ. Somatic whole-cell patch-clamp recordings were performed with an EPC-10 (HEKA, Lambrecht, Germany). Series resistances measured 10–30 MΩ. The intracellular solution contained [mM]: 110 CsCl, 10 HEPES, 4 $MgCl_2$, 10 TEA, 10 QX-314, 2.5 $Na_2ATP$, 0.4 NaGTP, 10 NaPhosphocreatine, 2 ascorbate, at pH 7.2. In addition, biocytin (5 mg/ml) was added to the intracellular solution for post hoc fluorescent labeling of the patched neurons. The bath aCSF (see above) in the experimental chamber was gassed with carbogen and kept at room temperature. The average resting potential of lateral septal neurons was −60 omV[42]. Leaky cells with a holding current beyond <−30 pA were not further used for experimentation. Spontaneous activity (i.e., sIPSCs) was recorded in voltage-clamp mode at resting membrane potential (−60 mV). The frequency and amplitude of sIPSCs were analyzed with Origin 2019 (OriginLab Corporation, Northampton, MA, USA).

*Histology*. After the experiment in vitro slices were post-fixed in 4% paraformaldehyde in PBS (4 °C, overnight) and prepared for fluorescent labeling.

**Perfusion**. After deep anesthesia with isoflurane followed by $CO_2$ rats were transcardially perfused first with 0.1 phosphate-buffered saline (PBS) followed by 4% PFA. Brains were post-fixed in 4% PFA (4 °C, overnight).

**Immunohistochemistry**. Brains were cut into 40 μm slices, which were collected in cryoprotectant solution and stored at −20 °C until usage. Typically, a series of 6–8 slices comprehending the whole anteroposterior axis of the region of interest were used for immunostaining. First, slices were washed in 0.1 PBS and then rinsed in Glycine buffer (0.1 M in PBS) for 20 min. Afterwards, slices were washed with PBST (0.1 PBS with 0.3% triton-x 100) and blocked for 1 h in blocking solution. Directly after blocking, slices were incubated in primary antibodies for 1–2 h at room temperature and then at 4 °C overnight. On the next day, slices were again left in room temperature for 1–2 h, washed in PBST and incubated with the secondary antibody. Next, slices were rinsed in 0.1 PBS and mounted on adhesive microscope slides (Superfrost Plus, Thermo Fisher Scientific Inc, USA). Slides were kept in the dark at 4 °C until imaging. Especially, for the pERK immunostaining slices were pre-incubate in ice-cold methanol at −20 °C for 10 min before the Glycine buffer step. Also for this staining primary antibody incubation lasted 64 h. For details on tissue, mounting medium, blocking solution, and antibodies, please see Supplementary Table 2. Imaging from the neural activity, patch-clamp, and molecular identification of the LS neurons was done using an inverted confocal laser scanning microscope (Leica TCS SP8, Leica Microsystems, Wetzlar, Germany). Chemo- and optogenetic imaging was performed using an epi-fluorescence microscope (Thunder Imaging Systems, Leica). Digital images were processed (Merging and Z-projections) using the Leica Application Suite X (Leica) and Fiji[70]. Cell counting was done by an experienced observer blind to the treatments. For patch-clamp, the detailed morphology of the neurites was reconstructed and analyzed from the z-stack with the Fiji plugin Simple Neurite Tracer[71]. From this analysis, the number of branch points, and total branch length of the neurites as well as the area of the soma were extracted and compared between dorsal and ventral neurons of the LS.

**Statistics**. Data normality was tested using the Kolmogorov–Smirnov test. If normality was reached, data were analyzed using either Student's *t* test (paired or unpaired), chi-square test, or analyses of variance (one or two-way analysis of variance) followed by a post hoc comparison corrected with Bonferroni, when appropriate. In case data were not normally distributed, either Mann–Whitney *U* test or Dunn's multiple comparison tests were performed. For detailed statistics information see Supplementary Data 1–3.

**Reporting summary**. Further information on research design is available in the Nature Research Reporting Summary linked to this article.

## Data availability

Further information and requests for resources, supporting, and source data will be fulfilled by the corresponding author I.N. Source data are provided with this paper.

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

## Acknowledgements

We thank Rodrigue Maloumby, Anne Pietryga-Krieger, Martina Fuchs, Andrea Havasi, and Thomas Grund for their technical help. VGAT-Venus transgenic rats were generated by Dres Y. Yanagawa, M. Hirabayashi, and Y. Kawaguchi in National Institute for Physiological Sciences, Okazaki, Japan, using pCS2-Venus provided by Dr. A. Miyawaki. The OXTR-A and V1aR-A were kindly provided by Maurice Manning (University of Toledo, Toledo, OH). Monoclonal antibodies anti-neurophysin-OXT (p38) and neurophysin-AVP (p41) were kindly provided by Dr. Harold Gainer (NIH, Bethesda). This work was supported by the EU FP7 Project: Neurobiology and Treatment of Adolescent Female with Conduct Disorder: The Central Role of Emotion Processing Fem-NATCD (602407), GRK2174, BO 1958/8-2, German Research Foundation (DFG) NE465/31 and NE465/33 to I.N. German Research Foundation (DFG) grants LU 2164/1–1 to M.L. and EG135/3-1, EG135/5-1 to V.E. German Research Foundation (DFG) grants GR 3619/7-1, GR 3619/8-1, and GR 3619/13-1, Collaborative Research Center SFB 1158, and Fritz Thyssen Research Foundation (Ref. 10.19.1.015MN) to V.G.

## Author contributions

V.E.M.O. designed and planned the experiments with the assistance of T.d.J. and I.D.N. V.E.M.O. conducted, and analyzed all experiments, prepared the figures, created the illustrations, and wrote the manuscript with the support and input of I.D.N., T.d.J., V.E., M.L., and V.G.; M.L and V.E. assisted with the design, conduction, analyses, and interpretation of electrophysiology experiments. H.W., E.D., A.L., A.L.M. assisted with behavioral experiments. A.B. extracted CSF. O.B. assisted with surgeries and to conduct optogenetic experiments. V.G. shared resources, i.e., rAAV, and assisted with the interpretation and analyses regarding virus infection. T.d.J., I.D.N., and V.E.M.O conceived the project. T.d.J., I.D.N., and V.G. supervised the project and revised the manuscript.

## Funding

## Competing interests

The authors declare no competing interests.
