## [Peer Review File · Nature Communications]

Reviewer #1 (Remarks to the Author):

This is an important and novel study demonstrating a role for the oxytocin (OXT) and vasopressin (AVP) systems acting in the lateral septum (LS) to modulate non-maternal female aggression. This type of aggression is understudied in model organisms, yet female aggression in humans is a key symptom in various psychopathologies for which there is currently no treatment. Thus, this study is an important step towards understanding its biological basis. The choice to focus on OXT and AVP is in line with abundant evidence suggesting key roles of these neuropeptides in various social behaviors with often neuropeptide-specific effects. Findings of this study further underline this by demonstrating opposing central release patterns of OXT and AVP associated with higher female aggression as well as opposing excitability of neurons in the LS by OXT and AVP. Pharmacological, chemogenetic, and optogenetic manipulations of the OXT and AVP systems in the LS provide further evidence for the complex involvement of the LS-OXT and LS-AVP systems in the regulation of non-maternal female aggression. Despite the interesting findings, I have several issues with the experimental design and interpretation of the data that will need to be addressed.

Major comments:

1. The collection of data is novel and interesting, but a major issue is that the authors are over interpreting their data and are drawing conclusions that are misleading. At best, the summarizing figure shows a theoretical model of OXT and AVP actions within the LS to modulate non-maternal female aggression. One weakness is that most experiments were performed with either GH or ISH rats. Complementary experiments with both GH and ISH females are required to provide support for the claim that a balance between OXT and AVP is required in the regulation of non-maternal female aggression. Furthermore, a major limitation is that all in vitro data were collected in juvenile rats (sex was not mentioned). These data in juveniles cannot be used in the context of the female aggression model, especially also because of well-known age differences in OXT and AVP parameters in the LS. Moreover, additional in vivo experiments are required to provide support for claims made about interactions between the vLS and dLS. Finally, in several instances the authors claim that OXT promotes aggression, but this is only true in non-aggressive females, while the opposite is found in aggressive females.
2. The authors indicate that they established a rat model of enhanced female aggression using a combination of social isolation and aggression training (IST). However, their data indicate that social isolation by itself is sufficient to increase aggression. Because subsequent experiments are only performed in IST females, it is unclear whether the observed changes in OXT and AVP systems are due to the FIT training or due to social isolation. Moreover, the levels of aggression induced by IST are highly variable across the experiments (ranging from less than 20% to more than 40%), which questions the reliability and stability of the model. It would have been very helpful to see a comparison with social isolation in all the experiments. Also, the word training in “aggression training” may suggest that there is a specific end goal to be reached, which is not the case here. Therefore, I suggest replacing the term “aggression training” by something along the lines of “repeated aggression exposure”. Finally, if it is the “aggression training” that causes the changes in OXT and AVP signaling in the LS, it is questionable why the authors repeatedly expose GH rats to the FIT in the optogenetic experiments because by doing so they in fact expose the rats to “aggression training”.
3. Overall, experimental detail is lacking. There are many important details lacking in the method and result sections, making it unclear whether the methods and corresponding results are rigorous

and reliable. Here are only a few of the most important ones: It is unclear what controls are used for the DREADDS and optogenetic studies. The authors indicate high specificity of their AAV transductions, but they base this on percentage of total cells, which is not an informative measure. Moreover, the percentage of total cells is exactly the same for the DREADDS and optogenetic studies, which seems impossible. The authors should quantify both the specificity (OXT cells + mCherry cells/mCherry cells) and efficiency (OXT cells + mCherry cells/OXT cells) of the AAV transductions. The observed mCherry cells outside of the PVN and SON should be indicated as percentage of total mCherry cells. It is unclear how the authors can differentiate between extracellular release of OXT and AVP in vLS and dLS, respectively when the microdialysis coordinates are basically the same. The length of the probes should be indicated. It is unclear why juvenile rats were used for the in vitro studies. The sex of the juvenile rats should be indicated. Both age and sex are not trivial factors because OXTR and V1aR binding in the LS differs with age and by sex, as acknowledged by the authors.

4. Correlations between aggression and a chemical measure (as depicted in several figures) is problematic and has little value when the experimental groups already show significant differences in levels of aggression. It would be more valuable to show whether such correlations exist within the IST group.

5. Receptor binding is not a linear measure and thus the density should be converted to disintegrations per minute/milligram (dpm/mg) tissue using a standard microscale.

6. The discussion is disorganized and at times redundant. A considerable amount of the literature that is discussed in detail is not directly relevant to the current findings.

Minor comments

7. Abstract: "In contrast to males, aggression in females has been rarely studied" That should be nuanced because maternal aggression is extensively studied.

8. It would be preferred to see individual data points for all bar graphs.

9. The authors should define vLS and dLS and indicate what rat brain atlas or other references they have used for this. The delineation of vLS and dLS shown in Figs 6 and 7 seems one that is not to be found in any rat brain atlas.

10. It seems that rats for which their behavior is shown in Figure 1 are also used for neuropeptide measures in Figure 2, but the numbers do not add up. Please clarify in the method section if rats in Fig 1 are used for other Figs as well.

11. Please show data for attack latency and number of attacks and please clarify attack %.

12. The SSRI experiment seems disjointed with all other experiments. I suggest to remove this or provide a better rationale for how it relates to the rest of the manuscript.

13. Page 6, line 14-15: "These results demonstrate that the pro-aggressive effect of endogenous OXT is mediated via OXTRs", I don't see experiments or data that provide proof for this.

14. The authors indicate that they infected the PVN and SON using rAAV1/2OTprhM3Dq:mCherry "to

chemogenetically stimulate intracerebral OXT release" (page 5, line 31). This method is not designed to do that. One can say to chemogenetically stimulate OXT synthesizing neurons, but it is unclear whether this chemogenetic stimulation leads to the release of OXT. Given that the authors used microdialysis in the previous experiments, it seems that they missed an opportunity to provide evidence that their DREADDs manipulation indeed increased OXT release within the LS.

15. The order of data is somewhat confusing with central OXT and AVP release as well as brain region specific binding of OTR and V1aR in Fig 2 followed by central and peripheral manipulations of OXT and AVP systems in Fig 3 followed by local brain OXT and AVP manipulations in Figs 4 and 5. I suggest moving the receptor binding data to Figs 4 and 5 respectively.

16. Page 6, line 31-32, to find proof that septal OXT neurotransmission promotes female aggression, the authors should have optogenetically inhibited axons of OXT-synthesizing neurons in the LS.

17. Fig 4C: It seems that one outlier determines significance. Please analyze the data without this outlier.

18. Fig 4f and g: It looks like ChR2-OXT enhances aggression before light stimulation and that a second light stimulation does not alter aggression, both issues should be addressed

19. Suppl Fig 3: It is unclear whether these rats are GH, IS, or IST. Are there any significant effects compared to AVP?

20. For the optogenetic study, females were tested within subject for estrous cycle effect, but it is unclear whether the test order was counterbalanced (repeated exposure to the FIT could have increased aggression) and rather than a two-way ANOVA, data should be analyzed with a three-way ANOVA to also include test day as factor.

21. The authors should indicate the background strain for the VGAT line.

22. The authors need to explain better the relevance of ERalpha and somatostatin. How does this relate to the rest of the manuscript?

Reviewer #2 (Remarks to the Author):

This is an interesting and impressive report investigating neural circuitry and signaling mechanisms underlying aggression in female rodents. This is an understudied area, which coupled with this rigorous analysis, suggest that the manuscript will have a substantial impact on the field. The authors have used a creative approach to enhance and study aggression in non-lactating female rats, with a combination of isolation housing and positive fighting experience (having a smaller intruder mouse introduced into the home cage). Most studies on female aggression have examined lactation-induced aggression, which although interesting, seems rather specific and somewhat lacking in translational relevance. By contrast, the current approach enables dissecting the circuitry and signaling molecules in females without the confounds of the postpartum period. The authors then identify oxytocin as a pro-aggression molecule, with vasopressin having opposite effects, using microdialysis and pharmacological interventions. Next they use chemogenetics and optogenetics to

show that different portions of the lateral septum are affected by these molecules, which together mediate aggressive responses. Lastly, they investigate the electrophysiological responses associated with these changes and find that activation of oxytocin receptors engages GABAergic inhibitory interneurons to inhibit the part of the lateral septum that is important for non-aggressive responses. Taken together, the findings shed much needed light on neural mechanisms of aggression in females. Attention to the following issues would improve the paper:

- 1) More detailed description of the aggressive behaviors is needed in the methods. For example, it is not clear what "threat" means and how it differs from "offensive grooming". Although these are labeled in one of the supplementary videos, they are not obviously different to the untrained viewer. This needs clarification.
- 2) The authors have nicely tracked the estrous cycle in their test rats and shown interesting differences in aggressive behavior at different stages in the cycle (lower aggression in proestrus and estrus, higher in metestrus and diestrus). It is not clear whether they also tracked the estrous cycle of the intruder mouse, which could contribute to differential aggressive responses in the resident. It would be good to explicitly state whether or not this was done and if not, how it is a factor that might affect the behavior.
- 3) Spelling - when referring to the cycle, it's estrous, not estrus. When referring to the individual stages of the cycle, the "o" is omitted and there is no hyphen between the prefix and estrus. For example, metestrus, not met-estrus. This should be corrected throughout the manuscript, including on the graphs.

Reviewer #3 (Remarks to the Author):

de Moura Oliveira et al. studied the participation of oxytocin and vasopressin within the lateral septum subdivisions in female aggression. They make use of a variety of microdialysis, pharmacological, chemogenetic, and electrophysiological methods to show the opposing actions of oxytocin and vasopressin in the modulation of female aggression. Overall, female aggression is not well understood and I believe that this study will be the foundation of important future research. The findings are novel and important and generally the presentation is of high quality. However, the individual values need to be presented for each bar, as has been the standard as of recently. I only have a few concerns, as listed below.

Major concern:

1. Regarding the DREADD experiment (line 31) said to reveal the "involvement of endogenous OXT in female aggression": the author's description with the phrase "endogenous" seems inaccurate. The DREADD expression - controlled by an OT promoter - is not conditionally controlled by the endogenous pattern of OT promoter expression (e.g. as made possible with Cre/loxP systems). Understandably, while many PVN and SON neurons endogenously express OT, any non-OT PVN/SON neurons that are transfected by the AAV will also now express an active OT promoter and hence DREADD. Inasmuch, these experiments do not seem adequately designed to address endogenous OXT. Please rephrase throughout the results, discussion, and manuscript to more accurately reflect what the experiment does address.

Minor concerns:

1. Error bars are missing in Fig 1c.
2. The conflicting findings of elevated OXT (Fig 2a), yet reduced OXTR binding (Fig 2c) in the IST group, warrants interpretation and commentary in the discussion.
3. Regarding Fig 3c, the significance stars are a bit confusing. Are both the OXT and OXTR-A/OXT groups significantly decreased from controls? If so, two sets of significance stars would make these two findings clearer.
4. It appears that you have a typo regarding the scale for the y-axis in Fig. 1d. Should this be scaled up by a factor of 10?
5. Also regarding Figure 1, specifically for the data presented throughout panels b-f: are the data presented from FIT4? Is it lumped from FIT1-4? In either case, this should be made clear. Additionally, the FIT 1, 2, and 3 data from Fig 1 should be separated and included as a supplement, so that consistency across FIT test days can be assessed by the reader, particularly because the FIT 1 data from the IS group is compared to FIT 4 data from the IST group. Furthermore, throughout the manuscript, it is unclear which FIT data is being presented. For example, in Fig 3b, is this the cumulative data of FIT 4 + FIT 6? If so, why does the IST group have two FIT tests combined (4 and 6), while the GH group has only one FIT test (4)? If that is the case, the design of these experiments is mis-balanced.
6. Regarding the statistics utilized for the data presented in Fig 3b: the figure legends describes that t-tests were performed, yet between-group post-hoc comparisons are presented as if an ANOVA was conducted. An ANOVA analysis seems appropriate. Otherwise, the figure should be separated into two figures, to visually indicate that a t-test was performed within each group.
7. There is alternating usage of the acronym SH (Figs. 3a-4a) and "isolated" (Fig 1a) between different figure schematics. This makes the comparison of experimental designs across figures confusing. For readership, please be consistent in this description across the figures.

EDITORIAL COMMENTS

Reviewer #1 (Remarks to the Author):

This is an important and novel study demonstrating a role for the oxytocin (OXT) and vasopressin (AVP) systems acting in the lateral septum (LS) to modulate non-maternal female aggression. This type of aggression is understudied in model organisms, yet female aggression in humans is a key symptom in various psychopathologies for which there is currently no treatment. Thus, this study is an important step towards understanding its biological basis. The choice to focus on OXT and AVP is in line with abundant evidence suggesting key roles of these neuropeptides in various social behaviors with often neuropeptide-specific effects. Findings of this study further underline this by demonstrating opposing central release patterns of OXT and AVP associated with higher female aggression as well as opposing excitability of neurons in the LS by OXT and AVP. Pharmacological, chemogenetic, and optogenetic manipulations of the OXT and AVP systems in the LS provide further evidence for the complex involvement of the LS-OXT and LS-AVP systems in the regulation of non-maternal female aggression. Despite the interesting findings, I have several issues with the experimental design and interpretation of the data that will need to be addressed.

Major comments:

1. The collection of data is novel and interesting, but a major issue is that the authors are over interpreting their data and are drawing conclusions that are misleading. At best, the summarizing figure shows a theoretical model of OXT and AVP actions within the LS to modulate non-maternal female aggression. One weakness is that most experiments were performed with either GH or ISH rats. Complementary experiments with both GH and ISH females are required to provide support for the claim that a balance between OXT and AVP is required in the regulation of non-maternal female aggression.

Reply: First, we would like to thank the reviewer for the encouraging comments. Indeed, we have performed on purpose some experiments either in GH or IST rats. We based this decision on the opposite patterns of intracerebral OXT and AVP release (CSF levels or intra-LS release by microdialysis; Figures 2a and 2d, 4c, 5b) during/after aggressive encounters. Our data show increased OXT release and decreased AVP release in highly aggressive IST females in comparison to GH rats, providing a first indication for an anti-aggressive effect of AVP and a pro-aggressive effect of OXT. In order to confirm this hypothesis, we used pharmacological, chemo- and optogenetic manipulations of those neuropeptidergic systems using those release patterns as guides, i.e. we increased OXT levels and OXT system activity in low aggressive GH rats to mimic the high OXT release seen in IST females, whereas we blocked OXTR signaling in highly aggressive IST females, to resemble the low activity and the lack of releases found in GH females; indeed those bidirectional manipulations confirmed that a high activity (release) of the OXT system underlies high levels of female aggression. Similarly, we administrated synthetic AVP to aggressive IST females (which have no AVP release during aggression), and found reduced aggression, whereas blockade of V1aRs was used to enhance their aggression. As the reviewer pointed out, we indeed did not perform all experiments in both directions in both low aggressive GH and high aggressive IST rats, as, for example, the low level of aggression of GH rats might not be further reduced, e.g. by inhibition of the OXT system, or

application of AVP (**floor effect**). However, in order to provide further evidence for this, and following the suggestion of the reviewer we performed an additional experiment and blocked V1aRs in GH rats and monitored their aggressive behavior; this data is now included in **Figure 5f** and in the **manuscript page 7 line 22-24**. Moreover, we added information about the housing conditions in **Figure 8** to clarify the patterns of neuropeptide release. In general, we would like to highlight our commitment to the 3R rule (replacement, reduction, and refinement of animal experiments), which is another reason to avoid performing manipulations across both housing conditions, when such experiments would definitely not help to support our scientific hypothesis. Thus, we hope that with the new data included from the additional experiment, the corrections performed in the ms and detailed explanations provided to meet the reviewer's expectations.

Furthermore, a major limitation is that all in vitro data were collected in juvenile rats (sex was not mentioned). These data in juveniles cannot be used in the context of the female aggression model, especially also because of well-known age differences in OXT and AVP parameters in the LS.

Reply: Indeed, we agree with the reviewer that sex and age effects are reported for the OXT and AVP receptor distribution¹. We apologize that we forgot to specify the sex of the rats used in the in-vitro study. We now report on **p 20-25 (page 7 line 33 and page 14 line 1)** that only **female juvenile rats** were used for the patch-clamp slice experiments. Although data on sex differences is rather robust and features large effects, data on age differences is not that clear. Smith et al. (2016)¹ reports lower septal AVP receptor binding in juvenile compared to adult male and female rats, whereas another publication from the same group reports only sex differences, but no age difference using also receptor autoradiography². Additionally, as shown by the earlier study, blocking septal V1aRs with a specific AVP receptor antagonist disrupts social discrimination, an AVP-dependent behavior, in both juvenile and adult female rats. This result indicates that the potential difference in receptor density may not be functionally relevant for social behaviors. Concerning OXT receptors, Smith et al. (2016)¹ demonstrates that juveniles have a significantly higher binding density compared to adults, however from an already high baseline (approx. 17000 in juveniles vs. 15000 in adults d.p.m/mg) with a lower effect size in females than in males. Whether this difference has functional relevance for social behavior is, to our knowledge, not tested yet. However, we now mention this limitation of our in-vitro results in the discussion **(page 13, line 11-14)**. Finally, we wish to point out that there are no main results of our study that rest solely on the electrophysiology data; rather, these data support core findings and also serve to generate further working hypotheses on the underlying network architecture.

Moreover, additional in vivo experiments are required to provide support for claims made about interactions between the vLS and dLS. Although your data support the hypothesis that OXT release in the vLS facilitates aggression via increasing tonic inhibition onto dLS neurons, whether those phenomena are functionally associated with each other remains to be determined in further studies.

Reply: As we mentioned in the discussion **page 13 line 8-11** there is compelling evidence for vLS neurons projecting to the dLS. In addition, our data further support this hypothesis by showing that i) OXT acting via OXTRs in the vLS promotes female aggression, ii) activation of

OXTRs in the vLS leads to increased tonic inhibition onto dLS neurons, and iii) inhibition of the dLS neurons via muscimol increases female aggression. However, as the reviewer pointed out, whether those three phenomena are functionally connected remains to be further investigated, therefore we have added such a statement to the discussion (page 13, line 11-14).

Finally, in several instances the authors claim that OXT promotes aggression, but this is only true in non-aggressive females, while the opposite is found in aggressive females.

Reply: The reviewer points to a fascinating phenomenon, which we have found and tried to explain. We kindly ask the reviewer to refer to our results (page 5-7) and discussion (page 9-13) sections. There, we clearly distinguish between the effects of *synthetic* and *endogenous* OXT in our animal model: “Taken together, these results demonstrate that the **pro-aggressive effect of endogenous OXT** is mediated **via OXTRs**, whereas the **anti-aggressive effects of synthetic OXT** and, particularly, synthetic AVP are mediated **via V1aRs (page 6 lines 4-6)”**. Thus, the anti-aggressive effect of synthetic OXT is due to the manifold description of OXT acting also unspecifically at AVP receptors^{3,4}. In this context it is important to highlight that we have confirmed the **pro-aggressive** effects of **endogenous OXT** acting via **OXTRs** by demonstrating that i) that OXT is released in the CSF and within the LS during aggression, ii) blockade of OXTR i.c.v. and within the vLS reduced female aggression, and iii) stimulation of either intracerebral or local (vLS) OXT release via chemo- and optogenetic approaches enhanced the mild levels of aggression displayed by GH rats. However, as the reviewer mentioned, in highly-aggressive IST females, which already show a high level of OXT release (**Figure 2 and 4**) during FIT exposure, further elevation of OXT levels via infusion with **synthetic OXT** evokes unspecific binding to V1aRs, probably due to the high occupancy of OXTRs by *endogenous* OXT. Altogether we assume this data strongly corroborates the fact that **endogenous OXT** acting via OXTRs promotes aggression in female rats, whereas activation of V1aR receptors either via **endogenous/synthetic AVP** or **synthetic OXT** decreases aggression.

2. The authors indicate that they established a rat model of enhanced female aggression using a combination of social isolation and aggression training (IST). However, their data indicate that social isolation by itself is sufficient to increase aggression. Because subsequent experiments are only performed in IST females, it is unclear whether the observed changes in OXT and AVP systems are due to the FIT training or due to social isolation.

Reply: The reviewer is correct that social isolation by itself is sufficient to increase female aggression. However, differently from IST rats, IS females are particularly influenced by the estrous-cycle (Figure 1e), e.g., exhibiting reduced levels of aggression, when they are in the proestrus or estrus phase of the cycle. Such cycle-related variability could potentially bias the interpretation of our pharmacology results, especially considering that those females would have been used aiming to reduce isolation-induced aggression. Moreover, only IST females present alterations in both neuropeptidergic systems after an aggressive encounter as seen in Figure 2. In detail, both IS and IST females show increased levels of OXT in their CSF, whereas only IST females show decreased levels of AVP in the CSF, which has been further confirmed by their reduced AVP release locally in the LS (data not shown in the main text, but depicted in **Figure 1 below**). Another argument to only use IST rats for most experiments was the fact

that their high level of aggression was more stable and reliable compared with IS rats. Therefore, we intentionally used only IST (modeling high aggression) and GH (modeling low aggression) females to bidirectionally manipulate female aggression via acting on both neuropeptidergic systems. To what extent the previous aggression training and experience of being aggressive alter OXT/AVP systems is unknown, but from a translational perspective, aggressive perpetrators surely engage in repeated aggressive interactions. **Answering the reviewer's question more specifically, as seen in Figure 2, the OXT system seems to be influenced mostly by isolation, as both IS and IST females presented higher levels of OXT in their CSF. In contrast, the AVP system seems to be mostly influenced by aggression experience since exclusively IST females presented reduced AVP content in their CSF and LS.**

Figure 1. Figure showing AVP release within the LS in isolated and trained (IST), isolated IS and group-housed (GH) female Wistar rats during female intruder test (FIT) exposure. *vs Baseline and # vs GH.

Moreover, the levels of aggression induced by IST are highly variable across the experiments (ranging from less than 20% to more than 40%), which questions the reliability and stability of the model.

Reply: As any other innate behavior displayed by rodents, aggression differs across populations as it has been shown by us⁵ and other groups^{6,7}. In fact, **male** Wistar rats are known to be quite gregarious, with approximately 55% of the population expressing low levels of aggression (under 15%)^{5,7}, which is in line with our findings on GH female Wistar rats (5-15% of time spent on aggression). Hence, establishing a behavioral protocol that raises the mild levels of female aggression seen in Wistar rats above this threshold (more than 15% of time spent on aggression) is already a great achievement in our opinion, as such a protocol has never been described before. Accordingly, using our isolation and training protocol we could evoke attacks in approximately 50% of the IST rats (**Figure 1c**). Additionally, those animals spent more than 25% of their time on aggression (2.5 min out of a 10 min FIT) when they were either in metestrus or diestrus (**see Figure 1c and e**). Apart from that, it is important to highlight that aggressive behavior is influenced by several factors such as seasons⁵, photoperiod⁸, social dominance status^{6,9}, and ventilation conditions¹⁰. Although we have kept other factors such as housing conditions, training protocol, experimental design, and group sizes as identical as possible across experiments, we expected and were not surprised to find some variability in different cohorts, especially knowing that the experiments presented in

this manuscript were conducted over the last 4-5 years. Importantly, even though aggressive behavior may have differed among cohorts, we could always confirm that social isolation and training increased aggression levels in all the cohorts evaluated, reflecting the reliability and robustness of our protocol. Nevertheless, we additionally provided in **Supplementary table 2** all means±SEMs of aggression levels displayed by IST rats from all experiments performed in this manuscript. Please note that the amount of aggression displayed by rats actually varies in dependence on the type of infusion the animals have undergone. Typically, after i.c.v. or i.p. infusion, IST rats displayed between 30-40% of the time with aggression, whereas after local infusions they displayed around 18-30 % of the time being aggression. From our point of view, this again reflects i) that we consistently were able to raise the levels of female aggression with our protocol above the typical threshold described in the literature (under 15%)^{5,7} and ii) the stability and strength of our model.

It would have been very helpful to see a comparison with social isolation in all the experiments.

Reply: As we have explained in detail above (**first answer for comment 2**), our aim in the present study was to establish a reliable rat model of enhanced female aggression to study the detailed involvement of septal OXT and AVP in female aggression. Thus, social isolation and aggression training were used here as tools to reliably instigate female aggression. Our aim was not to directly test, whether different housing conditions affect the OXT and AVP systems, which has to be addressed in future studies. Additionally, we want to again reinforce that having IS animals in all experiments would largely increase animal numbers without contributing to test our main hypotheses or to achieve our scientific goal which was to study the involvement of the OXT and AVP system on female aggression and not on social isolation. Therefore, we decided to avoid such unnecessary experiments also from an animal ethical point of view (3 R rule).

Also, the word training in “aggression training” may suggest that there is a specific end goal to be reached, which is not the case here. Therefore, I suggest replacing the term “aggression training” by something along the lines of “repeated aggression exposure”. Finally, if it is the “aggression training” that causes the changes in OXT and AVP signaling in the LS, it is questionable why the authors repeatedly expose GH rats to the FIT in the optogenetic experiments because by doing so they in fact expose the rats to “aggression training”.

Reply: Indeed, as defined in the manuscript introduction aggression training was part of the novel protocol which we established to instigate aggression in female Wistar rats (**page 2, line 19-23**) **“We predicted that a combination of social isolation and aggression-training by repeated exposure to the female intruder test (FIT), i.e. to an unknown same-sex intruder, enhances female aggressiveness, since both social isolation and repeated engagement in conflict with conspecifics (“winner effect”), exacerbate aggression in solitary and aggressive rodent species, independent of sex”.** We have now provided a clear definition of what aggression training means, i.e. repeated aggression exposure, **in our study**. Importantly, from a translational point of view perpetrators might engage in multiple aggressive interactions, in fact, there is compelling evidence showing that recidivism increases aggression in humans¹¹⁻¹⁴. We also kindly ask the reviewer to see the comments above in which we dissociate the effects of social isolation from the effects of repeated exposure itself. Indeed, as the reviewer pointed out, re-exposing GH animals to an aggressive encounter could

potentially increase their aggressive behavior. However, this does not seem to be the case in our experiments after 2 training exposures. As the reviewer can see in **Table 1 (below)** there is no significant difference between the levels of aggression displayed by GH rats in the first and the second FIT exposure for either the chemo- or optogenetic manipulations. In accordance with the data depicted in **Figure 3** and **4**. We hope our changes in the main text and answer can fully address the reviewer's concerns.

Experiment	Chemogenetics		Optogenetics	
	FIT1	FIT2	FIT1	FIT2
Aggression mean±SEM (%)	3.19±1.02	7.12±1.97	5.3±2.34	9.77±2.56
Wilcoxon test	p= 0.15		p= 0.31	

Table 1. Showing the effect of multiple FIT exposures on aggression in the chemo- and optogenetic experiments.

3. Overall, experimental detail is lacking. There are many important details lacking in the method and result sections, making it unclear whether the methods and corresponding results are rigorous and reliable. Here are only a few of the most important ones:

- It is unclear what controls are used for the DREADDS and optogenetic studies.

Reply: We apologize for not having included sufficient information regarding the detailed methods applied. Thus, we would like to thank the reviewer for the suggestion for increasing the transparency of the methods performed. Regarding the controls used for the DREADD experiment, controls consisted of three groups: (i) subjects infected with an empty rAAV1/2 OXTpr-mCherry into the PVN and SON, which received CNO (virus control), non-infected rats, which received either i.p. (ii) CNO or (iii) saline infusions (drug control). Importantly, since those groups did not show differences in terms of percentage of time displaying aggression, they were pooled in a single control group depicted as gray bar charts in **Figure 3**. For optogenetic studies controls consisted of GH rats that were infected with an empty rAAV1/2 OXTpr-mCherry and underwent the same stimulation protocol received by rAAV1/2 OXTpr-ChR2-mCherry rats. This information is now included in the methods section **page 16 lines 11-22**.

- The authors indicate high specificity of their AAV transductions, but they base this on percentage of total cells, which is not an informative measure. Moreover, the percentage of total cells is exactly the same for the DREADDS and optogenetic studies, which seems impossible. The authors should quantify both the specificity (OXT cells + mCherry cells/mCherry cells) and efficiency (OXT cells + mCherry cells/OXT cells) of the AAV transductions. The observed mCherry cells outside of the PVN and SON should be indicated as percentage of total mCherry cells.

Reply: As suggested by the reviewer this information is now included in the manuscript **pages 5 and 6 line 24-28 and 24-30**, respectively.

- It is unclear how the authors can differentiate between extracellular release of OXT and AVP in vLS and dLS, respectively when the microdialysis coordinates are basically the same. The length of the probes should be indicated.

Reply: We apologize for the mistake in the coordinates for the implantation of the microdialysis probes into either the dLS, which indeed differ. This information is now corrected in **Table 4**.

- *It is unclear why juvenile rats were used for the in vitro studies. The sex of the juvenile rats should be indicated. Both age and sex are not trivial factors because OXTR and V1aR binding in the LS differs with age and by sex, as acknowledged by the authors.*

Reply: We have included now the sex of the juveniles for the in vitro experiments in the main text now. Furthermore, we kindly point towards the second reply to the reviewer's first comment, where we also discuss the age and sex differences regarding the OXT and AVP systems.

4. Correlations between aggression and a chemical measure (as depicted in several figures) is problematic and has little value when the experimental groups already show significant differences in levels of aggression. It would be more valuable to show whether such correlations exist within the IST group.

Reply: We thank the reviewer for the valuable suggestion. Accordingly, correlations between aggression and chemical measures are now removed from the main text and figures.

5. Receptor binding is not a linear measure and thus the density should be converted to disintegrations per minute/milligram (dpm/mg) tissue using a standard microscale.

Reply: Indeed, the reviewer is completely right that receptor autoradiography (RAR) is a non-linear measure. Thus, the density of receptors is not a direct function of the light absorption by the film, but an indirect function that depends on several factors like the radioactivity used to label the probe, but also the binding affinity of the antagonist and others. Hence, as pointed out by the reviewer, the use of calibration standards is necessary to accurately measure binding densities in dpm (disintegrated points per minute) or cpm (counts per minute). However, even when calibration standards are used this method remains a **semi-quantitative approach**, as also these radioactivity measurements cannot directly be transformed into absolute numbers of receptor densities due to other missing parameters. Nevertheless, to address the concern of the reviewer, we want to elaborate that the measurable signal of the radioactive receptor probes follows a logarithmic scale, and consequently, the use of gray values could lead to an under-representation of actual receptor binding^{15,16}, the more as we did not use calibration standards for our films. However, we would like to emphasize that in our study RAR was simply used as a tool to reveal the brain area (LS) most likely to be affected by the isolation and training for subsequent functional experiments. Moreover, we do not draw any main conclusions from these RAR differences in the LS between the experimental groups, but rather that this brain area is most likely to be strongly involved in aggression regulation, what we indeed confirmed via pharmacologic and optogenetic manipulations. However, as relative OD has been accepted before as a valid measure of receptor binding $ROD = \log_{10}[1/\text{gray level} \times 256^{-1}]^{16}$, we converted all LS RAR data to ROD and could confirm our results presented in the main manuscript, which are now depicted **in Table 2 (below) and are included into the**

manuscript. We hope those comments together with the further analyses included (Table 2) can meet the reviewer's concerns.

Area	dLS		vLS	
Groups	GH	IST	GH	IST
OD mean±SEM	0.53±0.007	0.47±0.005	0.41±0.006	0.37±0.009
ANOVA Dunn's	p≤0.01		p≤0.05	

6. *The discussion is disorganized and at times redundant. A considerable amount of the literature that is discussed in detail is not directly relevant to the current findings*

Reply: According to the reviewer's general comment on the discussion, the authors carefully revised the entire discussion removing references that were not aligned with the topic and shortened the discussion.

Minor comments

7. *Abstract: "In contrast to males, aggression in females has been rarely studied" That should be nuanced because maternal aggression is extensively studied.*

Reply: The reviewer addresses the important differentiation between female aggression (expressed by virgins) and maternal aggression, a specific social behavior described as part of the behavioral repertoire of dams, which we have substantially studied during the last 20 years¹⁷⁻²⁶. We kindly invite the reviewer to go to the introduction, where we emphasize the importance of studies conducted in lactating rats on **page 2 line 10-11**. **"However, these models were mostly developed in male rodents, whereas females have been rather understudied, except during the physiologically unique period of lactation"**. In contrast to the many studies on lactating rats and maternal aggression (easy to study due to the high level of aggression they display) female aggression displayed by non-lactating, i.e. virgin females, has been by far understudied in animals and humans as stated by many researchers²⁷⁻³². Finally, it is important to bring into to picture the fact that lactation is a unique period, when several physiological systems, including the OXT and AVP systems, but also other hormonal systems, show dramatic changes, which do not reflect the physiology of virgin females^{17,19,27,30,33-35}. Taking into account that girls and women also exhibit aggression^{27,29,36-38}, it is essential to establish animal models to study the neurobiological mechanisms underlying female aggression in virgins, i.e. also for translational reasons. To emphasize this important point, we adapted the sentence on, **(page 1 line 19)**, "In contrast to males, aggression in **virgin** females has been rarely studied". We hope with this change we can meet the reviewer's concern.

8. *It would be preferred to see individual data points for all bar graphs.*

Reply: According to reviewers' #1 and #3 suggestions, we decided to include individual data points in all the main figures in the manuscript.

9. *The authors should define vLS and dLS and indicate what rat brain atlas or other references they have used for this. The delineation of vLS and dLS shown in Figs 6 and 7 seems one that is not to be found in any rat brain atlas.*

Reply: We thank the reviewer for this important suggestion. We defined the dLS and vLS delimitation according (i) to the Paxinos Rat Atlas (definition of LS) and (ii) to the distinct distribution of V1a and OXT receptor binding in either the dLS and vLS, respectively. This information is now included in the method section **(page 17 lines 10-11)**.

10. *It seems that rats for which their behavior is shown in Figure 1 are also used for neuropeptide measures in Figure 2, but the numbers do not add up. Please clarify in the method section if rats in Fig 1 are used for other Figs as well.*

Reply: We thank the reviewer for the valuable suggestion. Indeed, measures shown in **Figure 2** come from animals used in **Figure 1** this information is now highlighted in the method section **(pages 14 and 15 lines 27-33 and 4-8, respectively)**. We also now further explain that animals were split into two groups according to the peptide measured, i.e. half of the subjects were used for OXT RIA, whereas the other half was used for AVP RIA. Additionally, we had to exclude some animals from which we could not extract CSF or CSF samples were contaminated by blood. We hope this explanation clarifies the reviewer's concerns.

11. *Please show data for attack latency and number of attacks and please clarify attack %.*

Reply: We kindly invite the viewer to check the **Supplementary table 2**, where attack latencies and frequencies are displayed for all experiments. Attack (%) refers to the % of time spent attacking, this information is now included in **Figure legend 1**.

12. *The SSRI experiment seems disjointed with all other experiments. I suggest to remove this or provide a better rationale for how it relates to the rest of the manuscript.*

Reply: We thank the reviewer for this valuable suggestion, as we have already discussed this issue intensively. However, this is the first paper on this novel animal model on female aggression, and, thus, we believe that besides dissecting the effects of OXT and AVP on aggression we also aim to show its predicted validity by presenting the SSRI data. That is especially relevant because SSRIs are known to typically decrease male aggression³⁹, as we have mentioned on **page 4, line 5-7**. We hope the reviewer can agree to this decision.

13. *Page 6, line 14-15: "These results demonstrate that the pro-aggressive effect of endogenous OXT is mediated via OXTRs", I don't see experiments or data that provide proof for this.*

Reply: The experiment in which OXTRs were blocked via a specific OXTR antagonist **(Figure 3e)** provides evidence that the pro-aggressive effect of endogenous OXT is mediated via OXTRs. Specifically, we have first shown that IST females show increased OXT levels in the CSF after aggression **(Figure 2a)**, pointing towards the pro-aggressive effect of OXT in those animals. To make a causal link between *endogenous* OXT release and OXTR signaling we blocked central OXTRs with a selective OXTR-A before the FIT, which resulted in a reduction in aggressive behavior displayed by IST females confirming the involvement of OXTR signaling in female aggression **(Figure 3d)**. The use of the specific OXTR-A (provided by Maurice Manning) to suppress the OXTR-mediated behavioral effects of endogenous OXT has been established by us and many others in innumerable behavioral paradigms that are OXT-dependent, such as social memory^{40,41}, social preference⁴², social fear conditioning^{35,43}, intermale aggression⁴⁴, maternal aggression¹⁸, and pair-bonding and consolation

behavior⁴⁵.

14. *The authors indicate that they infected the PVN and SON using rAAV1/2OTprhM3Dq:mCherry “to chemogenetically stimulate intracerebral OXT release” (page 5, line 31). This method is not designed to do that. One can say to chemogenetically stimulate OXT synthesizing neurons, but it is unclear whether this chemogenetic stimulation leads to the release of OXT. Given that the authors used microdialysis in the previous experiments, it seems that they missed an opportunity to provide evidence that their DREADDs manipulation indeed increased OXT release within the LS.*

Reply: The reviewer raises an important issue here, i.e. whether chemogenetic activation of OXT neurons indeed results in increased OXT release. This has been successfully demonstrated before⁴⁶, i.e. we have shown that CNO application results in elevated OXT release in the PVN and into the blood. Although it has not specifically been shown whether chemogenetic stimulation of hypothalamic OXT neurons stimulates release within the septum, in our experiment, this approach has been used to target the brain as a whole without localizing the effects of OXT in a specific brain region.

As the reviewer can see throughout the section “OXT promotes, whereas AVP reduces female aggression” (pages 5-6) and our discussion (page 10), we did not claim that chemogenetic stimulation of OXT neurons in the PVN and SON triggers OXT release within the LS. We carefully state that chemogenetic stimulation of those neurons leads to intracerebral (central) OXT release. This has been indeed confirmed by another paper from our group where chemogenetic stimulation of OXT neurons resulted in OXT release in the PVN and blood⁴⁶; this reference has already been previously included in the result section (page 5, line 27). We hope this answer clarifies the reviewer's inquiries.

15. *The order of data is somewhat confusing with central OXT and AVP release as well as brain region specific binding of OTR and V1aR in Fig 2 followed by central and peripheral manipulations of OXT and AVP systems in Fig 3 followed by local brain OXT and AVP manipulations in Figs 4 and 5. I suggest moving the receptor binding data to Figs 4 and 5 respectively.*

Reply: We thank the reviewer for his thoughtful suggestion. However, after extensive discussion, we suggest to keep the order of presenting our results according to our scientific hypotheses, i.e.: i) showing the behavioral data and establishment of the behavioral model of female aggression (Figure 1), (ii) followed by the analyses of the endogenous OXT and AVP systems, i.e., neuropeptide content, release as well as receptor binding (Figure 2) in the very same animals used for behavioral analyses. (iii) Next, we focus on central (Figure 3) local (Figure 4 and 5) manipulations. (iv) Finally, in Figures 6 and 7 we explore the underlying circuits of female aggression. We hope the reviewer understands our perspective.

16. *Page 6, line 31-32, to find proof that septal OXT neurotransmission promotes female aggression, the authors should have optogenetically inhibited axons of OXT-synthesizing neurons in the LS.*

Reply: Indeed this is an interesting suggestion and additional option to show that LS OXT neurotransmission is involved in female aggression. However, as already explained above

in the reply to **comment 13**, the use of the OXTR-A antagonist is well accepted in the scientific community to suppress the effects of OXT receptor-mediated neurotransmission in OXT-dependent behaviors. Here, specifically, we have shown that IST females exhibit increased OXT release within the vLS; in order to build a causal link between OXT release and aggressive behavior we blocked the OXTRs using a specific OXTR antagonist, which prevents endogenous OXT of acting on OXTRs and, consequently, to promote female aggression.

17. Fig 4C: It seems that one outlier determines significance. Please analyze the data without this outlier.

Reply: As requested by the reviewer all the correlations are now removed from the manuscript.

18. Fig 4f and g: It looks like Chr2-OXT enhances aggression before light stimulation and that a second light stimulation does not alter aggression, both issues should be addressed

Reply: The reviewer points towards an important issue: As group-housed females were not pre-exposed to the FIT before surgery, those rats were not counterbalanced by the levels of aggression as explained in the Methods (**page 15 lines 19-22**), thus this created some asymmetry between groups. However, the levels of aggression of control animals did not significantly differ from the Chr2-OXT animals for pre-stimulation time points. Also, only Chr2- OXT animals showed a sharp and statistically significant increase in aggression in response to optogenetic stimulation (**Figure 4e and g**). Surprisingly for us, the second stimulation did not trigger an increased level of aggression, this could be due to several reasons, such as the depletion of the OXT system, ceiling effects due to already quite high levels of aggression, or binding to V1aR after further OXT release. These hypotheses, which need to be further addressed in follow up studies, are discussed in the manuscript (**page 7, lines 5-7**) now.

19. Suppl Fig 3: It is unclear whether these rats are GH, IS, or IST. Are there any significant effects compared to AVP?

Reply: Sorry for not being clearer, but the reviewer probably missed the information about the housing condition in the figure legend, where we describe that all animals used in those experiments were isolated and trained (IST). Additionally, there were only effects comparing AVP groups with the vehicle group, but none within AVP doses. We hope this answer clarifies the reviewer's questions.

20. For the optogenetic study, females were tested within subject for estrous cycle effect, but it is unclear whether the test order was counterbalanced (repeated exposure to the FIT could have increased aggression) and rather than a two-way ANOVA, data should be analyzed with a three-way ANOVA to also include test day as factor.

Reply: We kindly refer the reviewer to our answer to the major **comment number 2**, where we already addressed this point. As mentioned before as GH females were not trained, and, therefore, we did not counterbalance their levels of aggression as detailed in the Methods section (**page 15 lines 19-22**). As stated in the manuscript GH rats were tested twice in

different phases of the estrous cycle, which could potentially lead to training-enhanced aggression. However, as we depict in the **Rebuttal table 2** such effect is not likely, as the second re-exposition to the FIT did not influence the levels of aggression of those subjects. This is further reinforced by our data where three and not two re-expositions to the FIT are needed to elicit stable and high levels of aggression in the IST group **Supplementary table 6 and Supplementary figure 7**.

21. *The authors should indicate the background strain for the VGAT line.*

Reply: We apologize for not including this information previously. The background of VGAT animals belongs to the Wistar strain, and this information is now indicated on **page 14 line 2**.

22. *The authors need to explain better the relevance of ERalpha and somatostatin. How does this relate to the rest of the manuscript?*

Reply: We apologize for not having made this point clearer. In the result section, we more extensively argue now: “Apart from the specific expression of V1aRs and OXTRs, dorsal and ventral LS neurons also differed regarding the expression of other markers: somatostatin-positive cell bodies were only found in the dLS (Figure 6g), whereas estrogen receptor α (ER α) expressing cells were exclusively located in the vLS (Figure 6h), further reinforcing that vLS and the dLS neurons are distinct populations” (**page 8, line 7-11**), and hope this addresses the question of the reviewer.

Reviewer #2 (Remarks to the Author):

This is an interesting and impressive report investigating neural circuitry and signaling mechanisms underlying aggression in female rodents. This is an understudied area, which coupled with this rigorous analysis, suggest that the manuscript will have a substantial impact on the field. The authors have used a creative approach to enhance and study aggression in non-lactating female rats, with a combination of isolation housing and positive fighting experience (having a smaller intruder mouse introduced into the home cage). Most studies on female aggression have examined lactation-induced aggression, which although interesting, seems rather specific and somewhat lacking in translational relevance. By contrast, the current approach enables dissecting the circuitry and signaling molecules in females without the confounds of the postpartum period. The authors then identify oxytocin as a pro-aggression molecule, with vasopressin having opposite effects, using microdialysis and pharmacological interventions. Next they use chemogenetics and optogenetics to show that different portions of the lateral septum after affected by these molecules, which together mediate aggressive responses. Lastly, they investigate the electrophysiological responses associated with these changes and find that activation of oxytocin receptors engages GABAergic inhibitory interneurons to inhibit the part of the lateral septum that is important for non-aggressive responses. Taken together, the findings shed much needed light on neural mechanisms of aggression in females. Attention to the following issues would improve the paper:

Reply: We would like to thank the reviewer for the kind and very useful comments, which helped to improve the quality of our manuscript.

1) *More detailed description of the aggressive behaviors is needed in the methods. For example, it is not clear what "threat" means and how it differs from "offensive grooming". Although these are labeled in one of the supplementary videos, they are not obviously different to the untrained viewer. This needs clarification.*

Reply: We now included a detailed explanation of the aggressive behaviors quantified during the FIT in the methods **page 14 line 11-16**. We hope this clarifies the reviewer's question.

2) *The authors have nicely tracked the estrous cycle in their test rats and shown interesting differences in aggressive behavior at different stages in the cycle (lower aggression in proestrus and estrus, higher in metestrus and diestrus). It not clear whether they also tracked the estrous cycle of the intruder mouse, which could contribute to differential aggressive responses in the resident. It would be good to explicitly state whether not this was done and if not, how it is a factor that might affect the behavior.*

Reply: We very much appreciate this valuable comment. Unfortunately, we did not keep track of the intruder estrous cycle. We completely agree that this factor should be taken into account in future studies, as it could possibly impact the aggressive behavior of the resident. As suggested by the reviewer, we have added this information in the Methods section **(pages 14, lines 21-24)** now.

3) *Spelling - when referring to the cycle, it's estrous, not estrus. When referring to the individual stages of the cycle, the "o" is omitted and there is no hyphen between the prefix and estrus. For example, metestrus, not met-estrus. This should be corrected throughout the manuscript, including on the graphs.*

Reply: We apologize for this spelling mistake, this is now corrected over the entire manuscript.

Reviewer #3 (Remarks to the Author):

de Moura Oliveira et al. studied the participation of oxytocin and vasopressin within the lateral septum subdivisions in female aggression. They make use of a variety of microdialysis, pharmacological, chemogenetic, and electrophysiological methods to show the opposing actions of oxytocin and vasopressin in the modulation of female aggression. Overall, female aggression is not well understood and I believe that this study will be the foundation of important future research. The findings are novel and important and generally the presentation is of high quality. however, the individual values need to be presented for each bar, as has been the standard as of recently. I only have a few concerns, as listed below.

Reply: We appreciate the reviewer's encouraging and kind words. In the revised manuscript, all main figures are displayed showing single values now, as requested by both reviewers #1 and #3.

Major concern:

1. *Regarding the DREADD experiment (line 31) said to reveal the “involvement of endogenous OXT in female aggression”: the author’s description with the phrase “endogenous” seems inaccurate. The DREADD expression - controlled by an OT promoter - is not conditionally controlled by the endogenous pattern of OT promoter expression (e.g. as made possible with Cre/loxP systems). Understandably, while many PVN and SON neurons endogenously express OT, any non-OT PVN/SON neurons that are transfected by the AAV will also now express an active OT promoter and hence DREADD. Inasmuch, these experiments do not seem adequately designed to address endogenous OXT. Please rephrase throughout the results, discussion, and manuscript to more accurately reflect what the experiment does address.*

Reply: The issue regarding the specificity of neuronal DREADD expression raised by the reviewer is extremely important. However, we would like to emphasize that the rAAV vector used is equipped with OXT promoter allowing specific expression of DREADD or ChR2 only in oxytocin neurons, but not affecting OXT gene per se as it is not integrated into the genome. Therefore, the virally-delivered OXT promoter is not active in non-OXT cells due to the lack of transcriptional factors that are essential for OXT promoter transcription⁴⁶⁻⁵⁰. Thus, manipulation of OXT neuron activity via their infection with this virus expressing either DREADD or ChR2 results in changes of endogenous (i.e. intracerebral) OXT release including from distant oxytocinergic axons in the lateral septum³⁵.

Minor concerns:

1. *Error bars are missing in Fig 1c.*

Reply: We thank the reviewer for the careful reading, but this graph does not demand error bars, since it only depicts qualitative data representing the percentage of animals in each group which showed attacks.

2. *The conflicting findings of elevated OXT (Fig 2a), yet reduced OXTR binding (Fig 2c) in the IST group, warrants interpretation and commentary in the discussion.*

Reply: We thank the reviewer for pointing out this interesting phenomenon. Indeed, this result may appear conflicting at first glance. However, OXT release and OXTR binding are not linearly correlated, i.e. a high level of local OXT release is not always associated with increased OXTR binding³³, except in lactation, when suckling-induced OXT release is found at a stage of general increased OXTR expression and binding.³³ Also, when compared to females, male Wistar rats exhibit both higher OXTR binding and elevated OXT release in the posterior BNST after social contact⁴⁰. In contrast, there are many examples of a mismatch between OXT release and OXTR binding^{18,43,51}. For example, in the context of social fear, increased OXTR binding has been found in parallel to reduced OXT release in the lateral septum in social fear-conditioned mice⁴³. Accordingly, i.c.v. infusion of *synthetic* OXT, increasing its availability in the brain, results in reduced OXTR binding in several brain regions including the LS⁵¹. Additionally, other factors apart from OXT release are known to

influence OXTR binding³³. Thus, the precise factors regulating OXTR binding in the septum in association with housing conditions and aggression training are still unknown and need to be further dissected. As suggested by the reviewer, we have included a statement into the discussion **(page 11 lines 6-9)** pointing towards this phenomenon.

3. *Regarding Fig 3c, the significance stars are a bit confusing. Are both the OXT and OXTR-A/OXT groups significantly decreased from controls? If so, two sets of significance stars would make these two findings clearer.*

Reply: We apologize for the confusion. As stated in the figure legend, stars always indicate differences compared to the vehicle group. The figure is now modified as requested by the reviewer. We hope this alteration improves the understanding of the figure.

4. *It appears that you have a typo regarding the scale for the y-axis in Fig. 1d. Should this be scaled up by a factor of 10?*

Reply: Indeed, the scale is correct: attacks are extremely fast lasting only a few seconds. Therefore, they represent a very low percentage of time displayed in the FIT, which lasts around 10 min. Hopefully, this answer clarifies the reviewer's question.

5. *Also regarding Figure 1, specifically for the data presented throughout panels b-f: are the data presented from FIT4? Is it lumped from FIT1-4? In either case, this should be made clear. Additionally, the FIT 1, 2, and 3 data from Fig 1 should be separated and included as a supplement, so that consistency across FIT test days can be assessed by the reader, particularly because the FIT 1 data from the IS group is compared to FIT 4 data from the IST group. Furthermore, throughout the manuscript, it is unclear which FIT data is being presented. For example, in Fig 3b, is this the cumulative data of FIT 4 + FIT 6? If so, why does the IST group have two FIT tests combined (4 and 6), while the GH group has only one FIT test (4)? If that is the case, the design of these experiments is mis-balanced.*

Reply: Starting with Figure 1, the data presented for the IST rats represent only FIT 4 (not a mean of FIT 1-4), this information is now included in the figure legend. During the training sessions (FIT 1,2,3), we always live scored the behavior, and following the excellent suggestion given by the reviewer, these aggression data (attacks, attack latencies, total aggression) are now displayed in **Supplementary Table 6 and Supplementary figure 7**.

Regarding Figure 3b for the IST group, we present pooled data from FIT 4 and 6 as the reviewer assumed, this information is now included in the figure legend. As mentioned in the methods **(page 15, lines 17-22)** we typically performed within-subject designs for selected pharmacological experiments (e.g. agonist administration) in IST animals in order to reduce the number of animals used in the study.

However, the reviewer is completely right that using this protocol with 2 repeated FIT exposures (FIT4 and FIT6) for IST animals we face the issue that for direct comparisons GH animals would have to be exposed to another FIT as well. However, this creates the paradox of a second exposure potentially enhancing GH aggression. Thus, those experiments in GH rats had to be performed in a between-subjects design, i.e. each GH rat was only tested once. In any case, we would like to reinforce here that these were independent experiments conducted separately, so as the reviewer suggested we excluded

the statistics comparing the vehicle groups in this figure and separated the data in two different graphs (**Figure 3b and c**). We hope this answer together with the changes and adds to the manuscript fully address the reviewer's concerns.

6. Regarding the statistics utilized for the data presented in Fig 3b: the figure legends describes that t-tests were performed, yet between-group post-hoc comparisons are presented as if an ANOVA was conducted. An ANOVA analysis seems appropriate. Otherwise, the figure should be separated into two figures, to visually indicate that a t-test was performed within each group.

Reply: We thank the reviewer for this important suggestion. Indeed, as mentioned in the answer for **comment 5** these were independent experiments conducted separately and, therefore, t-tests have been performed. We follow the reviewer's suggestion and separated **Figure 3b** in two distinct graphs (**Figure 3b and c**) and removed the comparison between GH and IST vehicle groups now.

7. There is alternating usage of the acronym SH (Figs. 3a-4a) and "isolated" (Fig 1a) between different figure schematics. This makes the comparison of experimental designs across figures confusing. For readership, please be consistent in this description across the figures.

Reply: We thank the reviewer to point out this inconsistency, which indeed needs a better explanation. Single-housing (SH) was used to differentiate single-housing after surgical procedures, such as cannula implantation and viral delivery, from isolation to induce aggression. However, as this generated confusion we now replaced the abbreviation SH for IS according to the reviewer suggestion.

References

1. Smith, C. J. W. *et al.* Age and sex differences in oxytocin and vasopressin V1a receptor binding densities in the rat brain : focus on the social decision- making network. *Brain Struct. Funct.* **222**, 981–1006 (2016).
2. Veenema, A. H., Bredewold, R. & De Vries, G. J. Vasopressin regulates social recognition in juvenile and adult rats of both sexes, but in sex- and age-specific ways. *Horm. Behav.* **61**, 50–56 (2012).
3. Tan, O. *et al.* Oxytocin and vasopressin inhibit hyper-aggressive behaviour in socially isolated mice. *Neuropharmacology* **15**, (2019).
4. Manning, M. *et al.* Oxytocin and Vasopressin Agonists and Antagonists as Research Tools and Potential Therapeutics Neuroendocrinology. *J. Neuroendocrinol.* **24**, 609–628 (2012).
5. Neumann, I. D., Veenema, A. H. & Beiderbeck, D. I. Aggression and anxiety: social context and neurobiological links. *Front. Behav. Neurosci.* **4**, 1–12 (2010).
6. Miczek, K. A., Maxson, S. C., Fish, E. W. & Faccidomo, S. Aggressive behavioral phenotypes in mice. *Behav. Brain Res.* **125**, 167–181 (2001).
7. Koolhaas, J. M. *et al.* The Resident-intruder Paradigm : A Standardized Test for

- Aggression , Violence and Social Stress. *J. Vis. Exp.* **77**, 1–7 (2013).
8. Silva, A. L., Fry, W. H. D., Sweeney, C. & Trainor, B. C. Effects of photoperiod and experience on aggressive behavior in female California mice. *Behav. Brain Res.* **208**, 528–534 (2010).
 9. Williamson, C. M. *et al.* Social hierarchy position in female mice is associated with plasma corticosterone levels and hypothalamic gene expression. *Sci. Rep.* **9**, 1–14 (2019).
 10. Oliva, A. M. *et al.* Toward a Mouse Neuroethology in the Laboratory Environment. *PLoS One* **5**, 1–7 (2010).
 11. Reidy, D. E., Kearns, M. C. & DeGue, S. Reducing psychopathic violence: A review of the treatment literature. *Aggress. Violent Behav.* **18**, 527–538 (2013).
 12. Reidy, D. E. *et al.* Why psychopathy matters: Implications for public health and violence prevention. *Aggress. Violent Behav.* **24**, 214–225 (2015).
 13. Ortega-Campos, E., García-García, J., De la Fuente-Sánchez, L. & Zaldívar-Basurto, F. Assessing the interactions between strengths and risk factors of recidivism through the structured assessment of violence risk in youth (Savry). *Int. J. Environ. Res. Public Health* **17**, (2020).
 14. Senior, M., Fazel, S. & Tsiachristas, A. The economic impact of violence perpetration in severe mental illness: a retrospective, prevalence-based analysis in England and Wales. *Lancet Public Heal.* **5**, e99–e106 (2020).
 15. Clark, C. R. & Hall, M. D. Emerging Technique autoradiography : recent developments. *Trends Biochem. Sci.* **11**, 195–198 (1986).
 16. Baskin, D. G. & Stahl, W. L. Fundamentals of quantitative autoradiography by computer densitometry for in situ hybridization, with emphasis on 33P. *J. Histochem. Cytochem.* **41**, 1767–1776 (1993).
 17. Bosch, O. J. Maternal aggression in rodents: Brain oxytocin and vasopressin mediate pup defence. *Philos. Trans. R. Soc. B Biol. Sci.* **368**, (2013).
 18. Bosch, O. J., Meddle, S. L., Beiderbeck, D. I., Douglas, A. J. & Neumann, I. D. Brain Oxytocin Correlates with Maternal Aggression : Link to Anxiety. *J. Neurosci.* **25**, 6807–6815 (2005).
 19. Caughey, S. D. *et al.* Changes in the intensity of maternal aggression and central oxytocin and vasopressin V1a receptors across the peripartum period in the rat. *J. Neuroendocrinol.* **23**, 1113–1124 (2011).
 20. Bosch, O. J., Pfo, J., Beiderbeck, D. I., Landgraf, R. & Neumann, I. D. Maternal Behaviour is Associated with Vasopressin Release in the Medial Preoptic Area and Bed Nucleus of the Stria Terminalis in the Rat Neuroendocrinology. 420–429 (2010) doi:10.1111/j.1365-2826.2010.01984.x.
 21. Bosch, O. J. & Neumann, I. D. Brain vasopressin is an important regulator of maternal behavior independent of dams' trait anxiety. *Proc. Natl. Acad. Sci.* **105**, 17139–17144 (2008).

22. Bosch, O. J., Krömer, S. A., Brunton, P. J. & Neumann, I. D. Release of oxytocin in the hypothalamic paraventricular nucleus, but not central amygdala or lateral septum in lactating residents and virgin intruders during maternal defence. *Neuroscience* **124**, 439–448 (2004).
23. Bosch, O. J. & Neumann, I. D. Vasopressin released within the central amygdala promotes maternal aggression. *Eur. J. Neurosci.* **31**, 883–891 (2010).
24. Bayerl, D. S., Klampfl, S. M. & Bosch, O. J. Central V1b receptor antagonism in lactating rats: Impairment of maternal care but not of maternal aggression. *J. Neuroendocrinol.* **26**, 918–926 (2014).
25. Bayerl, D. S., Klampfl, S. M. & Bosch, O. J. More than reproduction: Central gonadotropin-releasing hormone antagonism decreases maternal aggression in lactating rats. *J. Neuroendocrinol.* **31**, 1–8 (2019).
26. Klampfl, S. M., Brunton, P. J., Bayerl, D. S. & Bosch, O. J. Hypoactivation of CRF receptors, predominantly type 2, in the medial-posterior BNST is vital for adequate maternal behavior in lactating rats. *J. Neurosci.* **34**, 9665–9676 (2014).
27. Denson, T. F., O’Dean, S. M., Blake, K. R. & Beames, J. R. Aggression in Women: Behavior, Brain and Hormones. *Front. Behav. Neurosci.* **12**, 1–20 (2018).
28. Campbell, A. Staying alive: Evolution, culture, and women’s intrasexual aggression. *Behav. Brain Sci.* **22**, 203–252 (1999).
29. Freitag, C. M. *et al.* Conduct disorder in adolescent females: current state of research and study design of the FemNAT-CD consortium. *Eur. Child Adolesc. Psychiatry* **9**, 1077–1093 (2018).
30. Hashikawa, K., Hashikawa, Y., Lischinsky, J. & Lin, D. The Neural Mechanisms of Sexually Dimorphic Aggressive Behaviors. *Trends Genet.* **10**, 755–776 (2018).
31. Trainor, B. C., Crean, K. K., Fry, W. H. D. & Sweeney, C. Activation of extracellular signal-regulated kinases in social behavior circuits during resident-intruder aggression tests. *Neuroscience* **165**, 325–336 (2010).
32. Terranova, J. I., Ferris, C. F. & Albers, H. E. Sex differences in the regulation of offensive aggression and dominance by Arginine-vasopressin. *Front. Endocrinol. (Lausanne)*. **8**, 1–12 (2017).
33. Jurek, B. & Neumann, I. D. The oxytocin receptor: From intracellular signaling to behavior. *Physiol. Rev.* **98**, 1805–1908 (2018).
34. Neumann I., Russel J.A., L. R. & Group, N. Oxytocin and vasopressin release within the supraoptic and paraventricular nuclei of pregnant, parturient and lactating rats: a microdialysis study. *Neuroscience* **53**, 65–75 (1993).
35. Menon, R. *et al.* Oxytocin Signaling in the Lateral Septum Prevents Social Fear during Lactation. *Curr. Biol.* **28**, 1–13 (2018).
36. Ackermann, K. *et al.* Relational Aggression in Adolescents with Conduct Disorder: Sex Differences and Behavioral Correlates. *J. Abnorm. Child Psychol.* **47**, 1625–1637 (2019).
37. Smaragdi, A. *et al.* Sex Differences in the Relationship Between Conduct Disorder and

- Cortical Structure in Adolescents. *J. Am. Acad. Child Adolesc. Psychiatry* **56**, 703–712 (2017).
38. Zhu, R. *et al.* Intranasal oxytocin reduces reactive aggression in men but not in women: A computational approach. *Psychoneuroendocrinology* **108**, 172–181 (2019).
 39. Carrillo, M. & Ricci, L. A. The effect of increased serotonergic neurotransmission on aggression: a critical meta-analytical review of preclinical studies. *Psychoneuroendocrinology* **205**, 349–368 (2009).
 40. Dumais, K. M., Alonso, A. G., Immormino, M. A., Bredewold, R. & Veenema, A. H. Involvement of the oxytocin system in the bed nucleus of the stria terminalis in the sex-specific regulation of social recognition. *Psychoneuroendocrinology* **64**, 79–88 (2015).
 41. Lukas, M., Toth, I., Veenema, A. H. & Neumann, I. D. Oxytocin mediates rodent social memory within the lateral septum and the medial amygdala depending on the relevance of the social stimulus: Male juvenile versus female adult conspecifics. *Psychoneuroendocrinology* **38**, 916–926 (2013).
 42. Lukas, M. *et al.* The Neuropeptide Oxytocin Facilitates Pro-Social Behavior and Prevents Social Avoidance in Rats and Mice. *Neuropsychopharmacology* **36**, 2159–2168 (2011).
 43. Zoicas, I., Slattery, D. A. & Neumann, I. D. Brain Oxytocin in Social Fear Conditioning and Its Extinction: Involvement of the Lateral Septum. *Neuropsychopharmacology* **39**, 3027–3035 (2014).
 44. Calcagnoli, F., De Boer, S. F., Althaus, M., Den Boer, J. A. & Koolhaas, J. M. Antiaggressive activity of central oxytocin in male rats. *Psychopharmacology (Berl)*. **229**, 639–651 (2013).
 45. Burkett, J. P. *et al.* Oxytocin-dependent consolation behavior in rodents. *Science (80-.)*. **351**, 375–378 (2016).
 46. Grund, T. *et al.* Chemogenetic activation of oxytocin neurons: Temporal dynamics, hormonal release, and behavioral consequences. *Psychoneuroendocrinology* **106**, 77–84 (2019).
 47. Knobloch, H. S., Grinevich, V. & Dabrowska, J. Evolution of oxytocin pathways in the brain of vertebrates. *Front. Behav. Neurosci.* **8**, 1–13 (2014).
 48. Eliava, M., Melchior, M., Knobloch-bollmann, H. S., Stoop, R. & Charlet, A. Article A New Population of Parvocellular Oxytocin Neurons Controlling Magnocellular Neuron Activity and Inflammatory Pain Processing Article A New Population of Parvocellular Oxytocin Neurons Controlling Magnocellular Neuron Activity and Inflammatory Pai. 1291–1304 (2016) doi:10.1016/j.neuron.2016.01.041.
 49. Tang, Y. *et al.* Social touch promotes interfemale communication via activation of parvocellular oxytocin neurons. *Nat. Neurosci.* **23**, 1125–1137 (2020).
 50. Hasan, M. T. *et al.* A Fear Memory Engram and Its Plasticity in the Hypothalamic Oxytocin System. *Neuron* **103**, 133-146.e8 (2019).
 51. Peters, S., Slattery, D. A., Uschold-Schmidt, N., Reber, S. O. & Neumann, I. D. Dose-dependent effects of chronic central infusion of oxytocin on anxiety, oxytocin receptor binding and stress-related parameters in mice. *Psychoneuroendocrinology* **42**, 225–236

(2014).

Reviewer #1 (Remarks to the Author):

This manuscript has improved and the authors have addressed some, but not all of my concerns. Findings of this study are important and novel, but the authors need to be more careful in what can be concluded and what cannot be concluded from these findings. They continue to struggle with this in the manuscript but also in their rebuttal in which the authors continue to overstate findings or make statements that are false and cannot be made based on their current findings. I also have concerns that the authors are naïve about the chemogenetic and optogenetic techniques and their limitations.

1. Most importantly, this manuscript is still plagued with statements that are incorrect, false and/or overstated. Statements about previous findings as well as conclusions and interpretations of the findings in this study will need to be toned down throughout the whole manuscript. I only provide here some examples.

For example, in the abstract, the authors state that “our data demonstrate that septal release of OXT and AVP affects female aggression by differential regulation of the excitatory-inhibitory balance within LS subnetworks”. The authors do not provide evidence that AVP alters the excitatory-inhibitory balance. The authors also do not provide evidence that the in vitro effects of OXT occur during female aggression. Words like “demonstrate” suggest otherwise. One could say “our data support a model in which....” This difference may sound trivial, but it isn’t and I strongly suggest that the authors take a careful look at all their conclusions and interpretations of the findings and tone down their statements where applicable throughout the manuscript.

Here is another example: “Accordingly, increased excitability of OXT-responsive neurons in the vLS and decreased excitability of AVP-responsive neurons in the dLS were essential to evoke female aggression.” Here, the authors make the assumption that OXT-responsive neurons are excited and that AVP-responsive neurons are inhibited, but they don’t provide proof of that. It can be that OXT-responsive neurons express an OXT receptor that is coupled to a Gi protein and mediates an inhibitory response upon binding of OXT to that neuron. It is also unclear what it means to say OXT-responsive and AVP-responsive neurons.

A third example, page 5, line 3 and the caption of Figure 3 read “OXT promotes, whereas AVP reduces female aggression.” However, Fig 3C shows that exogenous OXT decreases aggression on IST females. Yes, the authors indeed also show that exogenous OXT increases aggression in GH females. But one cannot dismiss the effect in IST females. Hence, these statements are misleading and should be rephrased.

A fourth example, page 6, line 6-7, “the pro-aggressive effect of endogenous OXT is mediated via OXTRs”. This statement cannot be made. Likewise, the authors do not show in Figs 2 and 3 “the pro-aggressive effects of endogenous OXT (Figures 2 and 3).” Although the authors have shown that blocking OXTRs decreases aggression in IST females, they have not demonstrated that this is because of endogenous OXT. Evidently, we all assume that this is the case, but it is unknown what endogenous substance activates OXTRs. This could potentially be AVP. Thus, this statement needs to be toned down. My suggestion would be to say something along the lines of “these results demonstrate that the pro-aggressive effect is mediated via activation of OXTRs (most likely through endogenous OXT), whereas the anti-aggressive effect is mediated via activation of V1aRs (via exogenous OXT and most likely through endogenous AVP).” This is a crucial point because this

comes back in the rest of the manuscript and therefore, this also needs to be toned down accordingly.

Example 6, I have objection to the statement “since icv synthetic OXT decreased aggression via activation of V1aRs” (page 6, line 1) and similar statements made elsewhere in the manuscript. The observation that co-administration of a V1aR antagonist restores the effect of OXT on aggression in IST females is not necessarily proof that OXT binds to V1aR. It would be important to show that the dose of V1aR antagonist used does not have an effect on its own on aggression. As is, it could be that the V1aR antagonist by itself increases aggression and thereby eliminating the decreasing effect of OXT on aggression. Indeed, the authors at least show the likelihood of this by demonstrating in Fig 5E that the V1aR antagonists applied to the dLS increases aggression. Notably, the OXT receptor couples to multiple G proteins and can form dimers, leading in both cases to different physiological responses, and it is likely that the concentration of OXT will play a role here (See e.g., research by Busnelli and Chini). That is, a higher concentration of OXT might induce coupling of the OXT receptor to different G proteins and/or dimerization, which could explain the decrease in aggression in IST females.

2. The authors indicate that “manipulation of OXT neuron activity via their infection with this virus expressing either DREADD or ChR2 results in changes of endogenous (i.e. intracerebral) OXT release including from distant oxytocinergic axons in the lateral septum³⁵. I am not aware of any article demonstrating this. And indeed, looking up reference #35 (Menon, R. et al. Oxytocin Signaling in the Lateral Septum Prevents Social Fear during Lactation. *Curr. Biol.* 28, 1–13 (2018), I couldn’t find any data showing OXT release in the LS or anywhere else in the brain upon DREADD or opto manipulation. In fact, Menon et al inhibited OXT-synthesizing neurons rather than stimulated these. Thus, the statement by the authors is false. I therefore strongly object to any reference and statement in the manuscript suggesting that DREADDs or optogenetic manipulation change the endogenous OXT release in the lateral septum. For example, the authors state on page 6, line 24 and page 7 line 5, “thereby releasing OXT in the vLS”. This needs to be removed or at least toned down considerably. Likewise, the authors state on page 10 line 2-3 that “chemogenetic and optogenetic stimulation of OXT release within the brain, and specifically within the vLS” should be changed to “chemogenetic stimulation of OXT neurons in the PVN and SON and optogenetic stimulation of OXT axons in the vLS” Please check for similar incorrect statements in other parts of the manuscript as well.

I also object to the suggestion that these chemo- and optogenetic manipulations ONLY affect OXT release. OXT-synthesizing neurons most likely co-release other neurotransmitters and peptides and if DREADDs or optogenetic manipulations of OXT-synthesizing neurons are effective in altering the activity of these neurons, then most likely the release of multiple chemicals other than OXT will be altered. This seems evident, but I get the feeling that this is not evident to the authors.

Related to this, I’m also puzzled by the statement of the authors that “Therefore, the virally-delivered OXT promoter is not active in non-OXT cells due to the lack of transcriptional factors that are essential for OXT promoter transcription”. Yes, in theory, but certainly no in practice. That’s why the authors report much less than 100% specificity and mCherry expression outside of the PVN and SON in non-OXT-expressing neurons.

Finally, the authors talk in their rebuttal and manuscript about “empty rAAV1/2 OXTpr-mCherry”. I can guess what the authors mean by this, but I’ve never heard of “empty” in this context. Please use

the appropriate scientific terms.

3. With the individual data points, it becomes clear that the data in Fig 1d are not normally distributed with two outliers in the IS group. Please check for this and change the statistical analysis accordingly.

4. The glucocorticoid data seem unnecessary to include. There doesn't seem to be a group difference and only a correlation. This data is also not further discussed.

5. It would be helpful to indicate in the results at what time point after FIT exposure, CSF and blood samples were taken. Because these measures were immediately taken after FIT exposure it may not be surprising

6. The authors argue that they performed experiments with either GH or IST based on OXT release patterns (see pages 1 and 2 in the rebuttal and newly added text on page 5, lines 3-7). I understand this logic, but then it comes as a surprise that they deviate from this by determining the effects of exogenous OXT on aggression in IST females (page 5- lines 8-19). This is then really confusing. If the authors truly wish to follow the 3R rule as they indicated in their rebuttal, what was the rationale to perform this experiment in IST females?

7. Page 5, line 26-27, it is hard to believe that 64.37% of PVN cells and 75.1% of SON cells were positive for both OXT and mCherry. It is unclear whether the authors mean 64.37% of all PVN cells and 75.1% of all SON cells or 64.37% of PVN OXT-positive cells and 75.1% of SON OXT-positive cells or 64.37% of PVN mCherry-positive cells and 75.1% of SON mCherry-positive cells.

Page 5, line 29-30, "a few mCherry-positive cells outside the PVN (10.8%) and SON (9.3%) devoid OXT immunosignals". This wording is confusing. What do the authors mean by the 10.8% and 9.3%. Is this based on total mCherry-positive cells in the PVN and SON areas? Same two issues exist for the opto experiments

8. I happened to randomly check one reference and that one is incorrectly cited in the reference list (Carillo & Ricci, 2009: wrong journal, more authors). Please double check all references.

9. The authors indicate in their rebuttal on page 3 that the binding of OXT to V1aRs is unspecific. What do the authors mean by unspecific and what proof do they have that this is unspecific?

10. The caption of Fig 4b indicates that "IST, but not GH females showed an increased rise in OXT release in the vLS during the FIT indicated by increased OXT content in 30-min microdialysates". First, please indicate what the percentage means. Second, the authors cannot say that there was an "increased OXT content" because OXT release data is not shown as absolute concentrations in microdialysates but rather as percentage. Please adjust accordingly.

11. Please indicate in the captions of Figs 3 and 4 what "control" means.

12. Please refer to virgin female aggression throughout the manuscript rather than female aggression, because the latter can be confused with maternal aggression.

13. What do the authors mean by "blunted" in "IST females showed blunted AVP release within the

dLS during an aggressive encounter” (page 10 line 6-7)? There was no change in % AVP release in these females, and thus using the term “blunted” is incorrect and misleading.

14. Page 10, line 27-28. “sex-dimorphic effects of OXT on social behaviors”. This is a bold statement that should be rephrased and toned down. It suggests that OXT always has sex-dimorphic effects on all social behaviors, which is not true. Please also consider using sex-specific effects as opposed to sex-dimorphic which literally means that there are two distinct forms of how OXT is acting and that too is often not true.

15. The purpose of the following paragraph is not very well developed and is missing detail about the context of the current finding i.e., in what animal and under what conditions: “Furthermore, we found a mismatch between OXT release and OXTR binding in the LS, i.e. high local OXT release and reduced OXTR binding. Although this might seem confusing at first sight, similar mismatches between release and receptor binding patterns have been described before in association with increased OXT availability.”

Reviewer #2 (Remarks to the Author):

This is an outstanding manuscript that is likely to make a strong impact on the field. The authors have responded to all of my questions adequately and the revised manuscript is now substantially improved over the original.

Reviewer #3 (Remarks to the Author):

The authors have adequately addressed all of my earlier concerns. The revised paper is very strong and interesting, I enjoyed reading it.

EDITORIAL COMMENTS

Reviewer #1 (Remarks to the Author):

*This manuscript has improved and the authors have addressed some, but not all of my concerns.*
*Findings of this study are important and novel, but the authors need to be more careful in what can be*
*concluded and what cannot be concluded from these findings. They continue to struggle with this in*
*the manuscript but also in their rebuttal in which the authors continue to overstate findings or make*
*statements that are false and cannot be made based on their current findings. I also have concerns that*
*the authors are naïve about the chemogenetic and optogenetic techniques and their limitations.*

**Reply:** First of all, we would like to thank the reviewer for recognizing the importance and the novelty
of our study. Regarding the opto- and chemogenetic techniques, we apologize for not offering
sufficient explanation before. In light of that, we would like to emphasize that Valery Grinevich (co-
author) and his colleagues introduced those approaches into the oxytocin field¹⁻⁴. Also, we have
successfully used chemogenetics in our previous studies⁵⁻⁷. We very much hope that the new version
of the manuscript and rebuttal letter can fully address all the concerns raised by the reviewer.

*1. Most importantly, this manuscript is still plagued with statements that are incorrect, false and/or*
*overstated. Statements about previous findings as well as conclusions and interpretations of the*
*findings in this study will need to be toned down throughout the whole manuscript. I only provide here*
*some examples.*

*For example, in the abstract, the authors state that “our data demonstrate that septal release of OXT*
*and AVP affects female aggression by differential regulation of the excitatory-inhibitory balance within*
*LS subnetworks”. The authors do not provide evidence that AVP alters the excitatory-inhibitory balance.*
*The authors also do not provide evidence that the in vitro effects of OXT occur during female aggression.*
*Words like “demonstrate” suggest otherwise. One could say “our data support a model in which....”*
*This difference may sound trivial, but it isn’t and I strongly suggest that the authors take a careful look*
*at all their conclusions and interpretations of the findings and tone down their statements where*
*applicable throughout the manuscript.*

**Reply:** Thank you for promoting a more careful interpretation of our data, which we appreciate.
Accordingly, we have re-phrased the misleading statement in the abstract: **“Overall, our data suggest**
**a model where septal release of OXT and AVP affects female aggression by modulating the inhibitory**
**tone within LS subnetworks”** (Page 1, lines 28-30). We hope this alteration fully meets the reviewer’s
expectations. However, regarding AVP we kindly ask the reviewer to look into **Supplementary figure**
**6** and section **“Spontaneous activity in neurons in the dLS and vLS is differentially modulated by**
**activation of OXTRs and V1aRs”**, where we show that AVP increases sIPSCs onto dorsal neurons in the
LS, i.e., that AVP changes the inhibitory tone in this network. Moreover, we would like to reinforce that
our *in vivo* data demonstrates that the display of aggression decreases the activity of GABAergic
neurons in the dLS, but increases the activity of GABAergic neurons in the vLS (**Figure 7**), thus
demonstrating a differential contribution to the inhibitory-excitatory balance in this network. This was
further confirmed by pharmacological inhibition of those LS sub-regions (dLS versus vLS) using
muscimol. Additionally, we could show using pharmacology and microdialysis that both increased OXT
release in the vLS, but decreased AVP release in the dLS underly high levels of female aggression in
virgin Wistar rats (**Figures 4 and 5**). Although these data cannot be causally linked to neuronal activity,
as we do not know whether the release alterations lead to the differences in neuronal activity, this link
is strongly supported by our *in vitro* electrophysiology data, which show that OXT (**Figure 6**) and AVP
(**Supplementary Figure 6**) differently modulate sIPSCs within the LS network.

*Here is another example: “Accordingly, increased excitability of OXT-responsive neurons in the vLS and*
*decreased excitability of AVP-responsive neurons in the dLS were essential to evoke female aggression.”*
*Here, the authors make the assumption that OXT-responsive neurons are excited and that AVP-*
*responsive neurons are inhibited, but they don’t provide proof of that. It can be that OXT-responsive*
*neurons express an OXT receptor that is coupled to a Gi protein and mediates an inhibitory response*
*upon binding of OXT to that neuron. It is also unclear what it means to say OXT-responsive and AVP-*
*responsive neurons.*

**Reply:** We thank the reviewer for pointing out the unclear statement, which we have re-phrased (Page
1, lines 25-27). Indeed, Gi-coupled OXTR signaling has been reported *in vitro*, however, only a few
studies have shown that Gi-coupled receptors have effects *in vivo*. In fact, there is only one paper
studying the effect of OXTGi-coupled receptors *in vivo* in the context of social behaviors⁸. In this paper
atosiban, which blocks the Gq pathway but activates the Gi pathway, decreased social approach.
However, as the compound acted in both signaling cascades, it remained unclear, whether the
observed effects are specific for either Gq or Gi pathway. Moreover, we are unaware of any study that
has shown the involvement of Gi-coupled OXTRs in aggression. Importantly, in the patch-clamp
experiments, the addition of bicuculline (GABA-A antagonist) to the bath (Page 8 lines 16-32, Figure 6,
and Supplementary Figure 6) showed that the inhibitory effects of OXT are mediated via GABAergic
tonic inhibition probably via exciting GABAergic OXTR-expressing neurons in the vLS, this makes the
participation of Gi-coupled OXT receptors rather unlikely. Nevertheless, as our statement on OXT-
responsive and AVP-responsive neurons was indeed confusing we re-phrased (Page 1, lines 25-27)
according to the reviewer’s suggestion. Now we state that increased “**activity of putative OXT-**
**receptor-positive neurons in the vLS, and decreased activity of putative AVP-receptor-positive**
**neurons in the dLS, were essential to evoke female aggression**”, as shown by our pERK experiment
(Figure 7).

*A third example, page 5, line 3 and the caption of Figure 3 read “OXT promotes, whereas AVP reduces*
*female aggression.” However, Fig 3C shows that exogenous OXT decreases aggression on IST females.*
*Yes, the authors indeed also show that exogenous OXT increases aggression in GH females. But one*
*cannot dismiss the effect in IST females. Hence, these statements are misleading and should be*
*rephrased.*

**Reply:** We thank the reviewer for pointing out this inconsistency in the manuscript. We rephrased the
captions and made them more specific (Page 5, line 3 and Figure 3) according to our conclusions for
this section (Page 6, lines 4-6).

*A fourth example, page 6, line 6-7, “the pro-aggressive effect of endogenous OXT is mediated via*
*OXTRs”. This statement cannot be made. Likewise, the authors do not show in Figs 2 and 3 “the pro-*
*aggressive effects of endogenous OXT (Figures 2 and 3).” Although the authors have shown that*
*blocking OXTRs decreases aggression in IST females, they have not demonstrated that this is because*
*of endogenous OXT. Evidently, we all assume that this is the case, but it is unknown what endogenous*
*substance activates OXTRs. This could potentially be AVP. Thus, this statement needs to be toned down.*
*My suggestion would be to say something along the lines of “these results demonstrate that the pro-*
*aggressive effect is mediated via activation of OXTRs (most likely through endogenous OXT), whereas*
*the anti-aggressive effect is mediated via activation of V1aRs (via exogenous OXT and most likely*
*through endogenous AVP).” This is a crucial point because this comes back in the rest of the*
*manuscript and therefore, this also needs to be toned down accordingly.*

**Reply:** We thank the reviewer for the valuable suggestion, which is now incorporated into the
manuscript (Page 6 lines 8-11)

*Example 6, I have objection to the statement “since icv synthetic OXT decreased aggression via*
*activation of V1aRs” (page 6, line 1) and similar statements made elsewhere in the manuscript. The*
*observation that co-administration of a V1aR antagonist restores the effect of OXT on aggression in IST*
*females is not necessarily proof that OXT binds to V1aR. It would be important to show that the dose*
*of V1aR antagonist used does not have an effect on its own on aggression. As is, it could be that the*
*V1aR antagonist by itself increases aggression and thereby eliminating the decreasing effect of OXT on*
*aggression. Indeed, the authors at least show the likelihood of this by demonstrating in Fig 5E that the*
*V1aR antagonists applied to the dLS increases aggression. Notably, the OXT receptor couples to*
*multiple G proteins and can form dimers, leading in both cases to different physiological responses, and*
*it is likely that the concentration of OXT will play a role here (See e.g., research by Busnelli and Chini).*
*That is, a higher concentration of OXT might induce coupling of the OXT receptor to different G proteins*
*and/or dimerization, which could explain the decrease in aggression in IST females.*

**Reply:** First of all, we would like to thank the reviewer for this important suggestion to improve the
clarity of our statements in the manuscript. Accordingly, as suggested by the reviewer, we have
included new data showing that the administration of V1aR-A (750ng/5µl) on its own (i.c.v.) does not
affect the levels of aggression displayed by highly aggressive IST females **(Figure 3d)**. This should
strongly support our hypothesis that the effects of **synthetic OXT** are mediated via V1aRs. We hope
this answer and the new data clarify the reviewer’s concerns. Moreover, regarding the Gi OXTR-
coupling, we kindly ask the reviewer to see the answer to the second part of comment 1. Indeed, OXTR
can couple to Gi protein *in vitro* as it has been nicely shown by Marta Busnelli and Bice Chini⁹. However,
we are unaware of any study, which has addressed the role of OXT Gi-coupled receptors on aggression.
As mentioned before, from the best of our knowledge, there is only one paper that has assessed the
effect of OXTGi-coupled receptors *in vivo* in the context of social behaviors⁸, which rather unclear
results in terms of a predominant Gi effect. Regarding our i.c.v OXT dose, we need to emphasize that
we used a comparably very low dose (50ng/5µl). For example, other studies performed to assess the
effects of i.c.v. OXT on male aggression used doses 5, 20, and even 80 times higher than the one we
used, i.e. 250ng, 1000ng, and 4000ng, respectively¹⁰. Finally, our electrophysiological data also suggest
that the septal effects of OXTR activation are not mediated via Gi-coupled OXTRs, as the application of
bicuculline (GABA-A antagonist) completely abolished sIPSCs, demonstrating that those inhibitory
currents have GABAergic origin and are unlikely to come directly from OXTR activation. Overall, the
addition of a new set of data in the manuscript showing that i.c.v. administration of V1aR-A on its own
does not affect the levels of aggression displayed by highly aggressive IST females **(Figure 3d)** should
**exclude any further uncertainties in this respect and** support our hypothesis that the effects of
**synthetic OXT** are mediated via V1aRs.

*2. The authors indicate that “manipulation of OXT neuron activity via their infection with this virus*
*expressing either DREADD or Chr2 results in changes of endogenous (i.e. intracerebral) OXT release*
*including from distant oxytocinergic axons in the lateral septum. I am not aware of any article*
*demonstrating this. And indeed, looking up reference #35 (Menon, R. et al. Oxytocin Signaling in the*
*Lateral Septum Prevents Social Fear during Lactation. Curr. Biol. 28, 1–13 (2018), I couldn’t find any*
*data showing OXT release in the LS or anywhere else in the brain upon DREADD or opto manipulation.*
*In fact, Menon et al inhibited OXT-synthesizing neurons rather than stimulated these. Thus, the*
*statement by the authors is false. I therefore strongly object to any reference and statement in the*
*manuscript suggesting that DREADDS or optogenetic manipulation change the endogenous OXT*
*release in the lateral septum. For example, the authors state on page 6, line 24 and page 7 line 5,*
*“thereby releasing OXT in the vLS”. This needs to be removed or at least toned down considerably.*
*Likewise, the authors state on page 10 line 2-3 that “chemogenetic and optogenetic stimulation of OXT*
*release within the brain, and specifically within the vLS” should be changed to “chemogenetic*

*stimulation of OXT neurons in the PVN and SON and optogenetic stimulation of OXT axons in the vLS*
*Please check for similar incorrect statements in other parts of the manuscript as well.*

**Reply:** We apologize for not making this clearer before. Regarding, the chemogenetic stimulation of
OXT release we ask the reviewer to see again our answer to **comment 14 on page 10 of the first**
**rebuttal letter**. As we explained there, the DREADD approach was used to trigger the intracerebral
(meaning non-region specific) release of OXT, mimicking the i.c.v infusion of OXT in GH rats. As
mentioned, the chemogenetically induced OXT release has been shown before by our group. Briefly,
Grund et. al. (Psychoneuroendocrinology,2019) have demonstrated that infection of OXT neurons in
the PVN, SON, and accessory nuclei with OXTprhM3Dq:mCherry (the same rAVV used here) and i.p.
administration of CNO leads to OXT release within the PVN (measured using *in vivo* microdialysis) and
OXT secretion from OXT neurohypophysial terminals into blood⁶. This triggered OXT release was found
to have an anxiolytic effect as reversed defeat-induced social avoidance, a behavior known to be OXT-
dependent¹¹. Apart from that, our recent paper published in Nature Neuroscience has shown that
administration of OXTR-A (i.c.v.) abolishes the increased social interaction induced by DREADD
stimulation of parvocellular OXT neurons in the PVN, again corroborating that chemogenetic
stimulation of OXT neurons triggers intracerebral OXT release³. However, regarding the local release
of OXT in OXT target regions, such as the septum, the reviewer is correct, as neither we nor others
have shown that optogenetic stimulation of OXT terminals results in OXT release within the LS.
Nevertheless, Knobloch et. al. (Neuron, 2012) have shown that *in vitro* stimulation of OXT terminals
within the central amygdala increases IPSCs and action potentials in this region, and those effects were
completely abolished by treatment with OXTR-A. These findings have been recently reproduced by
Hasan et al. (Neuron, 2019)¹. As a further indication of optogenetic-induced OXT release locally, Oetl,
et. al. (Neuron, 2016) have demonstrated that optogenetic stimulation of PVN-OXT axons within the
accessory olfactory nucleus (AON) increase post-synaptic current (PSCs) frequencies in this region,
those effects were again fully blocked by administration of OXTR-A to the bath¹². Altogether, these
findings strongly indicate that optogenetic stimulation of OXT axon terminals leads to OXT release^{12,13}.
In addition, we would like to mention that the measurement of actual amounts of OXT released after
blue light illumination by a limited number of axons within a distant brain area is technically impossible
due to methodological limitation (absolute recovery by the microdialysis probe) despite we have the
RIA, which has one of the highest sensitivities worldwide^{14,15}.

Finally, we would like to express that we are aware of the limitations of those techniques, and although
other papers have claimed that stimulation of OXT terminals triggers local OXT release, shown via
electrophysiology and pharmacology, we decided to tone down the claims regarding optogenetic-
stimulated vLS OXT release in the manuscript (**Please see Page 6, 25-27; Page 7, 6-9; Page 10, 5-8**).

*I also object to the suggestion that these chemo- and optogenetic manipulations ONLY affect OXT*
*release. OXT-synthesizing neurons most likely co-release other neurotransmitters and peptides and if*
*DREADDs or optogenetic manipulations of OXT-synthesizing neurons are effective in altering the*
*activity of these neurons, then most likely the release of multiple chemicals other than OXT will be*
*altered. This seems evident, but I get the feeling that this is not evident to the authors.*

**Reply:** Indeed it has been reported that OXT neurons express and may release other neurotransmitters
such as glutamate. However, high-frequency stimulations are thought to rather induce peptidergic
release instead of classical neurotransmitter release^{13,16}. Referring again to Knobloch et. al. (2012) and
the Oetl, et. al. (2016), as all the effects on IPSCs, action potential frequency, and PSCs, respectively,
triggered by optogenetic stimulation of OXT axon terminals within the target regions were abolished
by OXTR-A, and based on the fact that we used the same rAAV and blue-light stimulation pattern we
are strongly convinced that our effects are rather OXT-mediated. Also, it is important to highlight that
none of our conclusions relies solely on the chemo- or optogenetic experiments; they rather provide

further and complementary evidence and confirm the results obtained using neuropharmacology, *in*
*vivo* microdialysis, and CSF extraction, which clearly point towards the involvement of OXT on virgin
female aggression.

*Related to this, I'm also puzzled by the statement of the authors that "Therefore, the virally-delivered*
*OXT promoter is not active in non-OXT cells due to the lack of transcriptional factors that are essential*
*for OXT promoter transcription". Yes, in theory, but certainly no in practice. That's why the authors*
*report much less than 100% specificity and mCherry expression outside of the PVN and SON in non-*
*OXT-expressing neurons.*

**Reply:** We thank the reviewer for his/her comment, as we also have debated this finding extensively.
As reported by Knobloch et. al. (2012) the present virus is known to show a high degree of cell
specificity and efficiency especially under physiological challenges, such as lactation, when OXT
expression is drastically increased. However, we indeed did not show a 100% cell specificity what could
be a function of several reasons, such as the titer of virus infused, its volume, expression time, or
applied coordinates as well as immunostaining procedure *per se* and antibody's dilution.

It is worth mentioning that because a cell in the PVN or SON is mCherry, but not OXT positive does not
necessarily mean that this is a non-OXTergic neuron. For instance, a small number of cells could be
quiescent and not be producing OXT, or being depleted, at a moment of killing but still carries all the
machinery to synthesize and process OXT, including an OXT promoter. Thus, hypothetically this cell
would be infected by the virus, but would not be labeled by an antibody.

We would like to emphasize that we openly report some degree of non-specific ChR2 and DREADD
expression occurred in our hands. Indeed, in theory, the virus should be expressed only in those cells,
which carry endogenous OXT promoter. However, in practice, short promoter sequences (2.6 kb in the
case of mouse OXT promoter used here) requires a lot of adjustment and experience to achieve higher
or full cell-type specificity (such as using dilution series, range of volumes, and optimization of
expression time). The same holds true even for AAVs carrying floxed genes, which are supposed to not
present Cre-independent expression at all. Finally, as the reviewer is probably aware of, even
transgenic animal models, including BAC and knock-in animals, very often do not reach ideal transgene
expression in desired cell types.

Anyway, we reinforce here what was said in the comment above that none of our findings lie
exclusively on the DREADD or optogenetic experiments, they are rather complementary tools that
strengthen the behavioral pharmacology and release data. We again thank the reviewer for raising this
point and we hope that our answers and changes could fully address his/her concerns.

*Finally, the authors talk in their rebuttal and manuscript about "empty rAAV1/2 OXTpr-mCherry". I can*
*guess what the authors mean by this, but I've never heard of "empty" in this context. Please use the*
*appropriate scientific terms.*

**Reply:** This information is now corrected throughout the manuscript, we thank the reviewer for the
careful reading and important suggestion.

*3. With the individual data points, it becomes clear that the data in Fig 1d are not normally distributed*
*with two outliers in the IS group. Please check for this and change the statistical analysis accordingly.*

**Reply:** Indeed the data is not normally distributed that is why we have used the Kruskal-Wallis test
followed by Dunn's as referred to in the **legend of Figure 1.**

*4. The glucocorticoid data seem unnecessary to include. There doesn't seem to be a group difference*
*and only a correlation. This data is also not further discussed.*

**Reply:** We thank the reviewer for this valuable suggestion, as we have already discussed this issue
intensively. However, this is the first paper on this novel animal model on virgin female aggression,
and, thus, we believe that besides dissecting the effects of OXT and AVP on aggression we also aim to
show its construct validity by presenting the CORT data. The assessment of plasma corticosterone
levels is especially relevant because low levels of glucocorticoids in plasma have been consistently
associated with high levels of aggression in humans and animals as we mention on **(Pages 4 and 5, lines**
**30 -32 and 1-2, respectively)**. We hope the reviewer can agree to this decision.

*5. It would be helpful to indicate in the results at what time point after FIT exposure, CSF and blood*
*samples were taken. Because these measures were immediately taken after FIT exposure it may not be*
*surprising*

**Reply:** Indeed, samples were collected immediately after the FIT. We agree with the reviewer's
opinion that having this information already in the results would increase the clarity of the manuscript.
However, due to the word count limitation, we kindly invite the reviewer to see our methods section
where this information has been previously included **(Page 15, lines 9-13)**.

*6. The authors argue that they performed experiments with either GH or IST based on OXT release*
*patterns (see pages 1 and 2 in the rebuttal and newly added text on page 5, lines 3-7). I understand this*
*logic, but then it comes as a surprise that they deviate from this by determining the effects of exogenous*
*OXT on aggression in IST females (page 5- lines 8-19). This is then really confusing. If the authors truly*
*wish to follow the 3R rule as they indicated in their rebuttal, what was the rationale to perform this*
*experiment in IST females?*

**Reply:** We apologize for not making this clearer before. Here, we would like to reinforce our
commitment to the 3R rule. In fact, as explained in the first rebuttal (pages 1 and 2) we used the
patterns of release to determine the groups of further experiments. Therefore, we manipulated the
neuroptidergic systems in order to complement the effects seen in the release, for example
increasing OXT availability in low aggressive GH rats and blocking the OXTRs in highly aggressive IST
rats. Also as pointed out in the first rebuttal, the low level of aggression seen in GH rats might not be
further reduced, e.g. by inhibition of the OXT system, or application of AVP (**floor effect**), which
constitutes a limitation for bidirectional manipulations in both housing conditions. Nevertheless, as
the reviewer mentioned despite this logic we performed an experiment where we infused *synthetic*
OXT i.c.v. in highly aggressive IST rats. The rationale of this experiment was to make a parallel with
previous studies, which have shown anti-aggressive effects of OXT in male rats^{10,17,18}. Again, as this is
the first paper dissecting the role of OXT on virgin female aggression, we think it is important to
compare our findings with existing data, especially with this highly cited study in males in order to
properly validate the female model. This information is now included in the manuscript **(Page 5, lines**
**7-8)**. We hope this reply together with the new sentence added to the manuscript meets the reviewer's
concerns.

*7. Page 5, line 26-27, it is hard to believe that 64.37% of PVN cells and 75.1% of SON cells were positive*
*for both OXT and mCherry. It is unclear whether the authors mean 64.37% of all PVN cells and 75.1%*
*of all SON cells or 64.37% of PVN OXT-positive cells and 75.1% of SON OXT-positive cells or 64.37% of*
*PVN mCherry-positive cells and 75.1% of SON mCherry-positive cells.*

**Reply:** We apologize for not making this statement clearer. According to the reviewer's suggestion,
we now write: "The virus showed a high degree of cell-type efficiency as **64.37% of OXT cells** in the
**PVN** and **75.1% of OXT cells** in the **SON** were positive for **mCherry**, and specificity as **73.9% of mCherry**
**cells** in the **PVN** and **77.9% of mCherry cells in the SON** were positive for **OXT** in accordance with
previous data" **(Page 5, line 27-30)**. Similar changes were made in the optogenetics description **(Page**
**6, line 28-31)**. We hope those changes meet the reviewer's concerns.

*Page 5, line 29-30, “a few mCherry-positive cells outside the PVN (10.8%) and SON (9.3%) devoid OXT*
*immunosignals”. This wording is confusing. What do the authors mean by the 10.8% and 9.3%. Is this*
*based on total mCherry-positive cells in the PVN and SON areas? Same two issues exist for the opto*
*experiments*

**Reply:** As suggested by the reviewer in the first rebuttal this is shown as a percentage of total mCherry
cells. This information is now clearer in the manuscript **(Pages 5 and 7, lines 30-31 and lines 1-3,**
**respectively)** as: **“However, a few mCherry-positive cells outside the PVN (10.8% of total mCherry**
**cells) and SON (9.3% of total mCherry cells) devoid OXT immunosignals” and “The specificity of the**
**virus for targeting OXT neurons has been proven before¹⁶, although in the present experiment some**
**limited number of mCherry-positive cells outside the PVN (10.4% of total mCherry cells) and SON**
**(11.7% of total mCherry cells) devoid OXT immunosignals”.**

*8. I happened to randomly check one reference and that one is incorrectly cited in the reference list*
*(Carillo & Ricci, 2009: wrong journal, more authors). Please double check all references.*

**Reply:** We apologize for having overseen this mistake. All the references have been double-checked
in the current version.

*9. The authors indicate in their rebuttal on page 3 that the binding of OXT to V1aRs is unspecific. What*
*do the authors mean by unspecific and what proof do they have that this is unspecific?*

**Reply:** We apologize for not making that clearer before and have changed this sentence accordingly.
In fact, by using the term *unspecific*, we meant non-selective binding of OXT to V1aRs, as OXT has a
higher affinity and selectivity for the OXTR over the V1aR^{9,19,20}. Thus, as explained before, in highly-
aggressive IST females, which already show a high level of OXT release **(Figure 2 and 4)** during FIT
exposure, further elevation of OXT levels via infusion with synthetic OXT could evoke non-selective
binding to V1aRs, probably due to the high occupancy of OXTRs by endogenous OXT. Altogether we
assume this data indicates that **endogenous OXT most likely acting via OXTRs promotes aggression** in
virgin female rats, whereas **activation of V1aR receptors** either via **endogenous/synthetic AVP** or
**synthetic OXT decreases aggression**. In any case, this statement is now altered **(Page 6 lines 8-11)** to
match the reviewer's suggestion (please see the reply to **example 4, first reviewer comment**).

*10. The caption of Fig 4b indicates that “IST, but not GH females showed an increased rise in OXT release*
*in the vLS during the FIT indicated by increased OXT content in 30-min microdialysates”. First, please*
*indicate what the percentage means. Second, the authors cannot say that there was an “increased OXT*
*content” because OXT release data is not shown as absolute concentrations in microdialysates but*
*rather a percentage. Please adjust accordingly.*

**Reply:** We apologize for not specifying this better before, and we have accordingly changed the Fig
legends of **Figure 4 and 5**. Microdialysis data is presented here as a percentage of increase from the
baseline (OXT content of the FIT sample/OXT content of Baseline \times 100), this information is now
mentioned in the methods **(Page 16, lines 11-13)**. Thus, as there is no difference between the groups
regarding baseline levels of OXT, as displayed in **Figures 4b and 5b (please see insert)**, an increased
percentage of release ultimately reflects an increased OXT content in microdialysates sampled during
the FIT. We hope this explanation and the changes performed in the manuscript answer the reviewer's
comment.

*11. Please indicate in the captions of Figs 3 and 4 what “control” means.*

**Reply:** Thank you for pointing out the missing information. This information was previously included
in the Methods section **(Page 16, lines 22-33)** as: **“Importantly, for those experiments, we had 3**
**control groups: i) subjects infected with a control rAAV1/2 OXTpr-mCherry into the PVN and SON**

which received CNO (virus control), non-infected rats who received either ii) CNO or iii) saline
infusions (drug control). Since there was no difference among the levels of aggression displayed by
those three groups (not shown) they were pooled together in a single control group depicted in grey
in Figure 3g. In the optogenetic experiments, after optical fiber implantation rats were single-housed
for three days for recovery and to avoid damaging the fiber. Similarly to the microdialysis
experiments, both controls and ChR2 animals were connected to the optogenetic cables two hours
before the experiment to get used to the cables. In this experiment, the FIT lasted 12 minutes. Blue-
light stimulation (30ms pulses of 30Hz delivered for 2min; in analogy to Knobloch et al., 2012) was
delivered at the 2nd and again 8th minute after the beginning of the FIT. Here, controls consisted of
animals infected with a control rAAV1/2 OXTpr-mCherry into the PVN and SON". However, as
requested by the reviewer we now added this extra information on the legends of Figures 3 and 4.

*12. Please refer to virgin female aggression throughout the manuscript rather than female aggression,*
*because the latter can be confused with maternal aggression.*

**Reply:** We would like to reinforce our argument made in the first rebuttal letter (**Minor comment 7,**
**page 8**). In the introduction, we point out that the state of lactation is a unique physiological period,
where several neurobiological and neuroendocrine systems, including neuropeptide systems, are up-
or downregulated. Those adaptations are accompanied by behavioral changes including maternal
aggression²¹⁻²⁴. Additionally, from a translational perspective, using lactating animals to model the
neurobiological mechanisms controlling aggression in females might not be the appropriate approach
as most of the female perpetrators engage in aggression when they are non-lactating²⁵⁻²⁷. From a
semantic point of view, if we consider that aggression between virgin males is called intermale
aggression²⁸⁻³¹ would be logical to name its counterpart in virgin females as female aggression.
However, we understand the reviewer's point of view, therefore we define the term female aggression
more clear in our introduction to distinguish it from maternal aggression (**Page 2, lines 17-18**). We
hope this answer and changes fully address the reviewer's concerns.

*13. What do the authors mean by "blunted" in "IST females showed blunted AVP release within the dLS*
*during an aggressive encounter" (page 10 line 6-7)? There was no change in % AVP release in these*
*females, and thus using the term "blunted" is incorrect and misleading.*

**Reply:** Indeed, the reviewer is correct that the term blunted is not precise enough. In order to make
the sentence clearer we re-phrased as **"Highly aggressive IST females showed unchanged AVP release**
**within the dLS during an aggressive encounter"** (**page 10 line 9-10**). We hope this change meets the
reviewer's expectations.

*14. Page 10, line 27-28. "sex-dimorphic effects of OXT on social behaviors". This is a bold statement*
*that should be rephrased and toned down. It suggests that OXT always has sex-dimorphic effects on all*
*social behaviors, which is not true. Please also consider using sex-specific effects as opposed to sex-*
*dimorphic which literally means that there are two distinct forms of how OXT is acting and that too is*
*often not true.*

**Reply:** We thank the reviewer for his/her valuable suggestion. We changed the mentioned statement
accordingly to match the reviewer's suggestion (**Page 10, lines 30-32**).

*15. The purpose of the following paragraph is not very well developed and is missing detail about the*
*context of the current finding i.e., in what animal and under what conditions: "Furthermore, we found*
*a mismatch between OXT release and OXTR binding in the LS, i.e. high local OXT release and reduced*
*OXTR binding. Although this might seem confusing at first sight, similar mismatches between release*
*and receptor binding patterns have been described before in association with increased OXT*
*availability."*

**Reply:** This paragraph was added as a request of the third reviewer as the reviewer may see in the first
rebuttal letter (**Minor comment 2, page 14 reply to reviewer #3**) or below. In any case, we added the
missing information to this paragraph in the discussion (**Page 11, lines 16-17**) please see:
**“Furthermore, we found a mismatch between OXT release and OXTR binding in the LS, i.e. high local**
**OXT release and reduced OXTR binding. Although this might seem confusing at first sight, similar**
**mismatches between release and receptor binding patterns have been described before in**
**association with increased OXT availability in male mice after fear-conditioning³⁶”**. We hope these
changes together with the previous answer to reviewer #3 clarifies the reviewer’s concerns.

**“2. The conflicting findings of elevated OXT (Fig 2a), yet reduced OXTR binding (Fig 2c) in the IST**
**group, warrants interpretation and commentary in the discussion”**

**Answer to reviewer #3, first rebuttal letter:** We thank the reviewer for pointing out this interesting
phenomenon. Indeed, this result may appear conflicting at first glance. However, OXT release and
OXTR binding are not linearly correlated, i.e. a high level of local OXT release is not always associated
with increased OXTR binding²³, except in lactation, when suckling-induced OXT release is found at a
stage of general increased OXTR expression and binding.²³ Also, when compared to females, male
Wistar rats exhibit both higher OXTR binding and elevated OXT release in the posterior BNST after
social contact³². In contrast, there are many examples of a mismatch between OXT release and OXTR
binding³³⁻³⁵. For example, in the context of social fear, increased OXTR binding has been found in
parallel to reduced OXT release in the lateral septum in social fear-conditioned mice³⁴. Accordingly,
i.c.v. infusion of *synthetic* OXT, increasing its availability in the brain, results in reduced OXTR binding
in several brain regions including the LS³⁵. Additionally, other factors apart from OXT release are
known to influence OXTR binding²³. Thus, the precise factors regulating OXTR binding in the septum
in association with housing conditions and aggression training are still unknown and need to be further
dissected. As suggested by the reviewer, we have included a statement into the discussion (**page 11**
**lines 6-9**) pointing towards this phenomenon

**Reviewer #2 (Remarks to the Author):**

*This is an outstanding manuscript that is likely to make a strong impact on the field. The authors have*
*responded to all of my questions adequately and the revised manuscript is now substantially improved*
*over the original.*

**Reply:** We would like to thank the reviewer for his/her encouraging comments and for noticing the
novelty and impact of our manuscript.

**Reviewer #3 (Remarks to the Author):**

*The authors have adequately addressed all of my earlier concerns. The revised paper is very strong and*
*interesting, I enjoyed reading it.*

**Reply:** We would like to thank the reviewer for his/her kind words about our study. We are happy to
read that the reviewer enjoyed reading it.

References

- 1. Hasan, M. T. *et al.* A Fear Memory Engram and Its Plasticity in the Hypothalamic Oxytocin
System. *Neuron* **103**, 133–146.e8 (2019).
- 2. Knobloch, H. S. *et al.* Evoked axonal oxytocin release in the central amygdala attenuates fear
response. *Neuron* **73**, 553–566 (2012).
- 3. Tang, Y. *et al.* Social touch promotes interfemale communication via activation of parvocellular
oxytocin neurons. *Nat. Neurosci.* **23**, 1125–1137 (2020).
- 4. Eliava, M. *et al.* A new population of parvocellular oxytocin neurons controlling magnocellular
neuron activity and inflammatory pain processing. *Neuron* **89**, 1291–1304 (2016).
- 5. Menon, R. *et al.* Oxytocin Signaling in the Lateral Septum Prevents Social Fear during Lactation.
*Curr. Biol.* **28**, 1–13 (2018).
- 6. Grund, T. *et al.* Chemogenetic activation of oxytocin neurons : Temporal dynamics , hormonal
release , and behavioral consequences. *Psychoneuroendocrinology* **106**, 77–84 (2019).
- 7. Grund, T. *et al.* Neuropeptide S activates paraventricular oxytocin neurons to induce anxiolysis.
*J. Neurosci.* **37**, 12214–12225 (2017).
- 8. Williams, A. V. *et al.* Social approach and social vigilance are differentially regulated by oxytocin
receptors in the nucleus accumbens. *Neuropsychopharmacology* **45**, 1423–1430 (2020).
- 9. Busnelli, M., Bulgheroni, E., Manning, M., Kleinau, G. & Chini, B. Selective and potent agonists
and antagonists for investigating the role of mouse oxytocin receptors. *J. Pharmacol. Exp. Ther.*
**346**, 318–327 (2013).
- 10. Calcagnoli, F., De Boer, S. F., Althaus, M., Den Boer, J. A. & Koolhaas, J. M. Antiaggressive activity
of central oxytocin in male rats. *Psychopharmacology (Berl)*. **229**, 639–651 (2013).
- 11. Lukas, M. *et al.* The Neuropeptide Oxytocin Facilitates Pro-Social Behavior and Prevents Social
Avoidance in Rats and Mice. *Neuropsychopharmacology* **36**, 2159–2168 (2011).
- 12. Oettl, L. L. *et al.* Oxytocin Enhances Social Recognition by Modulating Cortical Control of Early
Olfactory Processing. *Neuron* **90**, 609–621 (2016).
- 13. Knobloch, H. S. *et al.* Evoked axonal oxytocin release in the central amygdala attenuates fear
response. *Neuron* **73**, 553–566 (2012).
- 14. Grinevich, V. & Neumann, I. D. Brain oxytocin: how puzzle stones from animal studies translate
into psychiatry. *Mol. Psychiatry* 1–15 (2020) doi:10.1038/s41380-020-0802-9.
- 15. Torner, L., Plotsky, P. M., Neumann, I. D. & de Jong, T. R. Forced swimming-induced oxytocin
release into blood and brain: Effects of adrenalectomy and corticosterone treatment.
*Psychoneuroendocrinology* **77**, 165–174 (2017).
- 16. Hökfelt, T. Neuropeptides in perspective: The last ten years. *Neuron* **7**, 867–879 (1991).
- 17. Calcagnoli, F. *et al.* Oxytocin microinjected into the central amygdaloid nuclei exerts anti-
aggressive effects in male rats. *Neuropharmacology* **90**, 74–81 (2015).
- 18. Calcagnoli, F., Kreutzmann, J. C., de Boer, S. F., Althaus, M. & Koolhaas, J. M. Acute and repeated
intranasal oxytocin administration exerts anti-aggressive and pro-affiliative effects in male rats.
*Psychoneuroendocrinology* **51**, 112–121 (2015).
- 19. Manning, M. *et al.* Oxytocin and vasopressin agonists and antagonists as research tools and
potential therapeutics. *J. Neuroendocrinol.* **24**, 609–628 (2012).

- 20. Tan, O. *et al.* Oxytocin and vasopressin inhibit hyper-aggressive behaviour in socially isolated
mice. *Neuropharmacology* **156**, 107573 (2019).
- 21. Bosch, O. J. & Neumann, I. D. Both oxytocin and vasopressin are mediators of maternal care
and aggression in rodents : From central release to sites of action. *Horm. Behav.* **61**, 293–303
(2012).
- 22. Bosch, O. J. Maternal aggression in rodents: Brain oxytocin and vasopressin mediate pup
defence. *Philos. Trans. R. Soc. B Biol. Sci.* **368**, (2013).
- 23. Jurek, B. & Neumann, I. D. The oxytocin receptor: From intracellular signaling to behavior.
*Physiol. Rev.* **98**, 1805–1908 (2018).
- 24. Hashikawa, K., Hashikawa, Y., Lischinsky, J. & Lin, D. The Neural Mechanisms of Sexually
Dimorphic Aggressive Behaviors. *Trends Genet.* **10**, 755–776 (2018).
- 25. Freitag, C. M. *et al.* Conduct disorder in adolescent females: current state of research and study
design of the FemNAT-CD consortium. *Eur. Child Adolesc. Psychiatry* **9**, 1077–1093 (2018).
- 26. Denson, T. F., O’Dean, S. M., Blake, K. R. & Beames, J. R. Aggression in Women: Behavior, Brain
and Hormones. *Front. Behav. Neurosci.* **12**, 1–20 (2018).
- 27. Campbell, A. Staying alive: Evolution, culture, and women’s intrasexual aggression. *Behav. Brain
Sci.* **22**, 203–252 (1999).
- 28. Comai, S., Tau, M. & Gobbi, G. The psychopharmacology of aggressive behavior: A translational
approach: Part 1: Neurobiology. *J. Clin. Psychopharmacol.* **32**, 83–94 (2012).
- 29. Nelson, R. J. & Trainor, B. C. Neural mechanisms of aggression. *Nat. Rev. Neurosci.* **8**, 536–546
(2007).
- 30. De Almeida, R. M. M., Ferrari, P. F., Parmigiani, S. & Miczek, K. A. Escalated aggressive behavior:
Dopamine, serotonin and GABA. *Eur. J. Pharmacol.* **526**, 51–64 (2005).
- 31. Miczek, K. A., Maxson, S. C., Fish, E. W. & Faccidomo, S. Aggressive behavioral phenotypes in
mice. *Behav. Brain Res.* **125**, 167–181 (2001).
- 32. Dumais, K. M., Alonso, A. G., Immormino, M. A., Bredewold, R. & Veenema, A. H. Involvement
of the oxytocin system in the bed nucleus of the stria terminalis in the sex-specific regulation
of social recognition. *Psychoneuroendocrinology* **64**, 79–88 (2015).
- 33. Bosch, O. J., Meddle, S. L., Beiderbeck, D. I., Douglas, A. J. & Neumann, I. D. Brain Oxytocin
Correlates with Maternal Aggression : Link to Anxiety. *J. Neurosci.* **25**, 6807–6815 (2005).
- 34. Zoicas, I., Slattery, D. A. & Neumann, I. D. Brain Oxytocin in Social Fear Conditioning and Its
Extinction : Involvement of the Lateral Septum. *Neuropsychopharmacology* **39**, 3027–3035
(2014).
- 35. Peters, S., Slattery, D. A., Uschold-Schmidt, N., Reber, S. O. & Neumann, I. D. Dose-dependent
effects of chronic central infusion of oxytocin on anxiety, oxytocin receptor binding and stress-
related parameters in mice. *Psychoneuroendocrinology* **42**, 225–236 (2014).

Reviewer #1 (Remarks to the Author):

This manuscript has further improved and I'm pleased to see that the authors made changes in the wording at critical points in the manuscript to tone down and be more careful with their statements and interpretations based on their data.

Allow me to provide some heartfelt advice to the authors that their elaborated replies often went far beyond the comments raised and felt at times condescending and patronizing. Additionally, I believe the authors would agree with me that, irrespective of expertise and past validations, we have to uphold the highest scientific standards for any new experiment and that the conclusions can only be as strong as the rigor of the experiments allows.

There remains one statement that I think is not justified by the data, which is the following revised sentence in the abstract: "increased activity of putative OXT- receptor-positive neurons in the vLS, and decreased activity of putative AVP-receptor-positive neurons in the dLS, were essential to evoke female aggression". The authors base this on combining in vitro (activity in neurons in vLS and dLS) with in vivo data (female aggression) and therefore in my opinion, the word "essential" is over the top because this link has not been demonstrated and something like "are likely to underlie female aggression" is much better.

The authors indicate that they "added new data showing that the administration of V1aR-A (750ng/5µl) on its own (i.c.v.) does not affect the levels of aggression displayed by highly aggressive IST females (Figure 3d)." Unfortunately, I couldn't find this data in Fig 3 nor in supplementary Fig 3. This is important because without this data the statement "since icv synthetic OXT decreased aggression via activation of V1aRs" (page 6, line 1) and similar statements made elsewhere in the manuscript are problematic.

The authors wished to include the glucocorticoid data because "this is the first paper on this novel animal model on virgin female aggression", but that seems incorrect because the authors published a paper on this novel animal model in 2014 in PLoS One.

Regarding the microdialysis data, I respectfully disagree with the authors indicating that "as there is no difference between the groups regarding baseline levels of OXT, an increased percentage of release ultimately reflects an increased OXT content in microdialysates sampled during the FIT." The authors are probably also aware that a percentage is certainly not the same as content and statistical analysis on these two different parameters can yield different results. Even with no baseline difference in absolute OXT concentrations, a conclusion about the absolute OXT concentrations during FIT can only be made if statistical analyses are performed on the absolute concentrations, not on the percentage. The authors could consider showing the absolute concentrations during FIT rather than the percentage.

Answers to Reviewer 1

Reviewer's comment: *This manuscript has further improved and I'm pleased to see that the authors made changes in the wording at critical points in the manuscript to tone down and be more careful with their statements and interpretations based on their data.*

Allow me to provide some heartfelt advice to the authors that their elaborated replies often went far beyond the comments raised and felt at times condescending and patronizing. Additionally, I believe the authors would agree with me that, irrespective of expertise and past validations, we have to uphold the highest scientific standards for any new experiment and that the conclusions can only be as strong as the rigor of the experiments allows.

Answer: We are pleased to read that the reviewer appreciated our changes in the revised manuscript. We also would like to emphasize that by no means we had the intention to patronize the reviewer in our answers to the reviewer's carefully addressed concerns. Indeed, we completely agree with the reviewer that the highest scientific standards should always be the major aim of experimental work including the careful interpretation of results. Accordingly, conclusions should be strongly drawn based on data displayed in a manuscript, and should also refer to existing knowledge and literature, as well as on past validations. Therefore, we are indeed thankful for the reviewer's careful reading and commenting on our manuscript. We should acknowledge that all of the reviewer's suggestions have substantially improved the quality of our manuscript.

Reviewer's comment: *There remains one statement that I think is not justified by the data, which is the following revised sentence in the abstract: "increased activity of putative OXT- receptor-positive neurons in the vLS, and decreased activity of putative AVP-receptor-positive neurons in the dLS, were essential to evoke female aggression". The authors base this on combining in vitro (activity in neurons in vLS and dLS) with in vivo data (female aggression) and therefore in my opinion, the word "essential" is over the top because this link has not been demonstrated and something like "are likely to underlie female aggression" is much better.*

Answer: We apologize for overseeing this statement, and in order to address the reviewer's concerns we now rephrased the sentence in the abstract as suggested: "**Accordingly, increased activity of putative OXT-receptor-positive neurons in the vLS, and decreased activity of putative AVP-receptor-positive neurons in the dLS, are likely underline female aggression**" (Page 1 lines 25-27). We sincerely hope that this change meets the reviewer's expectations.

Reviewer's comment: *The authors indicate that they "added new data showing that the administration of V1aR-A (750ng/5µl) on its own (i.c.v.) does not affect the levels of aggression displayed by highly aggressive IST females (Figure 3d)." Unfortunately, I couldn't find this data in Fig 3 nor in supplementary Fig 3. This is important because without this data the statement "since icv synthetic OXT decreased aggression via activation of V1aRs" (page 6, line 1) and similar statements made elsewhere in the manuscript are problematic.*

Answer: We apologize **for not having highlighted** the added new data in Figure 3d in the second revised version of the manuscript. We realized that it was indeed difficult to recognize this new data set in Figure 3d, which could be easily overseen by the reader. We had included an additional experiment and an additional control group, i.e. administration of V1aR-A alone, as requested by the reviewer (please see **Figure 3 d** below). As can be seen in **Figure 3 d**, we added the new bar outlined by the **blue box**, showing the additional group included in the statistical comparison (V1aR-A/VEH), and added new information into the corresponding figure legend.

Figure 3. Endogenous OXT promotes, whereas synthetic AVP and OXT reduce female aggression. **a** Experimental design for pharmacological and chemogenetic experiments targeting the OXT and AVP systems in isolated and trained (IST) and group-housed (GH) rats. (AAV= adeno-associated DREADD virus infusion into the hypothalamic paraventricular (PVN) and supraoptic (SON) nuclei; AVP= vasopressin; arrow= drug infusions; FIT= female intruder test; IS= social isolation; OXT= oxytocin; OXTR-A= OXT receptor antagonist; SURG= surgery; V1aR-A= V1a receptor antagonist; WO= Wash-out). **b** I.c.v. infusion of OXT (50ng/5ul) increased aggression in GH (two-tailed Student's t-test $t_{(19)}=2.46$, $p=0.024$), **c** but decreased aggression in IST females ($t_{(8)}=2.33$, $p=0.048$, data corresponds to FIT 4 and 6). **d** I.c.v. infusion of V1aR-A (750ng/5ul), but not OXTR-A (750ng/5ul), blocked the anti-aggressive effects of OXT in IST females (one-way ANOVA followed by Bonferroni $F_{(3,28)}=10.1$, $p=0.001$). Also, the V1aR-A alone did not affect aggression. Both, **e** i.c.v. infusion of OXTR-A ($t_{(28)}=4.96$, $p<0.0001$) and **f** i.c.v. AVP (0.1 or 1ng/5ul) reduced total aggressive behavior ($F_{(3,54)}=7.48$, $p=0.0003$) in IST rats. **g** Chemogenetic activation of OXT neurons in the PVN and SON increased aggression only in metestrus-diestrus GH rats (two-way ANOVA, factor treatment: $F_{(1,19)}=3.342$, $p=0.083$; estrous cycle: $F_{(1,19)}=6.68$, $p=0.018$; treatment x estrous cycle: $F_{(1,19)}=6.45$, $p=0.02$). **h** Confirmation of virus infection in the PVN (right) and SON (left). OXT-neurophysin I staining: green; mCherry (virus): red. Scale bars 300um. Data are shown as mean + SEM. * $p<0.05$; ** $p<0.01$; *** $p<0.0001$ vs either vehicle or control; ^o $p<0.05$ vs met-diestrus. OXT: GH: n=9-12; IST: n=9; AVP: n=9-18; OXTR-A: n=14-15; Combination OXT/OXTR-A/V1aR-A: n= 7-9; Chemogenetics: n= 6-15. Control group consisted of: i) rAAV1/2 OXTpr-mCherry+CNO, no virus infusion + ii) saline or iii) CNO.

As the reviewer may see, i.c.v. administration of V1aR-A alone did not affect female aggression in IST rats (second bar from left on Figure 3d in dark red), thus corroborating our conclusion that the anti-aggressive effects of **synthetic OXT** are indeed mediated via **V1aRs**. We hope that the present detailed explanation fully addresses the reviewer's remaining concerns.

Reviewer's comment: The authors wished to include the glucocorticoid data because "this is the first paper on this novel animal model on virgin female aggression", but that seems incorrect because the authors published a paper on this novel animal model in 2014 in PLoS One.

Answer: The reviewer is absolutely correct that we have already published a paper on female aggression (de Jong et al in 2014, PLoSOne) where we described the **behavioral test** used to measure aggressive behavior in female Wistar rats, namely the **Female Intruder Test (FIT)**. This **behavioral test** was introduced to characterize and quantify aggression levels in a big cohort of virgin and naïve female Wistar rats. In these otherwise non-manipulate female rats, aggression levels were measurable but very low (**below 10%**, similarly to those in group-housed females of the present study). In fact, those mild levels of aggression seen in the de Jong et al 2014 study motivated us to establish our **novel rat model** of exaggerated female aggression described in the present manuscript.

Therefore, in our current manuscript, we describe and validate this new **animal model** of exacerbated aggression in virgin female Wistar rats as stated in the manuscript: "In order to study the neurobiological mechanisms underlying the aggression of virgin female Wistar rats (defined

here as female aggression), we first established an animal model to robustly enhance their mild levels of aggression” (Page 2, lines 17-19). We reinforce that having such contrasting high versus low levels of aggression displayed by isolated and trained and group-housed females, respectively, configures an exceptionally useful tool to study the neurobiology of aggression, in our case assessing the role of the neuropeptides OXT and AVP in female aggression, which would be impossible in naïve, group-housed female rats due to their low and variable levels of aggression as reported by us in 2014.

However, the reviewer is correct in that the **behavioral test (FIT)** used to quantify female aggression levels, has already been published in the 2014 paper, which is also why we cited de Jong et al., 2014 in our introduction and methods (please see **pages 1 and 14, lines 19 and 17**, respectively).

We want to emphasize that the animal model of enhanced female aggression used here combining social isolation and aggression-training has never been published elsewhere. Therefore, as we explained in our answer to the reviewer in the second rebuttal letter, we find it important to properly validate the model in terms of predictive, construct, and face validity. That is also the reason to include the data on plasma CORT, as CORT has been extensively linked with intermale aggression. We also believe that this data placed in Supplementary Fig. 2 is minor and not suited in the center of major interpretations. Nevertheless, we still believe that it is relevant to be included in the manuscript to better characterize and **validate the novel animal model** of high female aggression as stated: **“Since low glucocorticoid levels have been implicated in the development of intermale aggression in humans and animals, we assessed plasma corticosterone concentrations after exposure to the FIT. Although there was no effect of housing or training conditions, plasma corticosterone indeed negatively correlated with aggression in females (Supplementary figure 2e)”** (Page 4, line 30-32). We sincerely hope that the reviewer can understand and agree with our point of view.

Reviewer's comment: Regarding the microdialysis data, I respectfully disagree with the authors indicating that "as there is no difference between the groups regarding baseline levels of OXT, an increased percentage of release ultimately reflects an increased OXT content in microdialysates sampled during the FIT." The authors are probably also aware that a percentage is certainly not the same as content and statistical analysis on these two different parameters can yield different results. Even with no baseline difference in absolute OXT concentrations, a conclusion about the absolute OXT concentrations during FIT can only be made if statistical analyses are performed on the absolute concentrations, not on the percentage. The authors could consider showing the absolute concentrations during FIT rather than the percentage.

Answer: We thank the reviewer to identify our weak statement. We completely agree with the reviewer that we used incorrect wording here, and conclusions based on a mix of absolute peptide levels and relative increase should be strictly avoided. Moreover, the statistical analyses used to compare group differences in percentages and absolute values (e.g. content) are different. In this context, as the reviewer pointed out, conclusions about the absolute OXT content (in microdialysates) during FIT can only be made, if statistical analyses are performed on the absolute concentrations. Therefore, we tried hard to fulfill this requirement in the previous version (2nd rebuttal) and further added the formula used to calculate the percentage as suggested by the reviewer. To make this point clearer, we have rephrased this description in legends of Figure 4 and 5:

Figure 4: b IST, but not GH females showed an increased **rise in the percentage of OXT release** (OXT content in microdialysates sampled during of the FIT/OXT content in microdialysates sampled during Baseline x 100) in the vLS during the FIT (One sample Student's t-test IST: $t_{(7)}=2.65$, $p=0.033$; GH: $t_{(7)}=0.83$, $p=0.43$), thus OXT release during FIT tended to be higher in IST compared with GH rats ($t_{(14)}=2.12$, $p=0.053$). Insert shows that absolute OXT content in microdialysates sampled under basal conditions did not differ between the groups ($t_{(14)}= 0.54$, $p=0.60$).

Figure 5: b GH, but not IST females showed an increased **rise in the percentage of AVP** release (AVP content in microdialysates sampled during the FIT/AVP content in microdialysates sampled during Baseline x 100) in the dLS during the FIT (Wilcoxon Signed Rank test GH: $W_{(9)}=45$, $p=0.0039$; IST: $W_{(7)}=-4.00$, $p=0.81$), thus AVP release during the FIT was higher in GH than in IST rats (Mann-Whitney U-test $U=6.00$, $p=0.0052$). Insert shows that absolute AVP content in microdialysates sampled under basal conditions did not differ between the groups.

We hope that our clarifications and revised version of the manuscript had properly addressed all reviewer's concerns, allowing us to publish this work.

Reviewer #1 (Remarks to the Author):

The authors have addressed my final concerns satisfactorily and I recommend publication in this journal. I do have one final suggestion and that is to make the colors for the V1aR-A/vehicle, the V1aR-A/OXT, and the AVP 1 ng groups in Fig 3 more distinctive. At least on my screen these colors look very similar.